# CARD: Channel Aligned Robust Blend Transformer for Time Series Forecasting

**Xue Wang**[*]   **Tian Zhou**[*]   **Qingsong Wen**[†]   **Jinyang Gao**   **Bolin Ding**   **Rong Jin**[‡]
`{xue.w,tian.zt,qingsong.wen,jinyang.gjy,bolin.ding,jinrong.jr}@alibaba-inc.com`

## Abstract

Recent studies have demonstrated the great power of Transformer models for time series forecasting. One of the key elements that lead to the transformer's success is the channel-independent (CI) strategy to improve the training robustness. However, the ignorance of the correlation among different channels in CI would limit the model's forecasting capacity. In this work, we design a special Transformer, i.e., **C**hannel **A**ligned **R**obust Blen**d** Transformer (CARD for short), that addresses key shortcomings of CI type Transformer in time series forecasting. First, CARD introduces a channel-aligned attention structure that allows it to capture both temporal correlations among signals and dynamical dependence among multiple variables over time. Second, in order to efficiently utilize the multi-scale knowledge, we design a token blend module to generate tokens with different resolutions. Third, we introduce a robust loss function for time series forecasting to alleviate the potential overfitting issue. This new loss function weights the importance of forecasting over a finite horizon based on prediction uncertainties. Our evaluation of multiple long-term and short-term forecasting datasets demonstrates that CARD significantly outperforms state-of-the-art time series forecasting methods. The code is available at the following repository: `https://github.com/wxie9/CARD`.

## 1 Introduction

Time series forecasting has emerged as a crucial task in various domains such as cloud computing, air quality forecasting, energy management, and traffic flow estimation(Qian et al., 2022; Liang et al., 2023; Zhu et al., 2023; Wen et al., 2023a). The rapid development of deep learning models has led to significant advancements in time series forecasting techniques, particularly in multivariate time series forecasting. Among various deep learning models developed for time series forecasting, RNN, CNN, MLP, transformer, and LLM-based models have demonstrated great performance thanks to their ability to capture complex long-term temporal dependencies (e.g., Zhou et al., 2021; Challu et al., 2022; Zeng et al., 2023; Zhou et al., 2022a; Wu et al., 2023b; Zhou et al., 2023; Jin et al., 2023).

For multivariate time series forecasting, a model is expected to yield a better performance by exploiting the dependence among different prediction variables, so-called channel-dependent (CD) methods. However, multiple recent works (e.g., Nie et al. 2023; Zeng et al. 2023) show that, in general, channel-independent (CI) forecasting models (i.e., all the time series variables are forecast independently) outperform the CD models. Analysis from (Han et al., 2023) indicates that CI forecasting models are more robust while CD models have higher modeling capacity. Given that time series forecasting usually involves high noise levels, typical transformer-based forecasting models with CD design can suffer from the issue of overfitting noises, leading to limited performance. These empirical studies and analyses raised an important question, i.e., how to build an effective transformer to utilize the cross-channel information for time series forecasting.

In this paper, we propose a **C**hannel **A**ligned **R**obust Blen**d** Transformer, or CARD for short, that effectively leverages the dependence among channels (i.e., forecasting variables) and alleviates the issue of overfitting noises in time series forecasting. Unlike typical transformers for time series

---

* Equal contribution.
† Work done at Alibaba Group, and now affiliated with Squirrel AI.
‡ Work done at Alibaba Group, and now affiliated with Meta.

analysis that only capture temporal dependency among signals through attention over tokens, the CARD also takes attention across different channels and hidden dimensions, which captures the correlation among prediction variables and aligns local information within each token. We observe that related approaches have been exploited in computer vision (Ding et al., 2022; Ali et al., 2021). Moreover, it is known that multi-scale information plays an important role in time series analysis. We design a token blend module to generate tokens with different resolutions. In particular, we propose to combine the adjacent tokens within the same head into the new token instead of merging the same position over different heads in multi-head attention. To improve the robustness and efficiency of the transformer for time series forecast, we further introduce an exponential smoothing layer over queries/keys tokens and a dynamic projection module when dealing with information among different channels. Finally, to alleviate the issue of overfitting noises, a robust loss function is introduced to weight each prediction by its uncertainty in the case of forecasting over a finite horizon. The overall model architecture is illustrated in Figure 1. We verify the effectiveness of the proposed model on various numerical benchmarks by comparing it to the state-of-the-art methods for Transformers and other models. Here we summarized our key contributions as follows:

1. We propose a **C**hannel **A**ligned **R**obust Blen**d** Transformer (CARD) which efficiently and robustly aligns the information among different channels and utilizes the multi-scale information.

2. CARD demonstrates superior performance in several benchmark datasets for forecasting and other prediction-based tasks, outperforming the state-of-the-art models. Our studies have confirmed the effectiveness of the proposed model.

3. We develop a robust signal decay-based loss function that utilizes signal decay to bolster the model's ability to concentrate on forecasting for the near future. Our empirical assessment has confirmed that this loss function is effective in improving the performance of other benchmark models as well.

The remainder of this paper is structured as follows. In Section 2, we provide a summary of related works relevant to our study. Section 3 presents the proposed detailed model architecture. Section 4 describes the loss function design with a theoretical explanation via maximum likelihood estimation of Gaussian and Laplacian distributions. In Section 5, we demonstrate the results of the numerical experiments in forecasting benchmarks and conduct a comprehensive analysis to determine the effectiveness of the self-attention scheme for time series forecasting. Additionally, we discuss ablations and other experiments conducted in this study. Finally, in Section 6, the conclusions and future research directions are discussed.

## 2 RELATED WORK

### 2.1 TRANSFORMERS FOR TIME SERIES FORECASTING

There is a large body of work that tries to apply Transformer models to forecast long-term time series in recent years (Wen et al., 2023b). We here summarize some of them. LogTrans (Li et al., 2019a) uses convolutional self-attention layers with LogSparse design to capture local information and reduce space complexity. Informer (Zhou et al., 2021) proposes a ProbSparse self-attention with distilling techniques to extract the most important keys efficiently. Autoformer (Wu et al., 2021) borrows the ideas of decomposition and auto-correlation from traditional time series analysis methods. FEDformer (Zhou et al., 2022b) uses Fourier enhanced structure to get a linear complexity. Pyraformer (Liu et al., 2022a) applies pyramidal attention module with inter-scale and intra-scale connections which also get a linear complexity. LogTrans avoids a point-wise dot product between the key and query, but its value is still based on a single time step. Autoformer uses autocorrelation to get patch-level connections, but it is a handcrafted design that doesn't include all the semantic information within a patch. A recent work PatchTST (Nie et al., 2023) studies using a vision transformer type model for long-term forecasting with channel independent design. The work closest to our proposed method is Crossformer (Zhang & Yan, 2023). This work designs an encoder-decoder model utilizing a hierarchy attention mechanism to leverage cross-dimension dependencies and achieves moderate performance in the same benchmark datasets that we use in this work. From the model architecture perspective, different from Crossformer, we employ an encoder-only structure, and the multi-scale information is induced via a lightweight token blend module instead of explicitly generating token

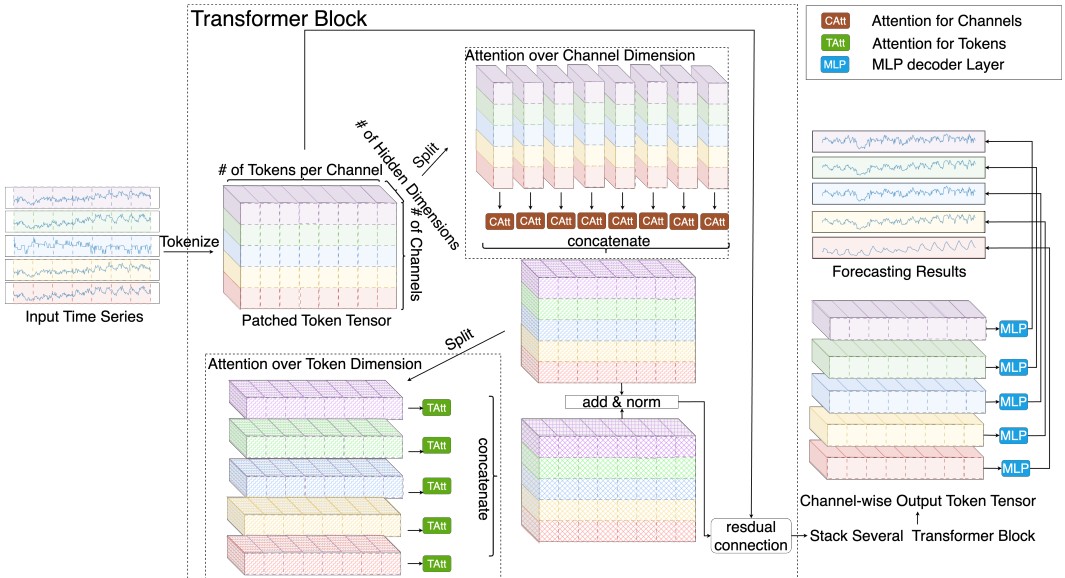

Figure 1: Illustration of the architecture of CARD.

hierarchies used in Crossformer. The designs significantly enhance the robustness of CARD and result in a substantial improvement in numerical performance.

## 2.2 RNN, MLP AND CNN MODELS FOR TIME SERIES FORECASTING

Besides transformers, other types of networks are also widely explored. For example, (Lai et al., 2018; Lim et al., 2021; Salinas et al., 2020; Smyl, 2020; Wen et al., 2017; Rangapuram et al., 2018; Zhou et al., 2022a; Gu et al., 2022) study the RNN/state-space models. In particular, (Smyl, 2020) considered equipping RNN with exponential smooth and, for the first time, beat the statistical models in forecasting tasks (Makridakis et al., 2018). (Chen et al., 2023; Oreshkin et al., 2020; Challu et al., 2022; Li et al., 2023; Zeng et al., 2023; Das et al., 2023; Zhang et al., 2022) explored MLP-type structures for time series forecasting. CNN models (e.g., Wu et al. 2023b; Wen et al. 2017; Sen et al. 2019) use the temporal convolution layer to extract the subsequence-level information. When dealing with multivariate forecasting tasks, the smoothness in adjacent covariates is assumed or the channel-independent strategy is used.

## 3 MODEL ARCHITECTURE

The illustration of the architecture of CARD is shown in Figure 1. Let $\boldsymbol{a}_t \in \mathbb{R}^C$ be the observation of time series at time $t$ with channel $C \geq 1$. Our objective is to use $L$ recent historical data points (e.g., $\boldsymbol{a}_{t-L+1}, ..., \boldsymbol{a}_t$) to forecast the future $T$ steps observations. (e.g., $\boldsymbol{a}_{t+1}, ..., \boldsymbol{a}_{t+T}$), where $L, T \geq 1$.

### 3.1 TOKENIZATION

We adopt the idea of patching (e.g., Nie et al. 2023; Zhang & Yan 2023) to convert the input time series into a token tensor. Let's denote $\boldsymbol{A} = [\boldsymbol{a}_{t-L+1}, ..., \boldsymbol{a}_t] \in \mathbb{R}^{C \times L}$ as the input data matrix, $S$ and $P$ as stride and patch length respectively. We unfold the matrix $\boldsymbol{A}$ into the raw token tensor $\tilde{X} \in \mathbb{R}^{C \times N \times P}$, where $N = \lfloor \frac{L-P}{S} + 1 \rfloor$. Here, we convert the time series into several $P$ length segments, and each raw token maintains part of the sequence-level semantic information, which makes the attention scheme more efficient compared to the vanilla point-wise counterpart.

We then use a dense MLP layer $F_1 : P \to d$, a extra token $\mathbf{T}_0 \in \mathbb{R}^{C \times d}$ and positional embedding $\boldsymbol{E} \in \mathbb{R}^{C \times N \times d}$ to generate the token matrix as follows:

$$\boldsymbol{X} = [\mathbf{T_0}, F_1(\tilde{\boldsymbol{X}}) + \boldsymbol{E}], \tag{1}$$

where $\boldsymbol{X} \in \mathbb{R}^{C \times (N+1) \times d}$ and $d$ is the hidden dimension. Compared to (Nie et al., 2023) and (Zhang & Yan, 2023), our token construction introduces a extra $\mathrm{T_0}$ token. The $\mathrm{T_0}$ token is an analogy to the *static covariate encoder* in (Lim et al., 2021) and allows us to have a place to inject the features summarized the longer history of the series.

We consider generating $\boldsymbol{Q}, \boldsymbol{K}$ and $\boldsymbol{V}$ via linear projection of the token tensor $\boldsymbol{X}$:

$$\boldsymbol{Q} = F_q(\boldsymbol{X}), \ \boldsymbol{K} = F_k(\boldsymbol{X}), \ \boldsymbol{V} = F_v(\boldsymbol{X}), \tag{2}$$

where $\boldsymbol{Q}, \boldsymbol{K}, \boldsymbol{V} \in \mathbb{R}^{C \times (N+1) \times d}$ and $F_q, F_k, F_v$ are MLP layers.

We next convert $\boldsymbol{Q}, \boldsymbol{K}, \boldsymbol{V}$ into $\{\boldsymbol{Q}_i\}, \{\boldsymbol{K}_i\}, \{\boldsymbol{V}_i\}$ where $\boldsymbol{Q}_i, \boldsymbol{K}_i, \boldsymbol{V}_i \in \mathbb{R}^{C \times (N+1) \times d_{\mathrm{head}}}$, $i = 1, 2, ..., H$. $H$ and $d_{\mathrm{head}}$ are number of heads and head dimension respectively. For each sample, the total number of tokens is $C(N+1)$. In order to fully utilize all cross-channel information, the ideal attention should be required $\mathcal{O}(C^2(N+1)^2)$ computation cost, which can be very time-consuming and potentially can lead to easily over-fitting when training sample size is limited. In this paper, we consider paying attention alternately over each dimension instead.

## 3.2 CARD ATTENTIONS OVER TOKENS

When make attention over tokens, we slice the $\boldsymbol{Q}_i, \boldsymbol{K}_i$ and $\boldsymbol{V}_i$ on channel dimension into $\{Q_i^{c:}\}$, $\{K_i^{c:}\}$ and $\{V_i^{c:}\}$ with $Q_i^{c:}, K_i^{c:}, V_i^{c:} \in \mathbb{R}^{(N+1) \times d_{\mathrm{head}}}$ and $c = 1, 2, ..., C$. Besides the standard attention in tokens, we also introduce an extra attention structure in hidden dimensions that helps capture the local information within each patch. The attention in both tokens and hidden dimensions is computed as follows:

$$\boldsymbol{A}_{i1}^{c:} = \mathrm{softmax}\left(\frac{1}{\sqrt{d}} \cdot \mathrm{EMA}(Q_i^{c:})\,(\mathrm{EMA}(K_i^{c:}))^\top\right) \tag{3}$$

$$\boldsymbol{A}_{i2}^{c:} = \mathrm{softmax}\left(\frac{1}{\sqrt{N}} \cdot (Q_i^{c:})^\top K_i^{c:}\right), \tag{4}$$

where $\boldsymbol{A}_{i1}^{c:} \in \mathbb{R}^{(N+1) \times (N+1)}$, $\boldsymbol{A}_{i2}^{c:} \in \mathbb{R}^{d_{\mathrm{head}} \times d_{\mathrm{head}}}$ and EMA denotes the Exponential Moving Average[1].

By applying EMA on $Q_i^{c:}$ and $K_i^{c:}$, each query token will be able to gain higher attention scores on more key tokens and thus the output becomes more robust. Similar techniques are also explored in (Ma et al., 2023) and (Woo et al., 2022). Different from those in the literature, we find that using a fixed EMA parameter that remains the same for all dimensions is enough to stabilize the training process. Thus, our EMA doesn't contain learnable parameters.

The outputs are computed as:

$$\boldsymbol{O}_{i1}^{c:} = \boldsymbol{A}_{i1}^{c:}\boldsymbol{V}_i^c, \quad \boldsymbol{O}_{i2}^{c:} = \boldsymbol{V}_i^{c:}\boldsymbol{A}_{i2}^{c:}. \tag{5}$$

We next apply the proposed token blend module to merge heads and generate tokens capturing multi-scale knowledge and the detailed discussions are deferred to section 3.4. The batch normalization (Ioffe & Szegedy, 2015) to $\boldsymbol{O}_{i1}^{c:}$ and $\boldsymbol{O}_{i2}^{c:}$ is then used to adjust the outputs' scale. Finally, the residual connection structure is used to generate the final output of the attention block.

The total number of tokens is on the order of $\mathcal{O}(L/S)$ per channel and the complexity in attention along tokens is upper bounded by $\mathcal{O}(C \cdot d^2 \cdot L^2/S^2)$, which is smaller than $\mathcal{O}(C \cdot d^2 \cdot L^2)$ complexity of the vanilla point-wise token construction. In practice, one can use efficient attention implementation (e.g., FlashAttention Dao et al. 2022) to further obtain nearly linear computational performance.

## 3.3 CARD ATTENTION OVER CHANNELS

We first compute $\{\boldsymbol{Q}_i\}, \{\boldsymbol{K}_i\}$ and $\{\boldsymbol{V}_i\}$ via Equation (2) and then slice them over token dimension into $\{Q_i^{:n}\}, \{K_i^{:n}\}$ and $\{V_i^{:n}\}$ with $Q_i^{:n}, K_i^{:n}, V_i^{:n} \in \mathbb{R}^{C \times d_{\mathrm{head}}}$ and $n = 1, 2, ..., N+1$. Due to the

---

[1]Formally, an EMA operator recursively calculates the output sequence $\{\boldsymbol{y}_i\}$ w.r.t. input sequence $\{\boldsymbol{x}_i\}$ as $\boldsymbol{y}_t = \alpha\boldsymbol{x}_t + (1-\alpha)\boldsymbol{y}_{t-1}$, where $\alpha \in (0,1)$ is the EMA parameter representing the degree of weighting decrease.

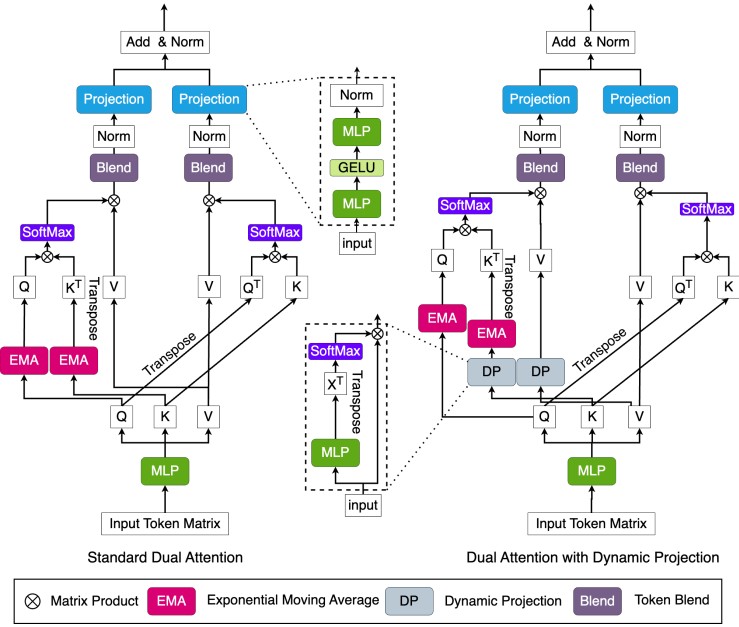

Figure 2: Architecture for the CARD attention block.

potential high-dimensionality issue of covariates, the vanilla method may suffer from computation overhead and overfitting. Take traffic dataset (PeMS) as an example, this dataset contains 862 covariates. When setting the lookback window size as 96, the attention over channels will require at least 80 times the computational cost of attention over tokens. The full attention will also merge a lot of noise patterns into the output token and lead to spurious correlation in the final forecasting results. In this paper, we consider using the dynamic projection technique (Zhu et al., 2021) to get "*summarized*" tokens to the $K_i^{:n}$ and $V_i^{:n}$ for $n$-th token dimension as shown in Figure 2. We first use MLP layers $F_{pk}$ and $F_{pv}$ to project head dimensions from $d_{\text{head}}$ to some fixed $r$ with $r \ll C$, and then we use softmax to normalized the projected tensors $\boldsymbol{P}_k^{:n}$ and $\boldsymbol{P}_v^{:n}$ as follow:

$$P_{ki}^{:n} = \text{softmax}(F_{pk}(K_i^{:n})), \quad P_{vi}^{:n} = \text{softmax}(F_{pv}(V_i^{:n})), \tag{6}$$

where $P_{ki}^{:n}, P_{vi}^{:n} \in \mathbb{R}^{C \times r}$. Next the "*summarized*" tokens are computed by

$$\tilde{K}_i^{:n} = (P_{ki}^{:n})^\top K_i^{:n}, \quad \tilde{V}_i^{:n} = (P_{vi}^{:n})^\top V_i^{:n}, \tag{7}$$

where $\tilde{K}_i^{:n}, \tilde{V}_i^{:n} \in \mathbb{R}^{r \times d_{\text{head}}}$.

Finally, the outputs are generated by applying $Q_i^{:n}$, $\tilde{K}_i^{:n}$ and $\tilde{V}^{:n}$ to equations from (3) to (5) for $n = 1, 2, ..., N+1$. The upper bound of total computational cost is reduced to $\mathcal{O}(L/S \cdot C \cdot r \cdot d^2)$ which is smaller than $\mathcal{O}(L/S \cdot C^2 \cdot d^2)$ cost of the standard attention.

### 3.4 TOKEN BLEND MODULE

Multi-scale knowledge plays a crucial role in forecasting tasks and has significantly enhanced the performance of diverse models. (e.g., Xu et al., 2021; Zeng et al., 2023; Wang et al., 2023b; Zhou et al., 2022b; Zhang & Yan, 2023). Most of these works initially decompose the time series into seasonal and trend components and then employ separate structures to process the seasonal and trend components individually. However, this approach, despite its simplicity, leads to higher model complexity, which in turn increases computation cost and makes it susceptible to overfitting issue.

In this work, we consider using a specially designed token blend mechanism to utilize the multi-scaling structural knowledge without additional computation costs. The token blend module replaces the standard token reconstruction after the multi-head attention by merging the adjacent token within the same head to produce the token for the next stage. The output token tensor $\boldsymbol{O}$ from the multi-head attention has 4-D with shape $C \times H \times (N+1) \times d_{\text{head}}$. The token blend module will first merge the second and third dimensions and reshape $\boldsymbol{O}$ into 3-D tensor with shape $C \times H(N+1) \times d_{\text{head}}$. We then decouple the second dimension into three dimensions, i.e., $H(N+1) \to h_1 \times h_2 \times h_3$ where

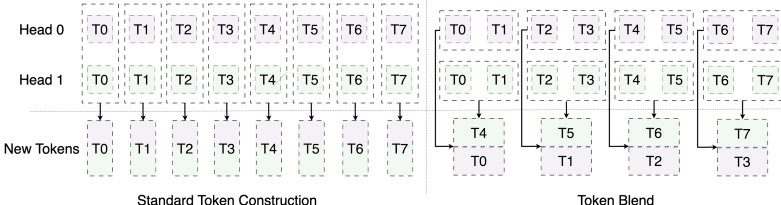

Figure 3: Illustration example of token blend block in CARD.

$h_1 = Hh_3$, $h_2 = N + 1$ and $h_3 \geq 1$. The final output $\boldsymbol{O}$ uses $h_3 \times h_1 \times d_{\text{head}}$ to construct the token dimension. Here we call $h_3$ as blend size. When $h_3 = 1$, the aforementioned operations generate the same outputs in the standard transformer. When $h_3 \geq 2$, the outputs will first combine the adjacent token within the same head, which would create the token that represents the knowledge over a larger range, i.e., lower resolution. With increasing the blend size $h_3$, more tokens within the same heads are merged and the attention module in the next stage could have more chance to capture long-term knowledge. An illustration example is shown in Figure 3. By consolidating temporally adjacent tokens within the same head, the resulting new tokens encompass knowledge over an extended time period. This enables more effective exploration of low-resolution knowledge by increasing attention on these tokens. Our token blend module is also different from the hierarchical adjacent tokens merging procedure in (Zhang & Yan, 2023). First, (Zhang & Yan, 2023) merges at the token level, the output token sequence at the coarse level has higher hidden dimensions and shorter sequence lengths. We consider merging at the head level instead which maintains the same output token sequence shape. Second, the merging size in (Zhang & Yan, 2023) is fixed as 2, while we allow a more flexible configuration. As a result, we achieve an implicit structure that enhances the extraction of multi-scale information without the need for an additional explicit signal disentanglement process.

## 4 SIGNAL DECAY-BASED LOSS FUNCTION

In this section, we discuss our loss function design. In literature, the Mean Squared Error (MSE) loss is commonly used to measure the discrepancy between the forecasting results and the ground truth observations. Let $\hat{\boldsymbol{a}}_{t+1}(\boldsymbol{A}), ...., \hat{\boldsymbol{a}}_{t+L}(\boldsymbol{A})$ and $\boldsymbol{a}_{t+1}(\boldsymbol{A}), ...., \boldsymbol{a}_{t+L}(\boldsymbol{A})$ be the predictions and real observations from time $t + 1$ to $t + L$ given historical information $\boldsymbol{A}$. The overall objective loss becomes:

$$\min \quad \mathbb{E}_{\boldsymbol{A}} \left[ \frac{1}{L} \sum_{l=1}^{L} \|\hat{\boldsymbol{a}}_{t+l}(\boldsymbol{A}) - \boldsymbol{a}_{t+l}(\boldsymbol{A})\|_2^2 \right]. \tag{8}$$

One drawback of plain MSE loss for forecasting tasks is that the different time steps' errors are equally weighted. In real practice, the correlation of historical information to far-future observations is usually smaller than that to near-future observations, implying that far-future observations have higher variance. Therefore, the near-future loss would contribute more to generalization improvement than the far-future loss. To see this, we assume that our time series follows the first-order Markov process, i.e., $\boldsymbol{a}_{t+1} \sim \mathcal{N}(G(\boldsymbol{a}_t), \sigma^2 I)$, where $G$ is the smooth transition function with Lipschitz constant 1, $\sigma > 0$ and $t = 1, 2, ....$. Then, we have

$$\text{var}(\boldsymbol{a}_{t+1}) = \text{var}(G(\boldsymbol{a}_t)) + \sigma^2 I \preceq \text{var}(\boldsymbol{a}_t) + \sigma^2 I, \tag{9}$$

where $\text{var}(\boldsymbol{a})$ denote the covariance matrix of $\boldsymbol{a}$. By recursively using Equation (9) from $t + L$ to $t$ and for all $l \in [t, t + L]$, we have

$$\text{var}(\boldsymbol{a}_{t+l}) \preceq l\sigma^2 I + \text{var}(\boldsymbol{a}_t). \tag{10}$$

When $\boldsymbol{a}_t$ is already observed, we have $\text{var}(\boldsymbol{a}_t) = 0$ and Equation (10) implies $\text{var}(\boldsymbol{a}_{t+l}) \preceq l\sigma^2 I$. If we use negative log-likelihood estimation over Gaussian distribution, we come up with the following approximated loss function:

$$\min \quad \mathbb{E}_{\boldsymbol{A}} \left[ \frac{1}{2} \sum_{l=1}^{L} (\hat{\boldsymbol{a}}_{t+l}(\boldsymbol{A}) - \boldsymbol{a}_{t+l}(\boldsymbol{A}))^\top \text{var}(\boldsymbol{a}_{t+l})^{-1} (\hat{\boldsymbol{a}}_{t+l}(\boldsymbol{A}) - \boldsymbol{a}_{t+l}(\boldsymbol{A})) \right]$$

$$\geq \mathbb{E}_{\boldsymbol{A}} \left[ \frac{1}{2} \sum_{l=1}^{L} \frac{\|\hat{\boldsymbol{a}}_{t+l}(\boldsymbol{A}) - \boldsymbol{a}_{t+l}(\boldsymbol{A})\|_2^2}{l\sigma^2} \right] \propto \mathbb{E}_{\boldsymbol{A}} \left[ \frac{1}{L} \sum_{l=1}^{L} l^{-1} \|\hat{\boldsymbol{a}}_{t+l}(\boldsymbol{A}) - \boldsymbol{a}_{t+l}(\boldsymbol{A})\|_2^2 \right]. \tag{11}$$

Compared Equation (11) to Equation (8), the far-future loss is scaled down to address the high variance. Since Mean Absolute Error (MAE) is more resilient to outliers than square error, we propose to use the loss function in the following form:

$$\min \mathbb{E}_{\boldsymbol{A}} \left[ \frac{1}{L} \sum_{l=1}^{L} l^{-1/2} \left\| \hat{\boldsymbol{a}}_{t+l}(\boldsymbol{A}) - \boldsymbol{a}_{t+l}(\boldsymbol{A}) \right\|_1 \right], \tag{12}$$

where Equation (12) can be derived via Equation (11) with replacing the Gaussian distribution by Laplace distribution.

## 5 EXPERIMENTS

### 5.1 LONG TERM FORECASTING

**Datasets** We conducted experiments on seven real-world benchmarks, including four Electricity Transform Temperature (ETT) datasets (Zhou et al., 2021) comprising of two hourly and two 15-minute datasets, one 10-minute weather forecasting dataset (Wetterstation), one hourly electricity consumption dataset (UCI), and one hourly traffic road occupancy rate dataset (PeMS).

**Baselines and Experimental Settings** We use the following recent popular models as baselines: FED-former (Zhou et al., 2022b), ETSformer (Woo et al., 2022), FiLM (Zhou et al., 2022a), LightTS (Zhang et al., 2022), MICN (Wang et al., 2023b), TimesNet (Wu et al., 2023b), Dlinear (Zeng et al., 2023), Crossformer (Zhang & Yan, 2023), and PatchTST (Nie et al., 2023). We use the experimental settings in (Wu et al., 2023b) applying reversible instance normalization (RevIN, Kim et al., 2022) to handle data heterogeneity and keeping the lookback length as 96 for fair comparisons. Each setting is repeated 10 times and average MSE/MAE results are reported. The full results are summarized in Table 7 in the Appendix. More details on model configurations, model code, and comparison with other early baselines can be found in Appendix D and Appendix B, respectively.

**Results** The results are summarized in Table 1. Regarding the average performance across four different output horizons, CARD gains the best performance in 6 out of 7 and 7 out of 7 in MSE and MAE, respectively. In single-length experiments, CARD achieves the best results in **82%** cases in MSE metric and **100%** cases in MAE metric.

For problems with complex covariate structures, the proposed CARD method beats the benchmarks by significant margins. For instance, in Electricity (321 covariates), CARD consistently outperforms the second-best algorithm by reducing MSE/MAE by more than **9.0%** on average in each forecasting horizon experiment. By leveraging 21 covariates for Weather and 862 covariates for Traffic, we achieve a large reduction in MSE/MAE of over **7.5%**. This highlights CARD's exceptional capability to incorporate extensive covariate information for improved prediction outcomes. Furthermore, Crossformer (Zhang & Yan, 2023) employs a comparable concept of integrating cross-channel data to enhance predictive accuracy. Remarkably, CARD significantly reduces the MSE/MAE by over **20%** on 6 benchmark datasets compared to Crossformer, which shows our attention design is much more effective in utilizing cross-channel information. It's also important to note that while Dlinear shows strong performance in those tasks using an MLP-based model, CARD still consistently reduces MSE/MAE by **5%** to **27.5%** across all benchmark datasets.

Recent works, such as Nie et al. 2023; Zeng et al. 2023; Zhang & Yan 2023) use the input length other than 96 and have shown performance improvement. In our study, we also report the numerical performance of CARD with a varying lookback length in Appendix G, and CARD consistently outperforms all baseline models when prolonging input sequence as well, demonstrating significantly lower MSE errors across all benchmark datasets.

### 5.2 RECONSTRUCTION BASED ANOMALY DETECTION

Reconstruction based anomaly detection can be viewed as a task to predict the input itself. In previous works, the reconstruction is a classical task for unsupervised point-wise representation learning, where the reconstruction error is a natural anomaly criterion. We follow the experimental settings in (Wu et al., 2023a) and consider five widely used anomaly detection benchmarks. The results are summarized in Table 2. CARD outperforms the existing best result by 3% in F1 score on average. In

Table 1: Long-term forecasting tasks. The lookback length is set as 96. All models are evaluated on 4 different prediction horizons {96, 192, 336, 720} and average MSE/MAE results of ten repeats are reported. The best model is in boldface and the second best is underlined.

| Models | CARD | | PatchTST | | MICN | | TimesNet | | Crossformer | | Dlinear | | LightTS | | FilM | | ETSformer | | FEDformer | |
|---|---|---|---|---|---|---|---|---|---|---|---|---|---|---|---|---|---|---|---|---|
| Metric | MSE | MAE | MSE | MAE | MSE | MAE | MSE | MAE | MSE | MAE | MSE | MAE | MSE | MAE | MSE | MAE | MSE | MAE | MSE | MAE |
| ETTm1 | **0.383** | **0.383** | 0.395 | 0.408 | _0.387_ | 0.411 | 0.400 | _0.406_ | 0.435 | 0.417 | 0.403 | 0.407 | 0.435 | 0.437 | 0.408 | 0.399 | 0.429 | 0.425 | 0.448 | 0.452 |
| ETTm2 | **0.271** | **0.316** | _0.283_ | _0.327_ | 0.284 | 0.340 | 0.291 | 0.333 | 0.609 | 0.521 | 0.350 | 0.401 | 0.409 | 0.436 | 0.287 | 0.328 | 0.292 | 0.342 | 0.305 | 0.349 |
| ETTh1 | _0.443_ | **0.429** | 0.455 | _0.444_ | **0.440** | 0.462 | 0.458 | 0.450 | 0.486 | 0.481 | 0.456 | 0.452 | 0.491 | 0.479 | 0.461 | 0.456 | 0.452 | 0.510 | **0.440** | 0.460 |
| ETTh2 | **0.367** | **0.390** | _0.384_ | _0.406_ | 0.402 | 0.437 | 0.414 | 0.427 | 0.966 | 0.690 | 0.559 | 0.515 | 0.602 | 0.543 | _0.384_ | _0.406_ | 0.439 | 0.452 | 0.437 | 0.449 |
| Weather | **0.240** | **0.262** | 0.257 | _0.280_ | _0.243_ | 0.299 | 0.259 | 0.287 | 0.250 | 0.310 | 0.265 | 0.317 | 0.261 | 0.312 | 0.269 | 0.339 | 0.271 | 0.334 | 0.309 | 0.360 |
| Electricity | **0.169** | **0.258** | 0.216 | 0.318 | _0.187_ | _0.295_ | 0.192 | _0.295_ | 0.273 | 0.363 | 0.212 | 0.300 | 0.229 | 0.329 | 0.223 | 0.303 | 0.208 | 0.323 | 0.214 | 0.327 |
| Traffic | **0.450** | **0.278** | _0.488_ | 0.327 | 0.542 | _0.316_ | 0.620 | 0.336 | 0.593 | 0.332 | 0.625 | 0.383 | 0.622 | 0.392 | 0.639 | 0.389 | 0.621 | 0.396 | 0.610 | 0.376 |

particular, CARD achieves 14.2% significant improvement in SMAP task. Those facts imply CARD could generate meaningful representation on time series.

Table 2: Anomaly detection. F1 scores are reported. The best model is in boldface and the second best is underlined.

| Models | CARD | PatchTST | MICN | TimesNet | Crossformer | ETSformer | LightTS | Dlinear | FEDformer | Stationary | Autoformer | Informer |
|---|---|---|---|---|---|---|---|---|---|---|---|---|
| SMD | **0.872** | _0.866_ | 0.800 | 0.858 | 0.778 | 0.831 | 0.825 | 0.771 | 0.851 | 0.847 | 0.851 | 0.855 |
| MSL | 0.817 | 0.823 | 0.816 | **0.852** | 0.820 | _0.850_ | 0.790 | 0.849 | 0.786 | 0.775 | 0.791 | 0.841 |
| SMAP | **0.857** | 0.695 | 0.656 | _0.715_ | 0.674 | 0.695 | 0.692 | 0.693 | 0.708 | 0.711 | 0.711 | 0.699 |
| SWaT | **0.945** | 0.909 | 0.875 | 0.921 | 0.886 | 0.849 | _0.933_ | 0.875 | 0.932 | 0.799 | 0.927 | 0.814 |
| PSM | 0.957 | 0.951 | 0.933 | **0.975** | 0.921 | 0.918 | 0.972 | 0.936 | 0.972 | _0.973_ | 0.933 | 0.771 |
| Avg | **0.890** | 0.849 | 0.816 | _0.864_ | 0.816 | 0.829 | 0.842 | 0.825 | 0.849 | 0.821 | 0.843 | 0.789 |

## 5.3 BOOSTING EFFECT OF SIGNAL DECAY-BASED LOSS FUNCTION

In this section, we present the boosting effect of our proposed signal decay-based loss function. In contrast to the widely used MSE loss function employed in previous training of long-term sequence forecasting models, our approach yields a reduction in MSE ranging from **3%** to **12%** across a spectrum of recent state-of-the-art baseline models, including Transformer, CNN, and MLP architectures as shown in Table 3. Our proposed loss function specifically empowers FEDformer and Autoformer, two algorithms that heavily rely on frequency domain information. This aligns with our signal decay paradigm, which acknowledges that frequency information carries variance/noise across time horizons. Our novel loss function can be considered a preferred choice for this task, owing to its superior performance compared to the plain MSE loss function. More detailed discussions are deferred to Section J in Appendix.

Table 3: Influence for signal decay-based loss function. The lookback length is set as 96. All models are evaluated on 4 different predication lengths {96, 192, 336, 720}. The average results are reported, and the full table is deferred to Table 19 in the Appendix. The model name with * uses the robust loss proposed in this work. The better results are in boldface.

| Models | CARD | | CARD* | | MICN-regre | | MICN-regre* | | TimesNet | | TimesNet* | | FEDformer | | FEDformer* | | Autoformer | | Autoformer* | |
|---|---|---|---|---|---|---|---|---|---|---|---|---|---|---|---|---|---|---|---|---|
| Metric | MSE | MAE | MSE | MAE | MSE | MAE | MSE | MAE | MSE | MAE | MSE | MAE | MSE | MAE | MSE | MAE | MSE | MAE | MSE | MAE |
| ETTm1 | 0.390 | 0.399 | **0.383** | **0.383** | 0.392 | 0.414 | **0.383** | **0.393** | 0.400 | 0.406 | **0.392** | **0.395** | 0.448 | 0.452 | **0.413** | **0.415** | 0.588 | 0.528 | **0.523** | **0.475** |
| ETTh1 | 0.449 | 0.440 | **0.443** | **0.425** | 0.559 | 0.535 | **0.527** | **0.499** | 0.458 | 0.450 | **0.449** | **0.438** | 0.440 | 0.460 | **0.436** | **0.442** | **0.496** | 0.487 | 0.514 | **0.481** |

## 5.4 INFLUENCE OF INPUT SEQUENCE LENGTH

Previous research (Zeng et al., 2023; Wen et al., 2023b) has highlighted a critical issue with the existing long-term forecasting transformers. They struggle to leverage extended input sequences, resulting in a decline in performance as the input length increases. We assert that this is not an inherent drawback of transformers, and CARD demonstrates robustness in handling longer and noisier historical sequence inputs, as evidenced by an **8.6%** and **8.9%** reduction in MSE achieved in

Table 4: Influence of prolonging input sequence. The lookback length is set as 96,192,336,720.

| Input Length | 96 | | 192 | | 336 | | 512 | | 720 | |
|---|---|---|---|---|---|---|---|---|---|---|
| Metric | MSE | MAE | MSE | MAE | MSE | MAE | MSE | MAE | MSE | MAE |
| ETTm1 | 0.383 | 0.384 | 0.363 | 0.372 | 0.352 | 0.367 | 0.402 | 0.420 | **0.349** | **0.368** |
| ETTh1 | 0.442 | 0.429 | 0.429 | 0.425 | 0.415 | 0.422 | 0.352 | 0.371 | **0.405** | **0.421** |

the ETTh1 and ETTm1 datasets, respectively, when input lengths were extended from 96 to 720, as shown in Table 4.

## 5.5 INFLUENCE OF TOKEN BLEND SIZE

In this section, we test the effect of the token blend module by varying blend size. The results are summarized in Figure 4. When setting the blend size to 1, the token blend module reduces to the standard token mix method in Transformer literature and we observe test errors in both MSE/MAE increase. While using a larger blend size, the multi-scale information is utilized and the errors are reduced in turn. However, in some cases, further increasing the blend size may damage the performance. we conjecture it is due to the nature of the dataset that only some scales of knowledge are useful for forecasting. A higher blend size may oversmooth that knowledge.

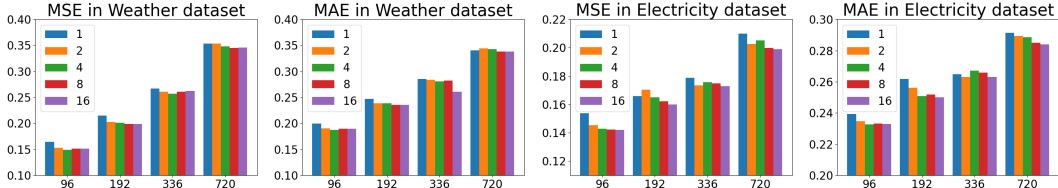

Figure 4: Experiments on token blend size. The blend size is varying in 1, 2, 4, 8, and 16.

## 5.6 OTHER EXPERIMENTS

We conduct a series of experiments, using both ablation and architecture variants, to evaluate each component in our proposed model. Our findings reveal that the channel branch made the greatest contribution to the reduction of MSE errors, as shown in Appendix Q.2. Furthermore, our experiments on sequential/parallel attention mixing design, detailed in Appendix Q.1, show that our model design is the preferred option. Visual aids and attention maps can be found in Appendix A and O, which effectively demonstrate our accurate predictions and utilization of covariate information. Another noteworthy experiment, concerning the impact of training data size, is presented in Appendix R.2. This study revealed that using 70% of training samples can significantly improve performance for half datasets affected by distribution shifts. Besides, Appendix L presents an error bar statistics table that demonstrates the robustness of CARD.More forecasting experiments on M4 (Makridakis et al., 2018) other datasets are presented in Appendix H and I.

## 6 CONCLUSION AND FUTURE WORKS

In this paper, we present a novel Transformer model, CARD, for time series forecasting. CARD is a channel-dependent model that aligns information across different variables and hidden dimensions effectively. CARD improves traditional transformers by applying attention to both tokens and channels. The new design of the attention mechanism helps explore local information within each token, making it more effective for time series forecasting. We also propose a token blend module to utilize the multi-scale information knowledge in time series. Furthermore, we introduce a robust loss function to alleviate the issue of overfitting noises, an important issue in time series analysis. As demonstrated through various numerical benchmarks, our proposed model outperforms state-of-the-art models.

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

## A  VISUALIZATION

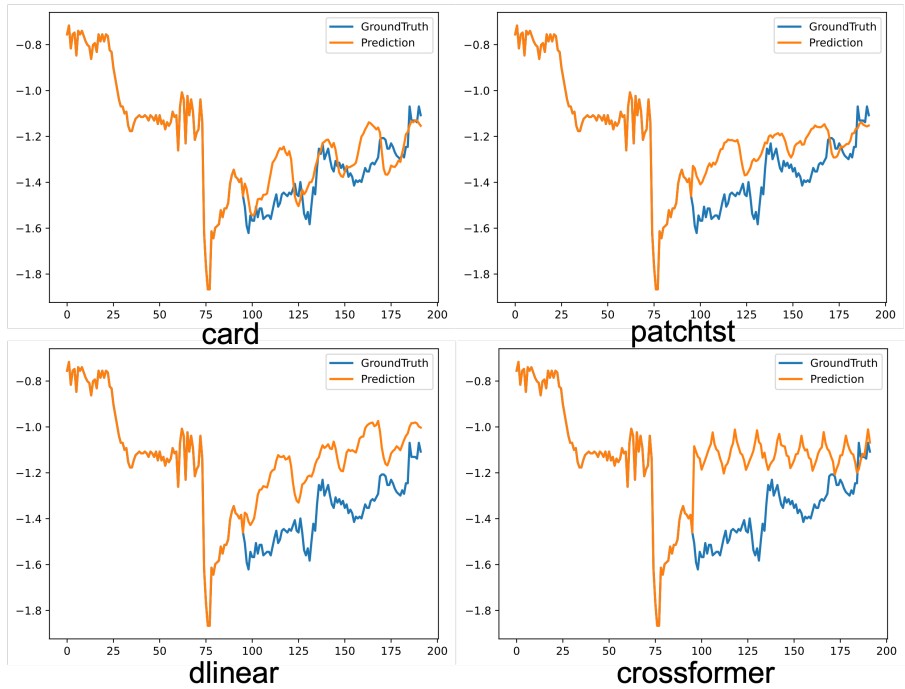

Figure 5: Sample prediction graph for ETTh1 long-term forecasting task

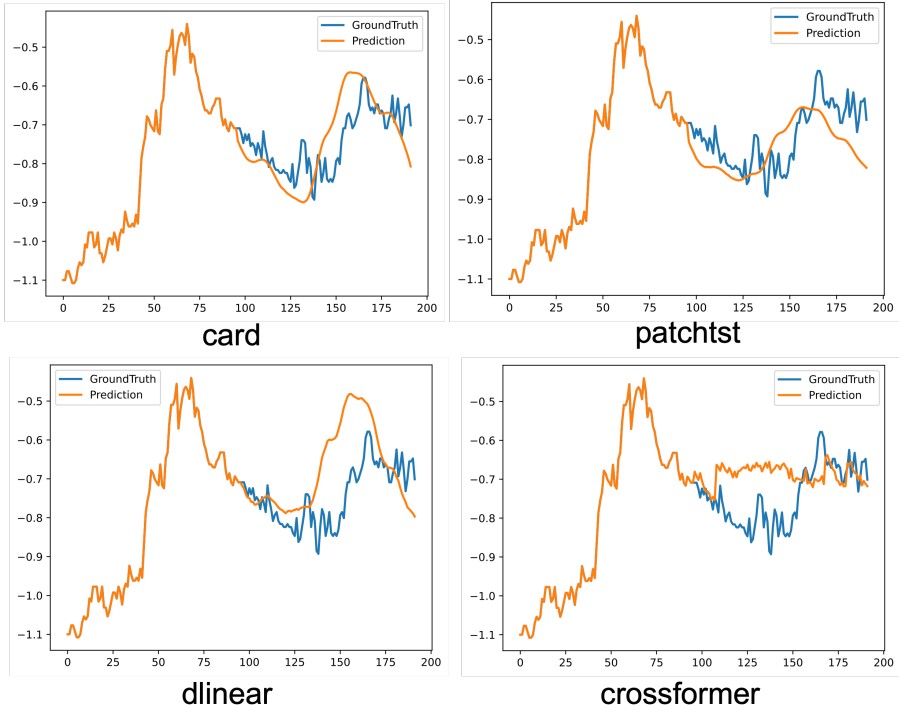

Figure 6: Sample prediction graph for ETTm1 long-term forecasting task

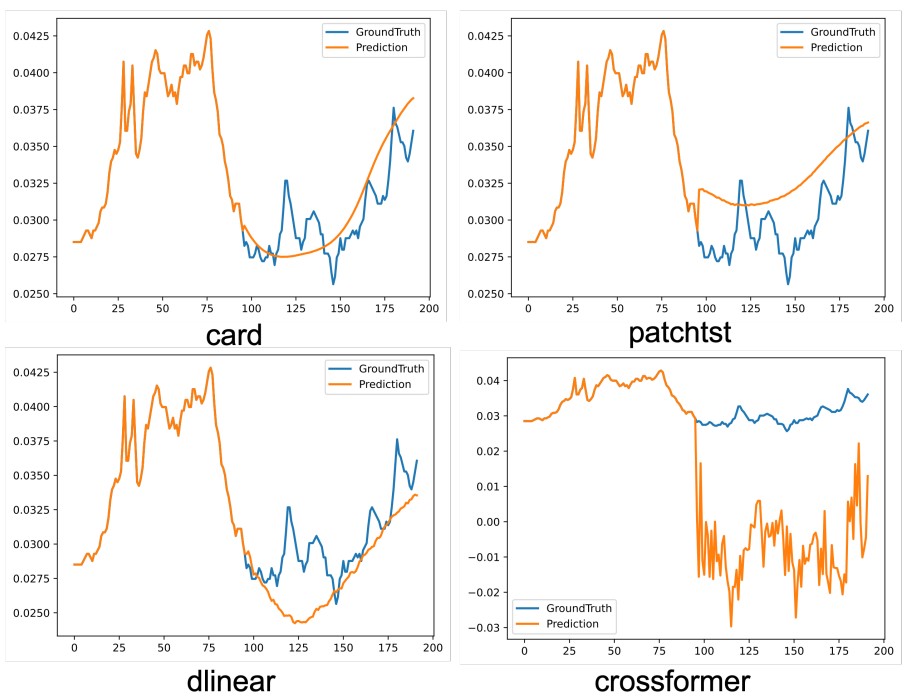

Figure 7: Sample prediction graph for Weather long-term forecasting task

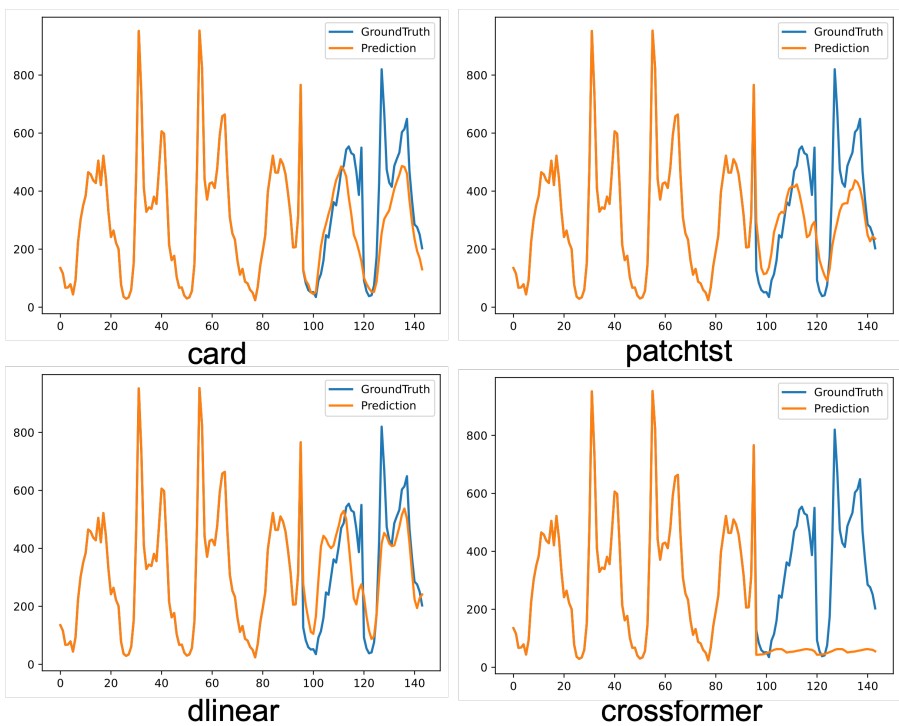

Figure 8: Sample prediction graph for M4 short-term forecasting task

## B  CARD'S ARCHITECTURE AND KEY COMPONENT'S SOURCE CODE

**Sample code of CARD**

```python
class CARD(nn.Module):
    def __init__(self, config, *args, **kwargs):

        super().__init__()
        self.patch_len = config.patch_len
        self.stride = config.stride
        self.d_model = config.d_model
        self.task_name = config.task_name
        patch_num = int((config.seq_len - self.patch_len)/self.stride + 1)
        self.patch_num = patch_num
        self.W_pos_embed = nn.Parameter(torch.randn(patch_num,config.d_model)*1e-2)
        self.total_token_number = self.patch_num + 1
        config.total_token_number = self.total_token_number

        # embeding layer related
        self.W_input_projection = nn.Linear(self.patch_len, config.d_model)
        self.input_dropout = nn.Dropout(config.dropout)
        self.cls = nn.Parameter(torch.randn(1,config.d_model)*1e-2)

        # mlp decoder
        self.W_out = nn.Linear((patch_num+1+self.model_token_number)*config.d_model, config.
            pred_len)

        # dual attention encoder related
        self.Attentions_over_token = nn.ModuleList([CARD_Attention(config) for i in range(
            config.e_layers)])
        self.Attentions_over_channel = nn.ModuleList([CARD_Attention(config,over_channel =
            True) for i in range(config.e_layers)])
        self.Attentions_mlp = nn.ModuleList([nn.Linear(config.d_model,config.d_model) for i in
             range(config.e_layers)])
        self.Attentions_dropout = nn.ModuleList([nn.Dropout(config.dropout) for i in range(
            config.e_layers)])
        self.Attentions_norm = nn.ModuleList([nn.Sequential(Transpose(1,2), nn.BatchNorm1d(
            config.d_model,momentum = config.momentum), Transpose(1,2)) for i in range(config.
            e_layers)])

    def forward(self, z, *args, **kwargs):

        b,c,s = z.shape

        # inputs nomralization
        z_mean = torch.mean(z,dim = (-1),keepdims = True)
        z_std = torch.std(z,dim = (-1),keepdims = True)
        z = (z - z_mean)/(z_std + 1e-4)

        # tokenization
        zcube = z.unfold(dimension=-1, size=self.patch_len, step=self.stride)
        z_embed = self.input_dropout(self.W_input_projection(zcube))+ self.W_pos_embed
        cls_token = self.cls.repeat(z_embed.shape[0],z_embed.shape[1],1,1)
        z_embed = torch.cat((cls_token,z_embed),dim = -2)

        # dual attention encoder
        inputs = z_embed
        b,c,t,h = inputs.shape
        for a_2,a_1,mlp,drop,norm in zip(self.Attentions_over_token, self.
            Attentions_over_channel,self.Attentions_mlp ,self.Attentions_dropout,self.
            Attentions_norm ):
            output_1 = a_1(inputs.permute(0,2,1,3)).permute(0,2,1,3)
            output_2 = a_2(output_1)
            outputs = drop(mlp(output_1+output_2))+inputs
            outputs = norm(outputs.reshape(b*c,t,-1)).reshape(b,c,t,-1)
            inputs = outputs

        # mlp decoder
        z_out = self.W_out(outputs.reshape(b,c,-1))
        # denomrlaization
        z = z_out *(z_std+1e-4) + z_mean
        return z
```

Sample code of CARD Smooth Attention

```python
class CARD_Attention(nn.Module):
    def __init__(self,config, over_channel = False, *args, **kwargs):
        super().__init__()
        self.over_channel = over_channel
        self.n_heads = config.n_heads
        self.merge_size = config.merge_size
        self.c_in = config.enc_in
        # attention related
        self.qkv = nn.Linear(config.d_model, config.d_model * 3, bias=True)
        self.attn_dropout = nn.Dropout(config.dropout)
        self.head_dim = config.d_model // config.n_heads
        self.dropout_mlp = nn.Dropout(config.dropout)
        self.mlp = nn.Linear( config.d_model, config.d_model)
        self.norm_post1 = nn.Sequential(Transpose(1,2), nn.BatchNorm1d(config.d_model,momentum
            = config.momentum), Transpose(1,2))
        self.norm_post2 = nn.Sequential(Transpose(1,2), nn.BatchNorm1d(config.d_model,momentum
            = config.momentum), Transpose(1,2))
        self.norm_attn = nn.Sequential(Transpose(1,2), nn.BatchNorm1d(config.d_model,momentum
            = config.momentum), Transpose(1,2))
        self.ff_1 = nn.Sequential(nn.Linear(config.d_model, config.d_ff, bias=True),nn.GELU(),
            nn.Dropout(config.dropout),
                        nn.Linear(config.d_ff, config.d_model, bias=True))
        self.ff_2= nn.Sequential(nn.Linear(config.d_model, config.d_ff, bias=True),nn.GELU(),
            nn.Dropout(config.dropout),
                        nn.Linear(config.d_ff, config.d_model, bias=True))
        # dynamic projection related
        self.dp_rank = config.dp_rank
        self.dp_k = nn.Linear(self.head_dim, self.dp_rank)
        self.dp_v = nn.Linear(self.head_dim, self.dp_rank)
        # EMA related
        ema_size = max(config.enc_in,config.total_token_number,config.dp_rank)
        ema_matrix = torch.zeros((ema_size,ema_size))
        alpha = config.alpha
        ema_matrix[0][0] = 1
        for i in range(1,config.total_token_number):
            for j in range(i):
                ema_matrix[i][j] = ema_matrix[i-1][j]*(1-alpha)
            ema_matrix[i][i] = alpha
        self.register_buffer('ema_matrix',ema_matrix)
    def ema(self,src):
        return torch.einsum('bnhad,ga_->bnhgd',src,self.ema_matrix[:src.shape[-2],:src.shape
            [-2]])
    def dynamic_projection(self,src,mlp):
        src_dp = mlp(src)
        src_dp = F.softmax(src_dp,dim = -1)
        src_dp = torch.einsum('bnhef,bnhec_->_bnhcf',src,src_dp)
        return src_dp
    def forward(self, src, *args,**kwargs):
        # construct Q,K,V
        B,nvars, H, C, = src.shape
        qkv = self.qkv(src).reshape(B,nvars, H, 3, self.n_heads, C // self.n_heads).permute(3,
            0, 1,4, 2, 5)
        q, k, v = qkv[0], qkv[1], qkv[2]
        if not self.over_channel:
            attn_score_along_token = torch.einsum('bnhed,bnhfd->bnhef', self.ema(q), self.ema(k
                ))/ self.head_dim ** -0.5
            attn_along_token = self.attn_dropout(F.softmax(attn_score_along_token, dim=-1) )
            output_along_token = torch.einsum('bnhef,bnhfd->bnhed', attn_along_token, v)
        else:
            # dynamic project V and K
            v_dp,k_dp = self.dynamic_projection(v,self.dp_v) , self.dynamic_projection(k,self.
                dp_k)
            attn_score_along_token = torch.einsum('bnhed,bnhfd->bnhef', self.ema(q), self.ema(
                k_dp))/ self.head_dim ** -0.5
            attn_along_token = self.attn_dropout(F.softmax(attn_score_along_token, dim=-1) )
            output_along_token = torch.einsum('bnhef,bnhfd->bnhed', attn_along_token, v_dp)
        # attention over hidden dimensions
        attn_score_along_hidden = torch.einsum('bnhae,bnhaf->bnhef', q,k)/ q.shape[-2] ** -0.5
        attn_along_hidden = self.attn_dropout(F.softmax(attn_score_along_hidden, dim=-1) )
        output_along_hidden = torch.einsum('bnhef,bnhaf->bnhae', attn_along_hidden, v)
        # token blend
        output1 = rearrange(output_along_token.reshape(B*nvars,-1,self.head_dim),
                        'bn_(hl1_hl2_hl3)_d_->_bn__hl2_(hl3_hl1)_d',
                        hl1 = self.n_heads//self.merge_size, hl2 = output_along_token.shape[-2]
                            ,hl3 = self.merge_size
                        ).reshape(B*nvars,-1,self.head_dim*self.n_heads)
        output2 = rearrange(output_along_hidden.reshape(B*nvars,-1,self.head_dim),
                        'bn_(hl1_hl2_hl3)_d_->_bn__hl2_(hl3_hl1)_d',
                        hl1 = self.n_heads//self.merge_size, hl2 = output_along_token.shape[-2]
                            ,hl3 = self.merge_size
                        ).reshape(B*nvars,-1,self.head_dim*self.n_heads)
        # post_norm
        output1 = self.norm_post1(output1).reshape(B,nvars, -1, self.n_heads * self.head_dim)
        output2 = self.norm_post2(output2).reshape(B,nvars, -1, self.n_heads * self.head_dim)
        # add & norm
        src2 = self.ff_1(output1)+self.ff_2(output2)
        src = src + src2
        src = src.reshape(B*nvars, -1, self.n_heads * self.head_dim)
        src = self.norm_attn(src)
        src = src.reshape(B,nvars, -1, self.n_heads * self.head_dim)
        return src
```

## C   DATASETS DETAILS FOR LONG TERM FORECASTING

**Datasets of Long-term Forecasting**   Table 5 summarizes details of statistics of long-term forecasting datasETSformer

Table 5: Dataset details in long-term forecasting.

| Dataset | Length | Dimension | Frequency |
|---|---|---|---|
| ETTm1 | 69680 | 7 | 15 min |
| ETTm2 | 69680 | 7 | 15 min |
| ETTh1 | 17420 | 7 | 1 hour |
| ETTh2 | 17420 | 7 | 1 hour |
| Weather | 52696 | 21 | 10 min |
| Electricity | 26304 | 321 | 1 hour |
| Traffic | 17544 | 862 | 1 hour |

## D   MODEL CONFIGURATIONS FOR LONG TERM FORECASTING

For all experiments, we use reversible instance normalization (RevIN, Kim et al., 2022) to handle data heterogeneity. As suggested in (Olivares et al., 2023) and (Salinas et al., 2020), other standardization methods are also useful when data enjoys certain patterns. We would like to defer the detailed analysis of them into future study. Moreover, the Adam optimizer (Kingma & Ba, 2017) with cosine learning rate decay after linear warm-up is used as training scheme. We train the proposed models with at most 8 NVIDIA Tesla V100 SXM2-16-GB GPUs. For all experiments, we fixed the number of encoder blocks, head dimensions and dynamic projection dimensions being 2, 8, and 8, respectively. The training epoch is set as 100. The default batch size is 128 and is adjusted due to GPU memory restriction. Other details of configurations are summarized in Table 6.

Table 6: Model configurations of CARD.

| Dataset | patch | stride | model dim | FFN dim | dropout | blend size | learning rate | warm-up | batch size |
|---|---|---|---|---|---|---|---|---|---|
| ETTm1 | 16 | 8 | 16 | 32 | 0.3 | 2 | 1e-4 | 0 | 128 |
| ETTm2 | 16 | 8 | 16 | 32 | 0.3 | 2 | 1e-4 | 0 | 128 |
| ETTh1 | 16 | 8 | 16 | 32 | 0.3 | 2 | 1e-4 | 0 | 128 |
| ETTh2 | 16 | 8 | 16 | 32 | 0.3 | 2 | 1e-4 | 0 | 128 |
| Weather | 16 | 8 | 128 | 256 | 0.2 | 16 | 1e-4 | 0 | 128 |
| Electricity | 16 | 8 | 128 | 256 | 0.2 | 16 | 1e-4 | 20 | 32 |
| Traffic | 16 | 8 | 128 | 256 | 0.2 | 16 | 1e-4 | 20 | 24 |

## E   EXTENDED NUMERICAL RESULTS OF CARD IN LONG-TERM FORECASTING WITH 96 INPUT LENGTH

We use the following recent popular models as baselines: FEDformer (Zhou et al., 2022b), ETS-former (Woo et al., 2022), FilM (Zhou et al., 2022a), LightTS (Zhang et al., 2022), MICN (Wang et al., 2023b), TimesNet (Wu et al., 2023b), Dlinear (Zeng et al., 2023), Crossformer (Zhang & Yan, 2023), and PatchTST (Nie et al., 2023). We use the experimental settings in (Wu et al., 2023b) applying reversible instance normalization (RevIN, Kim et al., 2022) to handle data heterogeneity and keeping the lookback length as 96 for fair comparisons. Each setting is repeated 10 times and average MSE/MAE results are reported.

In this section, we report the full results of long-term forecasting experiments in section 5.1. The MSE/MAE results are summarized in Table 7 and standard errors are reported in Table 8. CARD achieves 23/28 best performance in MSE and all the best results in MAE. It implies CARD can improve the baselines in a broad range of forecasting horizons. The standard deviation of CARD is on the order of 1e-3, which indicates our proposed framework is very robust. More baselines such as autoformer (Xu et al., 2021), nonstationary transformer (Liu et al., 2022b), Pyraformer (Liu et al., 2022a), LogTrans (Li et al., 2019b) and Informer (Zhou et al., 2021) can be found in Table 2 and Table 13 of (Wu et al., 2023b). CARD consistently outperforms those models in all forecasting horizons and we omit them for brevity.

Table 7: Long-term forecasting tasks. The lookback length is set as 96. All models are evaluated on 4 different prediction horizons {96, 192, 336, 720}. The best model is in boldface and the second best is underlined. For MICN, we report the better result between *MICN-regre* and *MICN-mean*.

| Models | | CARD | | PatchTST | | MICN | | TimesNet | | Crossformer | | Dlinear | | LightTS | | FiLM | | ETSformer | | FEDformer | |
|---|---|---|---|---|---|---|---|---|---|---|---|---|---|---|---|---|---|---|---|---|---|
| Metric | | MSE | MAE | MSE | MAE | MSE | MAE | MSE | MAE | MSE | MAE | MSE | MAE | MSE | MAE | MSE | MAE | MSE | MAE | MSE | MAE |
| ETTm1 | 96 | **0.316** | **0.347** | 0.342 | 0.378 | **0.316** | 0.364 | 0.338 | 0.375 | 0.366 | 0.400 | 0.345 | 0.372 | 0.374 | 0.400 | 0.348 | 0.367 | 0.375 | 0.398 | 0.764 | 0.416 |
| | 192 | **0.363** | **0.370** | 0.372 | 0.393 | **0.363** | 0.390 | 0.371 | 0.387 | 0.396 | 0.414 | 0.380 | 0.389 | 0.400 | 0.407 | 0.387 | 0.385 | 0.408 | 0.410 | 0.426 | 0.441 |
| | 336 | **0.392** | **0.390** | 0.402 | 0.413 | 0.408 | 0.426 | 0.410 | 0.411 | 0.439 | 0.443 | 0.413 | 0.413 | 0.438 | 0.438 | 0.418 | 0.405 | 0.435 | 0.428 | 0.445 | 0.459 |
| | 720 | **0.458** | **0.425** | 0.462 | 0.449 | 0.459 | 0.464 | 0.478 | 0.450 | 0.540 | 0.509 | 0.474 | 0453 | 0.527 | 0.502 | 0.479 | 0.440 | 0.499 | 0.462 | 0.543 | 0.490 |
| | avg | **0.383** | **0.384** | 0.395 | 0.408 | 0.387 | 0.411 | 0.400 | 0.406 | 0.435 | 0.417 | 0.403 | 0.407 | 0.435 | 0.437 | 0.408 | 0.399 | 0.429 | 0.425 | 0.448 | 0.452 |
| ETTm2 | 96 | **0.169** | **0.248** | 0.176 | 0.258 | 0.179 | 0.275 | 0.187 | 0.267 | 0.273 | 0.346 | 0.193 | 0.292 | 0.209 | 0.308 | 0.183 | 0.266 | 0.189 | 0.280 | 0.203 | 0.287 |
| | 192 | **0.234** | **0.292** | 0.244 | 0.304 | 0.262 | 0.326 | 0.249 | 0.309 | 0.350 | 0.421 | 0.284 | 0.362 | 0.311 | 0.382 | 0.247 | 0.305 | 0.253 | 0.319 | 0.269 | 0.328 |
| | 336 | **0.294** | **0.339** | 0.304 | 0.342 | 0.305 | 0.353 | 0.321 | 0.351 | 0.474 | 0.505 | 0.369 | 0.427 | 0.442 | 0.466 | 0.309 | 0.343 | 0.314 | 0.357 | 0.325 | 0.366 |
| | 720 | **0.390** | **0.388** | 0.408 | 0.403 | 0.389 | 0.407 | 0.497 | 0.403 | 1.347 | 0.812 | 0.554 | 0.522 | 0.675 | 0.587 | 0.407 | 0.398 | 0.414 | 0.413 | 0.421 | 0.415 |
| | avg | **0.272** | **0.317** | 0.283 | 0.327 | 0.284 | 0.340 | 0.291 | 0.333 | 0.609 | 0.521 | 0.350 | 0.401 | 0.409 | 0.436 | 0.287 | 0.328 | 0.292 | 0.342 | 0.305 | 0.349 |
| ETTh1 | 96 | 0.383 | 0.391 | 0.426 | 0.426 | 0.398 | 0.427 | 0.384 | 0.402 | 0.391 | 0.417 | 0.386 | 0.400 | 0.424 | 0.432 | 0.388 | 0.401 | 0.494 | 0.479 | **0.376** | 0.419 |
| | 192 | 0.435 | **0.420** | 0.469 | 0.452 | 0.430 | 0.453 | 0.436 | 0.429 | 0.449 | 0.452 | 0.437 | 0.432 | 0.475 | 0.462 | 0.443 | 0.439 | 0.538 | 0.504 | **0.420** | 0.448 |
| | 336 | 0.479 | **0.442** | 0.506 | 0.473 | **0.440** | 0.460 | 0.491 | 0.469 | 0.510 | 0.489 | 0.481 | 0.459 | 0.518 | 0.521 | 0.488 | 0.466 | 0.574 | 0.521 | 0.459 | 0.465 |
| | 720 | **0.471** | **0.461** | 0.504 | 0.495 | 0.491 | 0.509 | 0.521 | 0.500 | 0.594 | 0.567 | 0.519 | 0.516 | 0.547 | 0.533 | 0.525 | 0.519 | 0.562 | 0.535 | 0.506 | 0.507 |
| | avg | 0.442 | **0.429** | 0.455 | 0.444 | **0.440** | 0.462 | 0.458 | 0.450 | 0.486 | 0.481 | 0.456 | 0.452 | 0.491 | 0.479 | 0.461 | 0.456 | 0.452 | 0.510 | **0.440** | 0.460 |
| ETTh2 | 96 | **0.281** | **0.330** | 0.292 | 0.342 | 0.299 | 0.364 | 0.340 | 0.374 | 0.641 | 0.549 | 0.333 | 0.387 | 0.397 | 0.437 | 0.296 | 0.344 | 0.340 | 0.391 | 0.358 | 0.397 |
| | 192 | **0.363** | **0.381** | 0.387 | 0.400 | 0.422 | 0.441 | 0.402 | 0.414 | 0.896 | 0.656 | 0.477 | 0.476 | 0.520 | 0.504 | 0.389 | 0.402 | 0.430 | 0.439 | 0.429 | 0.439 |
| | 336 | **0.411** | **0.418** | 0.426 | 0.434 | 0.447 | 0.474 | 0.452 | 0.452 | 0.936 | 0.690 | 0.594 | 0.541 | 0.626 | 0.559 | 0.418 | 0.430 | 0.485 | 0.497 | 0.496 | 0.487 |
| | 720 | **0.416** | **0.431** | 0.430 | 0.446 | 0.442 | 0.467 | 0.462 | 0.468 | 1.390 | 0.863 | 0.831 | 0.657 | 0.863 | 0.672 | 0.433 | 0.448 | 0.500 | 0.497 | 0.463 | 0.474 |
| | avg | **0.368** | **0.390** | 0.384 | 0.406 | 0.402 | 0.437 | 0.414 | 0.427 | 0.966 | 0.690 | 0.559 | 0.515 | 0.602 | 0.543 | 0.384 | 0.406 | 0.439 | 0.452 | 0.437 | 0.449 |
| Weather | 96 | **0.150** | **0.188** | 0.176 | 0.218 | 0.161 | 0.229 | 0.172 | 0.220 | 0.164 | 0.232 | 0.196 | 0.255 | 0.182 | 0.242 | 0.193 | 0.234 | 0.237 | 0.312 | 0.217 | 0.296 |
| | 192 | **0.202** | **0.238** | 0.223 | 0.259 | 0.220 | 0.281 | 0.219 | 0.261 | 0.211 | 0.276 | 0.237 | 0.296 | 0.227 | 0.287 | 0.236 | 0.269 | 0.237 | 0.213 | 0.276 | 0.336 |
| | 336 | **0.260** | **0.282** | 0.277 | 0.297 | 0.278 | 0.331 | 0.280 | 0.306 | 0.269 | 0.327 | 0.283 | 0.335 | 0.282 | 0.334 | 0.288 | 0.304 | 0.298 | 0.353 | 0.339 | 0.380 |
| | 720 | 0.343 | 0.353 | 0.353 | 0.347 | **0.311** | 0.356 | 0.365 | 0.359 | 0.355 | 0.404 | 0.345 | 0.381 | 0.352 | 0.386 | 0.358 | 0.350 | 0.352 | 0.388 | 0.403 | 0.428 |
| | avg | **0.239** | **0.261** | 0.257 | 0.280 | 0.243 | 0.299 | 0.259 | 0.287 | 0.250 | 0.310 | 0.265 | 0.317 | 0.261 | 0.312 | 0.269 | 0.339 | 0.271 | 0.334 | 0.309 | 0.360 |
| Electricity | 96 | **0.141** | **0.233** | 0.190 | 0.296 | 0.164 | 0.269 | 0.168 | 0.272 | 0.254 | 0.347 | 0.197 | 0.282 | 0.207 | 0.307 | 0.198 | 0.276 | 0.187 | 0.304 | 0.193 | 0.308 |
| | 192 | **0.160** | **0.250** | 0.199 | 0.304 | 0.177 | 0.285 | 0.184 | 0.289 | 0.261 | 0.353 | 0.196 | 0.285 | 0.213 | 0.316 | 0.198 | 0.279 | 0.199 | 0.315 | 0.201 | 0.315 |
| | 336 | **0.173** | **0.263** | 0.217 | 0.319 | 0.193 | 0.304 | 0.198 | 0.300 | 0.273 | 0.364 | 0.209 | 0.301 | 0.230 | 0.333 | 0.217 | 0.301 | 0.212 | 0.329 | 0.214 | 0.329 |
| | 720 | **0.197** | **0.284** | 0.258 | 0.352 | 0.212 | 0.321 | 0.220 | 0.320 | 0.303 | 0.388 | 0.245 | 0.333 | 0.265 | 0.360 | 0.279 | 0.357 | 0.233 | 0.345 | 0.246 | 0.355 |
| | avg | **0.168** | **0.258** | 0.216 | 0.318 | 0.187 | 0.295 | 0.192 | 0.295 | 0.273 | 0.363 | 0.212 | 0.300 | 0.229 | 0.329 | 0.223 | 0.303 | 0.208 | 0.323 | 0.214 | 0.327 |
| Traffic | 96 | **0.419** | **0.269** | 0.462 | 0.315 | 0.519 | 0.309 | 0.593 | 0.321 | 0.558 | 0.320 | 0.650 | 0.396 | 0.615 | 0.391 | 0.649 | 0.391 | 0.607 | 0.392 | 0.587 | 0.366 |
| | 192 | **0.443** | **0.276** | 0.473 | 0.321 | 0.537 | 0.315 | 0.617 | 0.336 | 0.569 | 0.321 | 0.650 | 0.396 | 0.601 | 0.382 | 0.603 | 0.366 | 0.621 | 0.399 | 0.604 | 0.373 |
| | 336 | **0.460** | **0.283** | 0.494 | 0.331 | 0.534 | 0.313 | 0.629 | 0.336 | 0.591 | 0.328 | 0.605 | 0.373 | 0.613 | 0.386 | 0.613 | 0.371 | 0.622 | 0.396 | 0.621 | 0.383 |
| | 720 | **0.490** | **0.299** | 0.522 | 0.342 | 0.577 | 0.325 | 0.640 | 0.350 | 0.652 | 0.359 | 0.650 | 0.396 | 0.658 | 0.407 | 0.692 | 0.427 | 0.622 | 0.396 | 0.626 | 0.382 |
| | avg | **0.453** | **0.282** | 0.488 | 0.327 | 0.542 | 0.316 | 0.620 | 0.336 | 0.593 | 0.332 | 0.625 | 0.383 | 0.622 | 0.392 | 0.639 | 0.389 | 0.621 | 0.396 | 0.610 | 0.376 |

Table 8: Standard error results of CARD in long-term forecasting with 96 input length. Each setting is averaged over 10 random seeds.

| Tasks | ETTm1 | | ETTm2 | | ETTh1 | | ETTh2 | | Weather | | Electricity | | Traffic | |
|---|---|---|---|---|---|---|---|---|---|---|---|---|---|---|
| Metric | MSE | MAE | MSE | MAE | MSE | MAE | MSE | MAE | MSE | MAE | MSE | MAE | MSE | MAE |
| 96 | 1e-3 | 1e-3 | 1e-3 | 1e-3 | 1e-3 | 1e-3 | 2e-3 | 1e-3 | 1e-3 | 1e-3 | 2e-3 | 2e-3 | 2e-3 | 2e-3 |
| 192 | 1e-3 | 1e-3 | 2e-3 | 2e-3 | 1e-3 | 1e-3 | 2e-3 | 1e-3 | 3e-3 | 3e-3 | 3e-3 | 3e-3 | 2e-3 | 2e-3 |
| 336 | 1e-3 | 1e-3 | 3e-3 | 2e-3 | 2e-3 | 1e-3 | 2e-3 | 2e-3 | 4e-3 | 3e-3 | 3e-3 | 4e-3 | 4e-3 | 3e-3 |
| 720 | 2e-3 | 1e-3 | 5e-3 | 2e-3 | 5e-3 | 3e-3 | 4e-3 | 3e-3 | 6e-3 | 5e-3 | 5e-3 | 5e-3 | 6e-3 | 4e-3 |
| Avg | 1e-3 | 1e-3 | 3e-3 | 2e-3 | 2e-3 | 2e-3 | 3e-3 | 2e-3 | 4e-3 | 3e-3 | 4e-3 | 4e-3 | 3e-3 | 3e-3 |

# F   COMPARISON TO EARLY BASELINES

In this section, we report the comparison of CARD with early baselines including Nlinear, Linear, and Repret in Zeng et al. (2023). We use the experiment settings in subsection 5.1 and fix the input length as 96. The results are summarized in Table 9. Our model consistently outperforms those baselines.

# G   EXPERIMENTS ON ALL BENCHMARK DATASETS BY VARYING THE INPUT LENGTH TO ACHIEVE THE BEST RESULTS REPORTED IN BASELINE LITERATURE

We report the proposed model with 720 input length in Table 10. We follow the experimental settings used in (Nie et al., 2023). For each benchmark, we report the best results in the literature or conduct grid searches on input length to build strong baselines. In single-length experiments, CARD achieves

Table 9: Comparision to early baselines in long-term forecasting tasks. All models are evaluated on 4 different predication lengths {96, 192, 336, 720}. The best model is in boldface and the second best is underlined.

| Models | | ETTm1 | | ETTm2 | | ETTh1 | | ETTh2 | | Weather | | Electricity | | Traffic | |
|---|---|---|---|---|---|---|---|---|---|---|---|---|---|---|---|---|
| Metric | | MSE | MAE | MSE | MAE | MSE | MAE | MSE | MAE | MSE | MAE | MSE | MAE | MSE | MAE |
| CARD | 96 | 0.316 | 0.347 | 0.169 | 0.248 | 0.383 | 0.391 | 0.281 | 0.330 | 0.150 | 0.188 | 0.141 | 0.233 | 0.419 | 0.269 |
| | 192 | 0.363 | 0.370 | 0.234 | 0.292 | 0.435 | 0.420 | 0.363 | 0.381 | 0.202 | 0.238 | 0.160 | 0.259 | 0.443 | 0.276 |
| | 336 | 0.392 | 0.390 | 0.294 | 0.339 | 0.479 | 0.442 | 0.411 | 0.418 | 0.260 | 0.282 | 0.173 | 0.263 | 0.460 | 0.283 |
| | 720 | 0.458 | 0.425 | 0.390 | 0.388 | 0.471 | 0.461 | 0.416 | 0.431 | 0.343 | 0.353 | 0.197 | 0.284 | 0.490 | 0.299 |
| | avg | 0.383 | 0.384 | 0.272 | 0.317 | 0.442 | 0.429 | 0.368 | 0.390 | 0.239 | 0.353 | 0.168 | 0.258 | 0.453 | 0.282 |
| Nlinear | 96 | 0.368 | 0.385 | 0.187 | 0.271 | 0.556 | 0.494 | 0.326 | 0.373 | 0.203 | 0.242 | 0.216 | 0.300 | 0.663 | 0.404 |
| | 192 | 0.406 | 0.405 | 0.413 | 0.415 | 0.596 | 0.518 | 0.414 | 0.422 | 0.248 | 0.277 | 0.217 | 0.303 | 0.615 | 0.382 |
| | 336 | 0.439 | 0.426 | 0.312 | 0.348 | 0.621 | 0.531 | 0.453 | 0.453 | 0.300 | 0.314 | 0.231 | 0.318 | 0.623 | 0.384 |
| | 720 | 0.500 | 0.460 | 0.413 | 0.404 | 0.636 | 0.554 | 0.459 | 0.467 | 0.373 | 0.361 | 0.274 | 0.350 | 0.661 | 0.404 |
| | avg | 0.429 | 0.419 | 0.291 | 0.333 | 0.602 | 0.525 | 0.413 | 0.429 | 0.281 | 0.298 | 0.234 | 0.318 | 0.641 | 0.394 |
| Linear | 96 | 0.381 | 0.398 | 0.218 | 0.317 | 0.592 | 0.516 | 0.433 | 0.462 | 0.203 | 0.262 | 0.210 | 0.300 | 0.658 | 0.406 |
| | 192 | 0.413 | 0.415 | 0.305 | 0.379 | 0.602 | 0.529 | 0.570 | 0.534 | 0.242 | 0.299 | 0.209 | 0.301 | 0.607 | 0.380 |
| | 336 | 0.439 | 0.433 | 0.404 | 0.442 | 0.633 | 0.550 | 0.670 | 0.585 | 0.287 | 0.336 | 0.221 | 0.315 | 0.614 | 0.383 |
| | 720 | 0.496 | 0.467 | 0.569 | 0.532 | 0.673 | 0.595 | 0.922 | 0.700 | 0.350 | 0.385 | 0.256 | 0.346 | 0.655 | 0.404 |
| | avg | 0.432 | 0.428 | 0.374 | 0.417 | 0.625 | 0.547 | 0.649 | 0.570 | 0.271 | 0.316 | 0.224 | 0.316 | 0.633 | 0.393 |
| Repeat | 96 | 1.214 | 0.665 | 0.266 | 0.328 | 1.295 | 0.713 | 0.432 | 0.422 | 0.259 | 0.254 | 1.588 | 0.946 | 2.723 | 1.079 |
| | 192 | 1.261 | 0.690 | 0.340 | 0.371 | 1.325 | 0.733 | 0.534 | 0.473 | 0.309 | 0.292 | 1.595 | 0.950 | 2.756 | 1.087 |
| | 336 | 1.283 | 0.707 | 0.412 | 0.410 | 1.323 | 0.744 | 0.591 | 0.508 | 0.377 | 0.338 | 1.617 | 0.961 | 2.791 | 1.095 |
| | 720 | 1.319 | 0.729 | 0.521 | 0.465 | 1.339 | 0.756 | 0.588 | 0.517 | 0.465 | 0.394 | 1.647 | 0.975 | 2.811 | 1.097 |
| | avg | 1.269 | 0.698 | 0.385 | 0.394 | 1.321 | 0.737 | 0.536 | 0.480 | 0.353 | 0.320 | 1.612 | 0.958 | 2.770 | 1.090 |

the best results in **89%** cases in MSE metric and **86%** cases in MAE metric. In terms of average performance, CARD reaches the best results in all seven datasets.

Table 10: Long-term forecasting tasks. All models are evaluated on 4 different predication lengths {96, 192, 336, 720}. The best model is in boldface and the second best is underlined.

| Models | | CARD | | PatchTST | | MICN | | TimesNet | | Crossformer | | Dlinear | | LightTS | | FiLM | | ETSformer | | FEDformer | |
|---|---|---|---|---|---|---|---|---|---|---|---|---|---|---|---|---|---|---|---|---|---|
| Metric | | MSE | MAE | MSE | MAE | MSE | MAE | MSE | MAE | MSE | MAE | MSE | MAE | MSE | MAE | MSE | MAE | MSE | MAE | MSE | MAE |
| ETTm1 | 96 | **0.288** | **0.332** | 0.290 | 0.342 | 0.316 | 0.364 | 0.338 | 0.375 | 0.320 | 0.373 | 0.299 | 0.343 | 0.374 | 0.400 | 0.348 | 0.367 | 0.375 | 0.398 | 0.764 | 0.416 |
| | 192 | **0.332** | **0.357** | 0.332 | 0.369 | 0.363 | 0.390 | 0.371 | 0.387 | 0.372 | 0.411 | 0.355 | 0.365 | 0.400 | 0.407 | 0.387 | 0.385 | 0.408 | 0.410 | 0.426 | 0.441 |
| | 336 | **0.364** | **0.376** | 0.366 | 0.392 | 0.408 | 0.426 | 0.410 | 0.411 | 0.429 | 0.441 | 0.369 | 0.386 | 0.438 | 0.438 | 0.418 | 0.405 | 0.435 | 0.428 | 0.445 | 0.459 |
| | 720 | **0.414** | **0.407** | 0.416 | 0.420 | 0.459 | 0.464 | 0.478 | 0.450 | 0.573 | 0.531 | 0.425 | 0.421 | 0.527 | 0.502 | 0.479 | 0.440 | 0.499 | 0.462 | 0.543 | 0.490 |
| | avg | **0.350** | **0.368** | 0.351 | 0.381 | 0.387 | 0.411 | 0.400 | 0.406 | 0.424 | 0.439 | 0.362 | 0.379 | 0.435 | 0.437 | 0.408 | 0.399 | 0.429 | 0.425 | 0.448 | 0.452 |
| ETTm2 | 96 | **0.159** | **0.246** | 0.165 | 0.255 | 0.179 | 0.275 | 0.187 | 0.267 | 0.254 | 0.348 | 0.167 | 0.260 | 0.209 | 0.308 | 0.165 | 0.256 | 0.189 | 0.280 | 0.203 | 0.287 |
| | 192 | **0.214** | **0.285** | 0.220 | 0.292 | 0.262 | 0.326 | 0.249 | 0.309 | 0.370 | 0.433 | 0.224 | 0.303 | 0.311 | 0.382 | 0.222 | 0.296 | 0.253 | 0.319 | 0.269 | 0.328 |
| | 336 | **0.266** | **0.319** | 0.274 | 0.329 | 0.305 | 0.353 | 0.321 | 0.351 | 0.511 | 0.527 | 0.281 | 0.342 | 0.442 | 0.466 | 0.277 | 0.333 | 0.314 | 0.357 | 0.325 | 0.366 |
| | 720 | 0.379 | 0.390 | 0.362 | 0.385 | 0.389 | 0.407 | 0.497 | 0.403 | 0.901 | 0.689 | 0.397 | 0.421 | 0.675 | 0.587 | 0.371 | 0.398 | 0.414 | 0.413 | 0.421 | 0.415 |
| | avg | **0.254** | **0.310** | 0.255 | 0.315 | 0.284 | 0.340 | 0.291 | 0.333 | 0.509 | 0.522 | 0.256 | 0.331 | 0.409 | 0.436 | 0.259 | 0.321 | 0.292 | 0.342 | 0.305 | 0.349 |
| ETTh1 | 96 | **0.368** | **0.396** | 0.370 | 0.399 | 0.398 | 0.427 | 0.384 | 0.402 | 0.377 | 0.419 | 0.375 | 0.399 | 0.424 | 0.432 | 0.388 | 0.401 | 0.494 | 0.479 | 0.376 | 0.419 |
| | 192 | 0.406 | 0.418 | 0.413 | 0.421 | 0.430 | 0.453 | 0.436 | 0.429 | 0.410 | 0.439 | **0.405** | **0.416** | 0.475 | 0.462 | 0.443 | 0.439 | 0.538 | 0.504 | 0.420 | 0.448 |
| | 336 | **0.415** | **0.424** | 0.422 | 0.436 | 0.440 | 0.460 | 0.491 | 0.469 | 0.440 | 0.461 | 0.439 | 0.443 | 0.518 | 0.521 | 0.488 | 0.466 | 0.574 | 0.521 | 0.459 | 0.465 |
| | 720 | **0.416** | **0.448** | 0.447 | 0.466 | 0.491 | 0.509 | 0.521 | 0.500 | 0.519 | 0.524 | 0.472 | 0.490 | 0.547 | 0.533 | 0.525 | 0.519 | 0.562 | 0.535 | 0.506 | 0.507 |
| | avg | **0.401** | **0.421** | 0.413 | 0.431 | 0.440 | 0.462 | 0.458 | 0.450 | 0.437 | 0.461 | 0.423 | 0.437 | 0.491 | 0.479 | 0.461 | 0.456 | 0.452 | 0.510 | 0.440 | 0.460 |
| ETTh2 | 96 | **0.262** | **0.327** | 0.274 | 0.336 | 0.299 | 0.364 | 0.340 | 0.374 | 0.770 | 0.589 | 0.289 | 0.353 | 0.397 | 0.437 | 0.296 | 0.344 | 0.340 | 0.391 | 0.358 | 0.397 |
| | 192 | **0.322** | **0.369** | 0.339 | 0.379 | 0.422 | 0.441 | 0.402 | 0.414 | 0.848 | 0.657 | 0.383 | 0.418 | 0.520 | 0.504 | 0.389 | 0.402 | 0.430 | 0.439 | 0.429 | 0.439 |
| | 336 | **0.326** | **0.378** | 0.329 | 0.380 | 0.447 | 0.474 | 0.452 | 0.452 | 0.859 | 0.674 | 0.448 | 0.465 | 0.626 | 0.559 | 0.418 | 0.430 | 0.485 | 0.497 | 0.496 | 0.487 |
| | 720 | **0.373** | **0.419** | 0.379 | 0.422 | 0.442 | 0.467 | 0.462 | 0.468 | 1.221 | 0.825 | 0.605 | 0.551 | 0.863 | 0.672 | 0.433 | 0.448 | 0.500 | 0.497 | 0.463 | 0.474 |
| | avg | **0.321** | **0.373** | 0.330 | 0.379 | 0.402 | 0.437 | 0.414 | 0.427 | 0.454 | 0.446 | 0.259 | 0.321 | 0.602 | 0.543 | 0.384 | 0.406 | 0.439 | 0.452 | 0.437 | 0.449 |
| Weather | 96 | **0.145** | **0.186** | 0.149 | 0.198 | 0.161 | 0.229 | 0.172 | 0.220 | **0.145** | 0.211 | 0.152 | 0.237 | 0.182 | 0.242 | 0.193 | 0.234 | 0.237 | 0.312 | 0.217 | 0.296 |
| | 192 | **0.187** | **0.227** | 0.194 | 0.241 | 0.220 | 0.281 | 0.219 | 0.261 | 0.190 | 0.259 | 0.220 | 0.282 | 0.227 | 0.287 | 0.228 | 0.288 | 0.237 | 0.213 | 0.276 | 0.336 |
| | 336 | **0.238** | **0.258** | 0.245 | 0.282 | 0.278 | 0.331 | 0.280 | 0.306 | 0.259 | 0.326 | 0.265 | 0.319 | 0.282 | 0.334 | 0.267 | 0.323 | 0.298 | 0.353 | 0.339 | 0.380 |
| | 720 | **0.308** | **0.321** | 0.314 | 0.334 | 0.311 | 0.356 | 0.365 | 0.359 | 0.332 | 0.382 | 0.323 | 0.362 | 0.352 | 0.386 | 0.358 | 0.350 | 0.352 | 0.388 | 0.403 | 0.428 |
| | avg | **0.219** | **0.248** | 0.226 | 264 | 0.243 | 0.299 | 0.259 | 0.287 | 0.232 | 0.295 | 0.240 | 0.300 | 0.261 | 0.312 | 0.261 | 0.299 | 0.271 | 0.334 | 0.309 | 0.360 |
| Electricity | 96 | **0.129** | 0.223 | 0.129 | **0.222** | 0.164 | 0.269 | 0.168 | 0.272 | 0.186 | 0.281 | 0.153 | 0.237 | 0.207 | 0.307 | 0.152 | 0.267 | 0.187 | 0.304 | 0.193 | 0.308 |
| | 192 | 0.154 | 0.245 | **0.147** | **0.240** | 0.177 | 0.285 | 0.184 | 0.289 | 0.208 | 0.300 | 0.152 | 0.249 | 0.213 | 0.316 | 0.198 | 0.279 | 0.199 | 0.315 | 0.201 | 0.315 |
| | 336 | **0.161** | **0.257** | 0.163 | 0.259 | 0.193 | 0.304 | 0.198 | 0.300 | 0.323 | 0.369 | 0.169 | 0.267 | 0.230 | 0.333 | 0.188 | 0.283 | 0.212 | 0.329 | 0.214 | 0.329 |
| | 720 | **0.185** | **0.278** | 0.197 | 0.290 | 0.212 | 0.321 | 0.220 | 0.320 | 0.404 | 0.423 | 0.233 | 0.344 | 0.265 | 0.360 | 0.236 | 0.332 | 0.233 | 0.345 | 0.246 | 0.355 |
| | avg | **0.157** | **0.251** | 0.159 | 0.253 | 0.187 | 0.295 | 0.192 | 0.295 | 0.280 | 0.343 | 0.177 | 0.224 | 0.229 | 0.329 | 0.194 | 0.290 | 0.208 | 0.323 | 0.214 | 0.327 |
| Traffic | 96 | **0.341** | **0.229** | 0.360 | 0.249 | 0.519 | 0.309 | 0.593 | 0.321 | 0.511 | 0.292 | 0.410 | 0.282 | 0.615 | 0.391 | 0.416 | 0.294 | 0.607 | 0.392 | 0.587 | 0.366 |
| | 192 | **0.367** | **0.243** | 0.379 | 0.256 | 0.537 | 0.315 | 0.617 | 0.336 | 0.523 | 0.311 | 0.423 | 0.287 | 0.601 | 0.382 | 0.408 | 0.288 | 0.621 | 0.399 | 0.604 | 0.373 |
| | 336 | **0.388** | **0.254** | 0.392 | 0.264 | 0.534 | 0.313 | 0.629 | 0.336 | 0.530 | 0.300 | 0.436 | 0.296 | 0.613 | 0.386 | 0.425 | 0.298 | 0.622 | 0.396 | 0.621 | 0.383 |
| | 720 | **0.427** | **0.276** | 0.432 | 0.286 | 0.577 | 0.325 | 0.640 | 0.350 | 0.573 | 0.313 | 0.466 | 0.315 | 0.658 | 0.407 | 0.520 | 0.353 | 0.622 | 0.396 | 0.626 | 0.382 |
| | avg | **0.381** | **0.251** | 0.391 | 0.264 | 0.542 | 0.316 | 0.620 | 0.336 | 0.534 | 0.304 | 0.434 | 0.295 | 0.622 | 0.392 | 0.442 | 0.308 | 0.621 | 0.396 | 0.610 | 0.376 |

# H  M4 Short Term Forecasting

We also conduct experiments on short forecasting M4 tasks. M4 dataset (Makridakis et al., 2018) consists 100k time series. It covers time sequence data in various domains, including business, financial, and economy, and the sampling frequencies range from hourly to yearly. We follow the test setting suggested in (Wu et al., 2023b). Each experiment is repeated 10 times and average Symmetric Mean Absolute Percentage Error (SMAPE), Mean Absolute Scaled Error (MASE), and Overall Weighted Average (OWA) are reported. We benchmark our model with N-BEATS (Oreshkin et al., 2020), N-HiTS (Challu et al., 2022), Informer (Zhou et al., 2021), Autoformer (Wu et al., 2021) and 7 baselines in long-term forecasting. Details for datasets and training configurations can be found in Table 11 and Table 12 respectively.

The results are summarized in Table 13. Our proposed model consistently outperforms benchmarks in all tasks. Specifically, we outperform the state-of-the-art MLP-based method N-BEATS (Oreshkin et al., 2020) by **1.8%** in SMAPE reduction. We also outperform the best Transformer-based method PatchTST (Nie et al., 2023) and the best CNN-based method TimesNet (Wu et al., 2023b) by **1.5%** and **2.2%** in SMAPE reductions respectively. Since the M4 dataset only contains univariate time series, the attention to channels in our model plays a very limited role here. Thus good numerical performance indicates CARD's design with attention to hidden dimensions and token blend are also effective in univariate time series scenarios and can significantly boost forecasting performance.

The standard errors are reported in Table 14. Since the SAMPE score is not normalized, we observe the absolute value is on the order of 1e-2 while the MASE and OWA remain on the order of 1e-3 which is the same as in long-term forecasting experiments. After normalizing SAMPE with the corresponding mean value, the standard error of SMAPE will also reduce to the order of 1e-3.

Table 11: Datasets and mapping details of M4 dataset.

| Dataset | Length | Horizon |
|---|---|---|
| M4 Yearly | 23000 | 6 |
| M4 Quarterly | 24000 | 8 |
| M4 Monthly | 48000 | 18 |
| M4 Weekly | 359 | 13 |
| M4 Daily | 4227 | 14 |
| M4 Hourly | 414 | 48 |

Table 12: Model configurations for M4 experiment.

| Dataset | patch | stride | model dim | FFN dim | dropout | blend size | learning rate | warm-up | batch size |
|---|---|---|---|---|---|---|---|---|---|
| M4 Hourly | 16 | 1 | 128 | 512 | 0.1 | 2 | 5e-4 | 0 | 128 |
| M4 Weekly | 16 | 1 | 128 | 512 | 0.1 | 2 | 5e-4 | 0 | 128 |
| M4 Daily | 16 | 1 | 128 | 512 | 0.1 | 2 | 5e-4 | 0 | 128 |
| M4 Monthly | 16 | 1 | 128 | 512 | 0.1 | 2 | 5e-4 | 0 | 128 |
| M4 Quarterly | 4 | 1 | 128 | 512 | 0.1 | 2 | 5e-4 | 0 | 128 |
| M4 Yearly | 3 | 1 | 128 | 512 | 0.1 | 2 | 5e-4 | 0 | 128 |

Table 13: Short-term Forecasting tasks on M4 dataset. The average results of ten repeats are reported. The best model is in boldface and the second best is underlined.

| Models | | CARD | PatchTST | MICN | TimesNet | N-HiTS | N-BEATS | ETSformer | LightTS | Dlinear | FEDformer | Autoformer | Informer |
|---|---|---|---|---|---|---|---|---|---|---|---|---|---|
| Yearly | SMAPE | **13.215** | 13.258 | 14.935 | 13.387 | 13.418 | 13.436 | 18.009 | 14.247 | 16.965 | 13.728 | 13.974 | 14.727 |
| | MASE | **2.972** | 2.985 | 3.523 | 2.996 | 3.045 | 3.043 | 4.487 | 3.109 | 4.283 | 3.048 | 3.134 | 3.418 |
| | OWA | **0.778** | 0.781 | 0.900 | 0.786 | 0.793 | 0.794 | 1.115 | 0.827 | 1.058 | 0.803 | 0.822 | 0.881 |
| Quarterly | SMAPE | **9.958** | 10.179 | 11.452 | 10.100 | 10.202 | 10.124 | 13.376 | 11.364 | 12.145 | 10.792 | 11.338 | 11.360 |
| | MASE | **1.163** | 1.212 | 1.389 | 1.182 | 1.194 | 1.169 | 1.906 | 1.328 | 1.520 | 1.283 | 1.365 | 1.401 |
| | OWA | **0.876** | 0.904 | 1.026 | 0.890 | 0.899 | 0.886 | 1.302 | 1.000 | 1.106 | 0.958 | 1.012 | 1.027 |
| Monthly | SMAPE | **12.414** | 12.641 | 13.773 | 12.670 | 12.791 | 12.667 | 14.588 | 14.014 | 13.514 | 14.260 | 13.958 | 14.062 |
| | MASE | **0.907** | 0.930 | 1.076 | 0.933 | 0.969 | 0.937 | 1.368 | 1.053 | 1.037 | 1.102 | 1.103 | 1.141 |
| | OWA | **0.856** | 0.867 | 0.983 | 0.878 | 0.899 | 0.880 | 1.149 | 0.981 | 0.956 | 1.012 | 1.002 | 1.024 |
| Others | SMAPE | **4.522** | 4.851 | 6.716 | 4.891 | 5.061 | 4.925 | 7.267 | 15.880 | 6.709 | 4.954 | 5.458 | 24.460 |
| | MASE | **3.021** | 3.238 | 4.717 | 3.302 | 3.216 | 3.391 | 5.240 | 11.434 | 4.953 | 3.264 | 3.865 | 20.960 |
| | OWA | **0.962** | 1.021 | 1.451 | 1.035 | 1.040 | 1.053 | 1.591 | 3.474 | 1.487 | 1.036 | 1.187 | 5.879 |
| Avg | SMAPE | **11.614** | 11.807 | 13.130 | 11.829 | 11.927 | 11.851 | 14.718 | 13.252 | 13.639 | 12.840 | 12.909 | 14.086 |
| | MASE | **1.553** | 1.590 | 1.896 | 1.585 | 1.613 | 1.599 | 2.408 | 2.111 | 2.095 | 1.701 | 1.771 | 2.718 |
| | OWA | **0.832** | 0.834 | 0.980 | 0.851 | 0.861 | 0.855 | 1.172 | 1.051 | 1.051 | 0.918 | 0.939 | 1.230 |

Table 14: Standard error results of CARD in M4 short-term forecasting. The results normalized with the corresponding mean value are reported in parentheses. Each setting is averaged over 10 random seeds.

| Metric | Yearly | Quarterly | Monthly | Other | Average |
|---|---|---|---|---|---|
| SAMPE | 0.022 (0.001) | 0.008 (0.001) | 0.032 (0.002) | 0.024 (0.005) | 0.018 (0.002) |
| MASE | 0.007 (0.003) | 0.003 (0.002) | 0.003 (0.003) | 0.026 (0.008) | 0.003 (0.002) |
| OWA | 0.003 (0.002) | 0.001 (0.001) | 0.032 (0.037) | 0.004 (0.004) | 0.001 (0.001) |

## I  OTHER FORECASTING TASKS

In this section, we report the results of Illness and Exchange tasks. The Illness (CDC) and Exchange (Lai et al., 2018) contains the weekly data on influenza-like illness from Jan-2002 to Jun-2020 and the daily exchange rates of eight foreign countries including Australia, British, Canada, Switzerland, China, Japan, New Zealand, and Singapore ranging from 1990 to 2016 respectively. We follow the test setting suggested in (Wu et al., 2023b). Each experiment is repeated 10 times and MSE and MAE are reported. We benchmark our model with the baselines in long-term forecasting. Details for datasets and training configurations can be found in Table 15 and Table 16 respectively.

The results are summarized in Table 17. Our proposed model outperforms benchmarks in 4/8 cases in MSE and 6/8 cases in MAE. The standard errors are reported in Table 18.

Table 15: Datasets and mapping details of Illness and Exchange datasets.

| Dataset | Length | Horizon | Frequency |
|---|---|---|---|
| Illness | 966 | 7 | Weekly |
| Exchange | 7588 | 8 | Daily |

Table 16: Model configurations for Illness and Exchange tasks.

| Dataset | patch | stride | model dim | FFN dim | dropout | blend size | learning rate | warm-up | batch size | epochs |
|---|---|---|---|---|---|---|---|---|---|---|
| Illness | 36 | 1 | 16 | 32 | 0.3 | 2 | 2.5e-3 | 0 | 128 | 100 |
| Exchange | 16 | 8 | 16 | 32 | 0.3 | 2 | 1e-4 | 0 | 64 | 10 |

## J  EXTENDED RESULTS OF SIGNAL-BASED LOSS FUNCTION

The full results of experiments in section 5.3 are reported in Table 19 and Table 20. Moreover, we also conduct an experiment on switching to the decay function other than the two forms considered in section 4. The results are summarized in Table 21. in Table 21, we consider the following decay function: $f(t) = t^{-1/4}$, $f(t) = t^{-1/3}$, $f(t) = t^{-1}$, $f(t) = t^{-2}$, and $f(t) = t^{-3}$. In the ETTm1 task, we find that the decay function from $f(t) = t^{-1/4}$ and $f(t) = t^{-1/3}$ gives a similar MSE performance and slightly worse (by 0.001) MAE performance on average compared to the squared root decay. In the ETTh1 task, $f(t) = t^{-1/4}$, $f(t) = t^{-1/3}$, and $f(t) = t^{-1}$ work the same good as squared root decay. In practice, we believe the function that is not "decaying" faster than $f(t) = t^{-1}$ might be the candidate choice when no further information/assumptions on datasets could be obtained. For the slow decaying function (e.g., $f(t) = t^{-1/4}$ and $f(t) = t^{-1/3}$), vert slight performance improvement is observed in individual tasks when it is getting close to the squared root decay. It implies that the proposed loss is robustness for slow decaying function.

We also provide an illustration example to show the rationality of the proposed signal-based loss function. Let's consider a 1D autoregressive model $x_{t+1} = \beta^{\text{true}} x_t + \epsilon_t$ with $\epsilon_t \sim \mathcal{N}(0,1)$, $\beta^{\text{true}} \in (0,1)$ and $|x_t| \leq 1$. And we want to use $x_t$ to forecast $x_{t+1}$ and $x_{t+2}$. The plain loss function would be as follows:

$$\min_\beta \sum_{t=1}^{T} [\|x_t\beta - x_{t+1}\|_2^2 + \|x_t\beta^2 - x_{t+2}\|_2^2]. \tag{13}$$

Table 17: Exchange and Illness tasks. All models are evaluated on 4 different predication lengths $\{96, 192, 336, 720\}$. The best model is in boldface and the second best is underlined.

| Models | | CARD | | PatchTST | | MICN | | TimesNet | | Crossformer | | Dlinear | | LightTS | | FiLM | | ETSformer | | FEDformer | |
|---|---|---|---|---|---|---|---|---|---|---|---|---|---|---|---|---|---|---|---|---|---|
| Metric | | MSE | MAE | MSE | MAE | MSE | MAE | MSE | MAE | MSE | MAE | MSE | MAE | MSE | MAE | MSE | MAE | MSE | MAE | MSE | MAE |
| Exchange | 96 | **0.084** | **0.202** | 0.088 | 0.205 | 0.102 | 0.235 | 0.107 | 0.234 | 0.256 | 0.367 | 0.086 | 0.218 | 0.116 | 0.262 | 0.141 | 0.282 | 0.085 | 0.204 | 0.148 | 0.278 |
| | 192 | 0.179 | **0.298** | 0.176 | 0.299 | **0.172** | 0.316 | 0.226 | 0.344 | 0.469 | 0.509 | 0.176 | 0.315 | 0.215 | 0.359 | 0.241 | 0.364 | 0.348 | 0.428 | 0.271 | 0.380 |
| | 336 | 0.333 | 0.418 | 0.300 | **0.397** | **0.272** | 0.407 | 0.367 | 0.448 | 1.267 | 0.883 | 0.313 | 0.427 | 0.377 | 0.466 | 0.425 | 0.488 | 0.348 | 0.428 | 0.460 | 0.500 |
| | 720 | 0.851 | **0.691** | 0.901 | 0.713 | **0.714** | **0.658** | 0.964 | 0.746 | 1.767 | 1.068 | 0.839 | 0.695 | 0.831 | 0.699 | 0.993 | 0.747 | 1.025 | 0.774 | 1.195 | 0.841 |
| | avg | 0.362 | **0.402** | 0.366 | 0.404 | **0.315** | 0.404 | 0.416 | 0.443 | 0.940 | 0.707 | 0.354 | 0.414 | 0.385 | 0.447 | 0.450 | 0.473 | 0.410 | 0.427 | 0.519 | 0.500 |
| Illness | 96 | **2.043** | **0.863** | 2.234 | 0.891 | 3.457 | 1.288 | 2.317 | 0.934 | 3.461 | 1.237 | 2.398 | 1.040 | 8.313 | 2.144 | 3.589 | 1.420 | 2.527 | 1.020 | 3.228 | 1.260 |
| | 192 | 2.300 | **0.917** | 2.316 | 0.932 | 2.711 | 1.123 | **1.972** | 0.920 | 3.762 | 2.175 | 2.646 | 1.088 | 6.631 | 1.902 | 4.009 | 1.330 | 2.615 | 1.007 | 2.679 | 1.080 |
| | 336 | **1.899** | 0.846 | 2.153 | 0.900 | 2.775 | 1.145 | 2.359 | 0.972 | 3.853 | 1.307 | 2.614 | 1.086 | 7.299 | 1.982 | 3.785 | 1.492 | 2.359 | 0.972 | 2.622 | 1.078 |
| | 720 | **1.993** | **0.876** | 2.029 | 0.910 | 3.024 | 1.197 | 2.487 | 1.016 | 4.035 | 1.344 | 2.804 | 1.146 | 7.283 | 1.985 | 3.722 | 1.373 | 2.487 | 1.016 | 2.857 | 1.157 |
| | avg | **2.058** | **0.876** | 2.183 | 0.908 | 2.992 | 1.173 | 2.139 | 0.931 | 3.778 | 1.516 | 2.616 | 1.090 | 7.382 | 2.003 | 3.776 | 1.404 | 2.497 | 1.004 | 2.847 | 1.144 |

Table 18: Standard error results of CARD in Illness and Exchange tasks. Each setting is averaged over 10 random seeds.

| Tasks | Illness | | Exchange | |
|---|---|---|---|---|
| Metric | MSE | MAE | MSE | MAE |
| 96 | 0.172 | 0.029 | 4e-4 | 1e-3 |
| 192 | 0.173 | 0.026 | 7e-3 | 7e-3 |
| 336 | 0.059 | 0.019 | 1e-2 | 7e-3 |
| 720 | 0.072 | 0.018 | 1e-2 | 5e-3 |
| Avg | 0.119 | 0.023 | 7e-3 | 5e-3 |

Table 19: Influence for signal decay-based loss function. The lookback length is set as 96. All models are evaluated on 4 different predication lengths $\{96, 192, 336, 720\}$. The model name with * uses the robust loss proposed in this work. The better results are in boldface.

| Models | | CARD | | CARD* | | MICN-regre | | MICN-regre* | | TimesNet | | TimesNet* | | FEDformer | | FEDformer* | | Autoformer | | Autoformer* | |
|---|---|---|---|---|---|---|---|---|---|---|---|---|---|---|---|---|---|---|---|---|---|
| Metric | | MSE | MAE | MSE | MAE | MSE | MAE | MSE | MAE | MSE | MAE | MSE | MAE | MSE | MAE | MSE | MAE | MSE | MAE | MSE | MAE |
| ETTm1 | 96 | 0.329 | 0.364 | **0.316** | **0.347** | 0.316 | 0.362 | **0.313** | **0.350** | 0.338 | 0.375 | **0.321** | **0.356** | 0.379 | 0.419 | **0.344** | **0.380** | 0.505 | 0.475 | **0.450** | **0.442** |
| | 192 | 0.368 | 0.385 | **0.363** | **0.370** | 0.363 | 0.390 | **0.359** | **0.372** | **0.374** | 0.387 | 0.377 | **0.385** | 0.426 | 0.441 | **0.390** | **0.404** | 0.553 | 0.537 | **0.540** | **0.477** |
| | 336 | 0.400 | 0.405 | **0.393** | **0.390** | 0.408 | 0.426 | **0.392** | **0.399** | 0.410 | 0.411 | **0.401** | **0.400** | 0.445 | 0.459 | **0.436** | 0..433 | 0.621 | 0.537 | **0.594** | **0.505** |
| | 720 | 0.468 | 0.444 | **0.458** | **0.426** | 0.481 | 0.476 | **0.466** | **0.451** | 0.478 | 0.450 | **0.470** | **0.437** | 0.543 | 0.490 | **0.480** | **0.461** | 0.671 | 0.561 | **0.507** | **0.476** |
| | avg | 0.391 | 0.400 | **0.383** | **0.384** | 0.392 | 0.414 | **0.383** | **0.393** | 0.400 | 0.406 | **0.392** | **0.395** | 0.448 | 0.452 | **0.413** | **0.415** | 0.588 | 0.528 | **0.523** | **0.475** |
| ETTh1 | 96 | 0.387 | 0.399 | **0.383** | **0.391** | 0.421 | 0.431 | **0.403** | **0.412** | **0.384** | 0.402 | 0.389 | **0.400** | 0.376 | 0.419 | **0.371** | **0.400** | 0.449 | 0.459 | 0.453 | **0.445** |
| | 192 | 0.438 | 0.431 | **0.435** | **0.420** | 0.474 | 0.487 | **0.471** | **0.451** | **0.436** | 0.429 | **0.436** | **0.425** | 0.420 | 0.448 | **0.419** | **0.432** | **0.500** | **0.482** | 0.544 | 0.493 |
| | 336 | 0.486 | 0.454 | **0.479** | **0.461** | 0.569 | 0.551 | **0.513** | **0.496** | 0.491 | 0.469 | **0.475** | **0.450** | 0.459 | 0.465 | **0.461** | **0.455** | **0.521** | 0.496 | 0.535 | **0.491** |
| | 720 | 0.480 | 0.472 | **0.471** | **0.429** | 0.770 | 0.672 | **0.720** | **0.636** | 0.521 | 0.500 | **0.494** | **0.477** | 0.506 | 0.507 | **0.491** | **0.482** | **0.514** | **0.512** | 0.524 | 0.495 |
| | avg | 0.448 | 0.439 | **0.442** | **0.425** | 0.559 | 0.535 | **0.527** | **0.499** | 0.458 | 0.450 | **0.449** | **0.438** | 0.440 | 0.460 | **0.436** | **0.442** | 0.496 | 0.487 | 0.514 | **0.481** |

Table 20: Extended results on the signal decay-based loss function. The model name with * uses the robust loss proposed in this work. The better results are in boldface.

| Models | | Crossformer | | Crossformer* | | LightTS | | LightTS* | | FiLM | | FiLM* | | ETSformer | | ETSformer* | | Stationary | | Stationary* | |
|---|---|---|---|---|---|---|---|---|---|---|---|---|---|---|---|---|---|---|---|---|---|
| Metric | | MSE | MAE | MSE | MAE | MSE | MAE | MSE | MAE | MSE | MAE | MSE | MAE | MSE | MAE | MSE | MAE | MSE | MAE | MSE | MAE |
| ETTm1 | 96 | 0.366 | 0.400 | **0.353** | **0.364** | 0.374 | 0.400 | **0.332** | **0.360** | 0.348 | 0.367 | **0.335** | **0.359** | **0.375** | 0.398 | 0.382 | **0.391** | 0.386 | 0.398 | **0.345** | **0.364** |
| | 192 | 0.396 | 0.414 | **0.381** | **0.372** | 0.400 | 0.407 | **0.365** | **0.385** | 0.387 | 0.385 | **0.371** | **0.372** | 0.408 | 0.410 | **0.388** | **0.404** | 0.459 | 0.444 | **0.423** | **0.409** |
| | 336 | 0.439 | 0.443 | **0.431** | **0.415** | 0.438 | 0.438 | **0.414** | **0.408** | 0.418 | 0.405 | **0.399** | **0.413** | 0.435 | 0.428 | 0.442 | 0.431 | 0.495 | 0.464 | **0.430** | **0.415** |
| | 720 | 0.540 | 0.509 | **0.512** | **0.472** | 0.527 | 0.502 | **0.497** | **0.463** | 0.479 | 0.440 | **0.485** | **0.432** | 0.499 | 0.462 | **0.472** | **0.439** | 0.585 | 0.516 | **0.561** | **0.500** |
| | avg | 0.435 | 0.417 | **0.419** | **0.406** | 0.435 | 0.437 | **0.402** | **0.404** | 0.408 | 0.399 | **0.398** | **0.394** | 0.429 | 0.425 | **0.421** | **0.416** | 0.481 | 0.456 | **0.440** | **0.422** |
| ETTh1 | 96 | 0.391 | 0.417 | **0.388** | **0.397** | 0.424 | 0.432 | **0.412** | **0.418** | 0.388 | 0.401 | **0.387** | **0.389** | 0.494 | 0.479 | 0.499 | **0.457** | 0.513 | 0.419 | **0.509** | **0.394** |
| | 192 | 0.499 | 0.452 | **0.489** | **0.430** | 0.475 | 0.462 | **0.459** | **0.445** | 0.443 | 0.439 | 0.448 | **0.430** | 0.538 | 0.504 | **0.453** | **0.491** | 0.534 | 0.504 | **0.449** | **0.429** |
| | 336 | 0.510 | 0.489 | **0.493** | **0.472** | 0.518 | 0.521 | **0.499** | **0.502** | 0.488 | 0.466 | 0.490 | **0.458** | 0.574 | 0.521 | 0.589 | 0.527 | 0.588 | 0.535 | **0.560** | **0.501** |
| | 720 | 0.594 | 0.567 | **0.578** | **0.533** | 0.547 | 0.533 | **0.505** | **0.504** | 0.525 | 0.519 | 0.527 | **0.516** | 0.562 | 0.535 | **0.497** | **0.504** | 0.643 | 0.616 | 0.682 | 0.643 |
| | avg | 0.499 | 0.481 | **0.487** | **0.458** | 0.491 | 0.479 | **0.469** | **0.467** | 0.461 | 0.456 | 0.463 | **0.448** | 0.542 | 0.510 | **0.510** | **0.495** | 0.570 | 0.537 | **0.550** | **0.492** |

Via standard generalization analysis procedures and Hoeffding's Inequality, we have with probability $1 - \delta$,

$$|\beta - \beta^{true}| \leq \frac{\sqrt{\frac{5}{2} T \log(1/\delta)}}{\sum_{t=1}^{T} x_t^2}. \tag{14}$$

Table 21: Influences on the delay function to the loss function. The best results are in boldface and the second best is underlined.

| Function | $f(t) = 1$ | | $f(t) = t^{-0.25}$ | | $f(t) = t^{-0.33}$ | | $f(t) = t^{-0.5}$ | | $f(t) = t^{-1}$ | | $f(t) = t^{-2}$ | | $f(t) = t^{-3}$ | |
|---|---|---|---|---|---|---|---|---|---|---|---|---|---|---|
| Metric | MSE | MAE | MSE | MAE | MSE | MAE | MSE | MAE | MSE | MAE | MSE | MAE | MSE | MAE |
| ETTm1 96 | 0.329 | 0.364 | 0.319 | 0.349 | 0.318 | 0.349 | **0.316** | **0.347** | 0.317 | **0.347** | 0.334 | 0.356 | 0.345 | 0.363 |
| 192 | 0.368 | 0.385 | 0.362 | 0.370 | **0.361** | 0.370 | 0.363 | 0.370 | 0.363 | **0.369** | 0.379 | 0.377 | 0.430 | 0.416 |
| 336 | 0.400 | 0.405 | **0.393** | 0.391 | **0.393** | 0.390 | 0.393 | 0.390 | 0.396 | 0.391 | 0.414 | 0.402 | 0.605 | 0.505 |
| 720 | 0.468 | 0.444 | 0.459 | 0.427 | 0.459 | 0.427 | **0.458** | 0.426 | 0.466 | 0.429 | 0.491 | 0.449 | 0.760 | 0.578 |
| avg | 0.391 | 0.400 | **0.383** | 0.384 | **0.383** | 0.384 | **0.383** | 0.383 | 0.386 | 0.384 | 0.405 | 0.396 | 0.535 | 0.491 |
| ETTh1 96 | 0.387 | 0.399 | **0.382** | 0.391 | **0.382** | 0.390 | 0.382 | 0.390 | 0.383 | 0.391 | 0.387 | 0.396 | 0.410 | 0.413 |
| 192 | 0.438 | 0.431 | 0.437 | 0.421 | 0.436 | 0.420 | **0.435** | 0.420 | 0.436 | 0.421 | 0.439 | 0.426 | 0.559 | 0.494 |
| 336 | 0.486 | 0.454 | **0.478** | 0.443 | 0.478 | 0.442 | 0.478 | 0.442 | 0.479 | 0.443 | 0.485 | 0.453 | 0.712 | 0.566 |
| 720 | 0.480 | 0.472 | 0.472 | 0.462 | 0.472 | 0.462 | 0.471 | 0.462 | **0.470** | **0.460** | 0.551 | 0.508 | 0.786 | 0.613 |
| avg | 0.448 | 0.439 | **0.442** | 0.429 | **0.442** | 0.429 | **0.442** | 0.429 | **0.442** | **0.429** | 0.466 | 0.396 | 0.617 | 0.522 |

In this case, our proposed loss becomes:

$$\min_{\beta} \sum_{t=1}^{T} [\|x_t\beta - x_{t+1}\|_2^2 + \frac{1}{2}\|x_t\beta^2 - x_{t+2}\|_2^2]. \tag{15}$$

Follows the same analysis procedures, we have with probability $1 - \delta$

$$|\beta - \beta^{true}| \leq \frac{\sqrt{\frac{3}{2}T\log(1/\delta)}}{\sum_{t=1}^{T} x_t^2}. \tag{16}$$

Here the constant is improved from $\frac{5}{2}$ to $\frac{3}{2}$, which implies the new loss may yield better convergence upper bound.

## K  EXTENDED RESULTS OF ANOMALY DETECTION

The full results of the anomaly detection experiment in the section 5.2 are reported in Table 22. For each setting, we repeat 5 replicates.

Table 22: Full results for the anomaly detection task. The P, R and F1 represent the precision, recall and F1-score respectively.

| Datasets | SMD | | | MSL | | | SMAP | | | SWaT | | | PSM | | | Avg F1 |
|---|---|---|---|---|---|---|---|---|---|---|---|---|---|---|---|---|
| Metrics | P | R | F1 | P | R | F1 | P | R | F1 | P | R | F1 | P | R | F1 | |
| CARD | 0.883 | 0.861 | 0.872 | 0.896 | 0.750 | 0.817 | 92.93 | 0.794 | 0.857 | 0.928 | 0.962 | 0.945 | 0.982 | 0.933 | 0.957 | 0.890 |
| PatchTST | 0.802 | 0.942 | 0.866 | 0.898 | 0.760 | 0.823 | 89.97 | 0.566 | 0.695 | 0.919 | 0.899 | 0.909 | 0.992 | 0.913 | 0.951 | 0.849 |
| MICN | 0.765 | 0.838 | 0.780 | 0.892 | 0.752 | 0.816 | 0.895 | 0.518 | 0.656 | 0.913 | 0.841 | 0.875 | 0.987 | 0.885 | 0.933 | 0.816 |
| TimesNet | 0.887 | 0.831 | 0.858 | 0.839 | 0.864 | 0.852 | 0.925 | 0.583 | 0.715 | 0.883 | 0.962 | 0.921 | 0.982 | 0.968 | 0.975 | 0.864 |
| Crossformer | 0.722 | 0.844 | 0.778 | 0.907 | 0.749 | 0.820 | 0.895 | 0.541 | 0.674 | 0.919 | 0.856 | 0.886 | 0.971 | 0.876 | 0.921 | 0.816 |
| ETSformer | 0.874 | 0.792 | 0.831 | 0.851 | 0.849 | 0.850 | 0.923 | 0.558 | 0.695 | 0.900 | 0.804 | 0.849 | 0.99.3 | 0.853 | 0.918 | 0.829 |
| LightTS | 0.871 | 0.784 | 0.825 | 0.824 | 0.758 | 0.790 | 0.926 | 0.553 | 0.692 | 0.920 | 0.947 | 0.933 | 0.984 | 0.960 | 0.972 | 0.842 |
| DLinear | 0.836 | 0.715 | 0.771 | 0.843 | 0.854 | 0.849 | 0.923 | 0.554 | 0.693 | 0.809 | 0.953 | 0.875 | 0.983 | 0.893 | 0.936 | 0.825 |
| FEDformer | 0.880 | 0.824 | 0.851 | 0.771 | 0.801 | 0.786 | 0.905 | 0.581 | 0.708 | 0.902 | 0.964 | 0.932 | 0.973 | 0.972 | 0.972 | 0.850 |
| Stationary | 0.883 | 0.812 | 0.846 | 0.686 | 0.891 | 0.775 | 0.894 | 0.590 | 0.711 | 0.680 | 0.968 | 0.799 | 0.978 | 0.968 | 0.973 | 0.821 |
| Autoformer | 0.881 | 0.824 | 0.851 | 0.773 | 0.809 | 0.791 | 0.904 | 0.586 | 0.711 | 0.899 | 0.958 | 0.927 | 0.991 | 0.882 | 0.933 | 0.843 |
| Informer | 0.866 | 0.773 | 0.817 | 0.818 | 0.865 | 0.841 | 0.901 | 0.571 | 0.699 | 0.703 | 0.968 | 0.814 | 0.643 | 0.963 | 0.771 | 0.788 |

## L  ROBUSTNESS EXPERIMENTS

### L.1  INFLUENCE OF DIFFERENT INPUT LENGTHS

In this section, we report the robustness test when varying input length. We conduct experiments on ETTh1, ETTm1, Weather, and M4 datasets and repeat each setting with 10 random seeds. The robust experiment results are summarized in Figure 9-Figure 17. In general, we observe the longer input length may yield better performance, and the variance is also enlarged slightly.

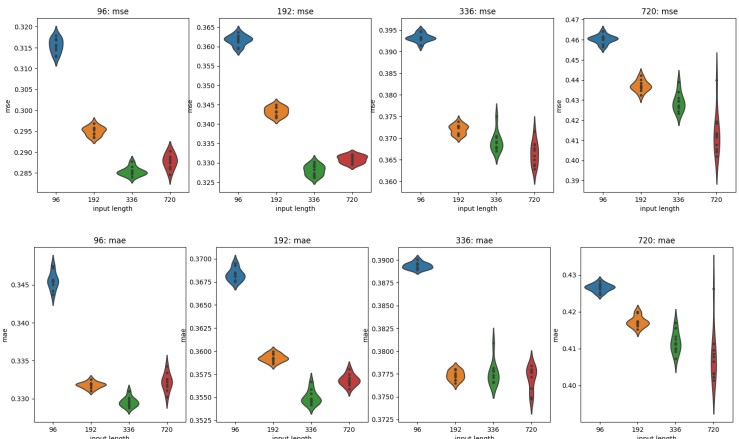

Figure 9: ETTm1 experiments with different input lengths.

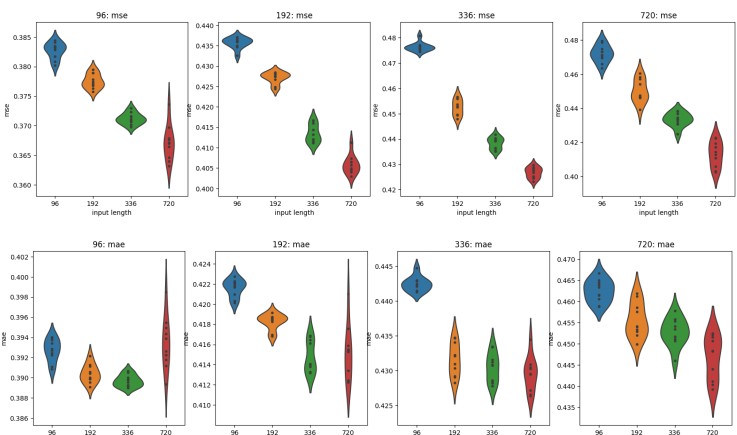

Figure 10: ETTm1 experiments with different input lengths.

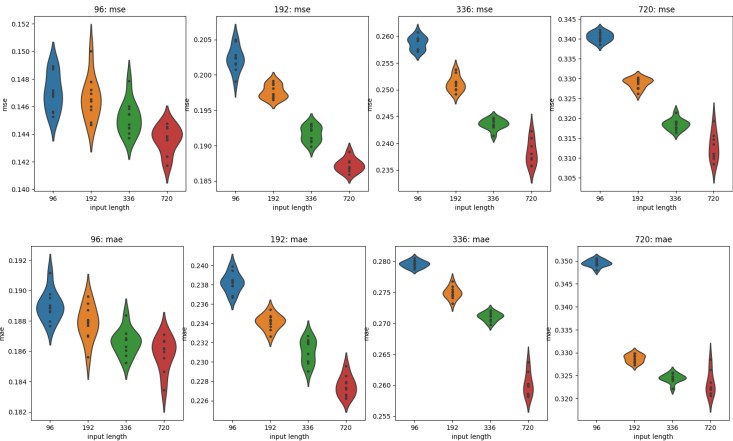

Figure 11: Weather experiments with different input lengths.

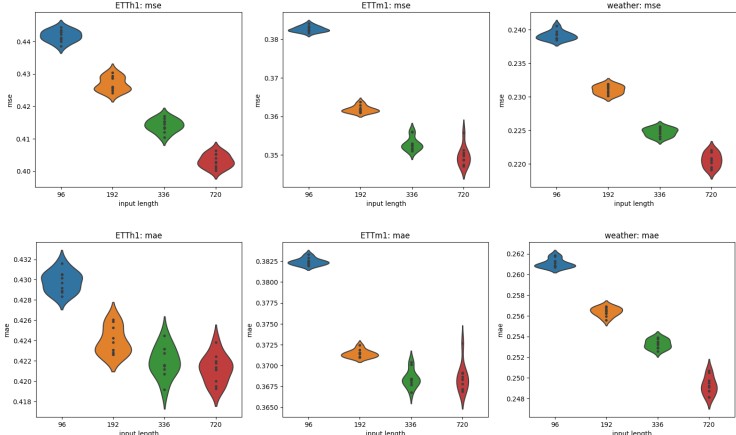

Figure 12: The average results of ETTm1, ETTh1 and Weather experiments with different input lengths.

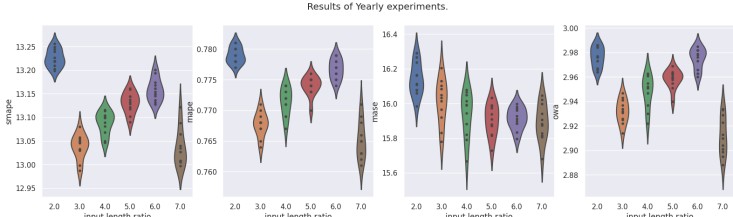

Figure 13: M4 Yearly experiments with different input lengths. The x axis "input length ratio" represents the ratio between input length and forecasting length.

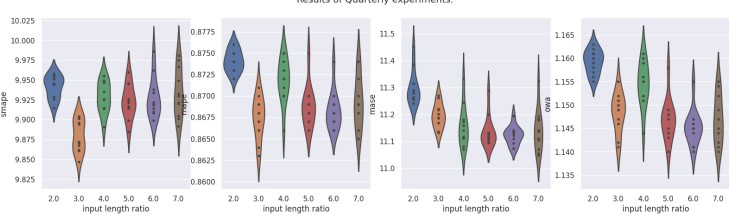

Figure 14: M4 Quarterly experiments with different input lengths. The x axis "input length ratio" represents the ratio between input length and forecasting length.

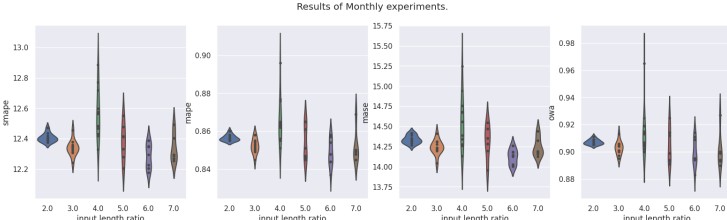

Figure 15: M4 Monthly experiments with different input lengths. The x axis "input length ratio" represents the ratio between input length and forecasting length.

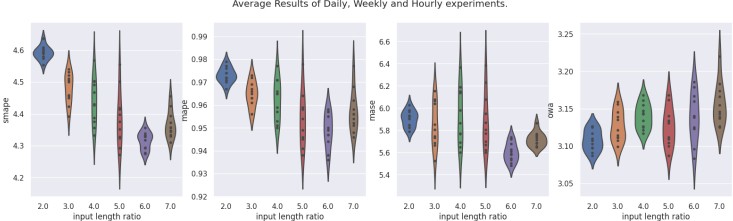

Figure 16: M4 average results of Daily, Weekly and Hourly experiments with different input lengths. The x axis "input length ratio" represents the ratio between input length and forecasting length.

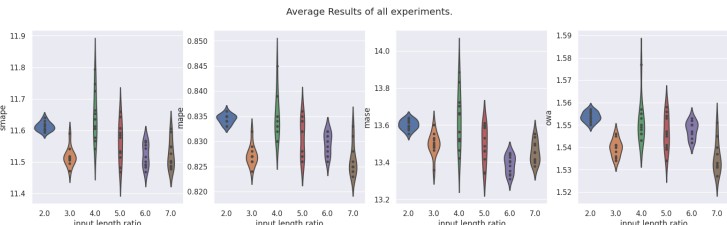

Figure 17: Average results of all M4 experiments with different input lengths. The x axis "input length ratio" represents the ratio between input length and forecasting length.

### L.1.1 INFLUENCE OF DIFFERENT MODEL SIZES.

In this section, we report the robustness test when varying model size. We conduct experiments on ETTh1, ETTm1, Weather, and M4 datasETSformer The model size (hidden dimension in attention) changes from 16 to 128 and the MLP layer dimension is set to be 2 times the model size. We repeat each setting with 10 random seeds. The robust experiment results are summarized in Figure 18-Figure 26. In terms of average performance (e.g., Figure 21 and Figure 26), the larger model size gives better results. For each individual task, we observe that the model with a large hidden dimension (e.g., 128) tends to overfit in low complexity tasks like ETTh1. For the high complexity task, the larger model size enables bigger learning capacity and gives better performance.

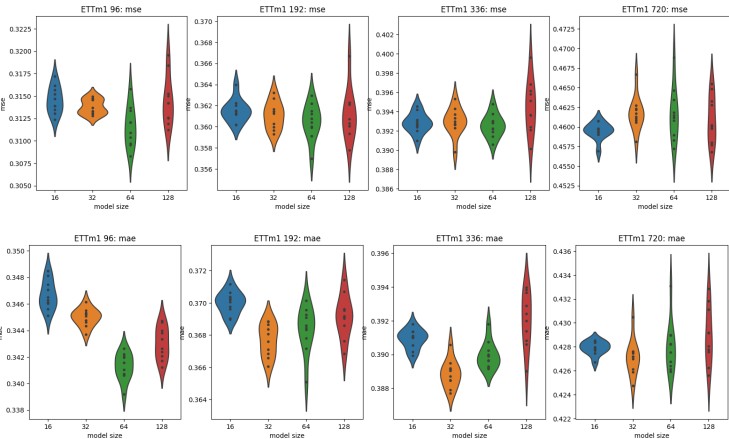

Figure 18: ETTm1 experiments with different model sizes.

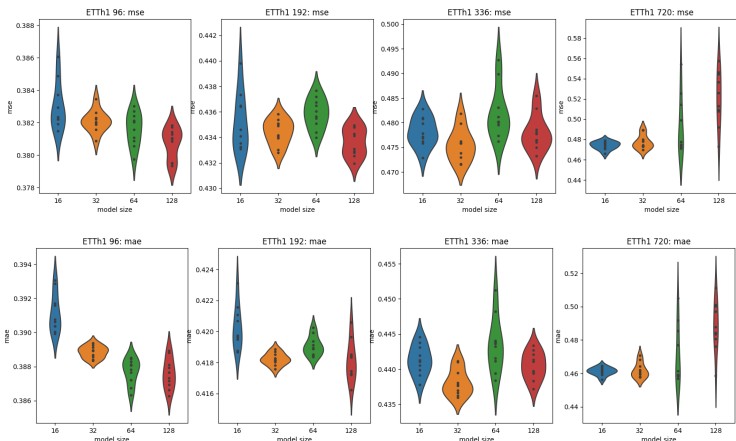

Figure 19: ETTh1 experiments with different model sizes.

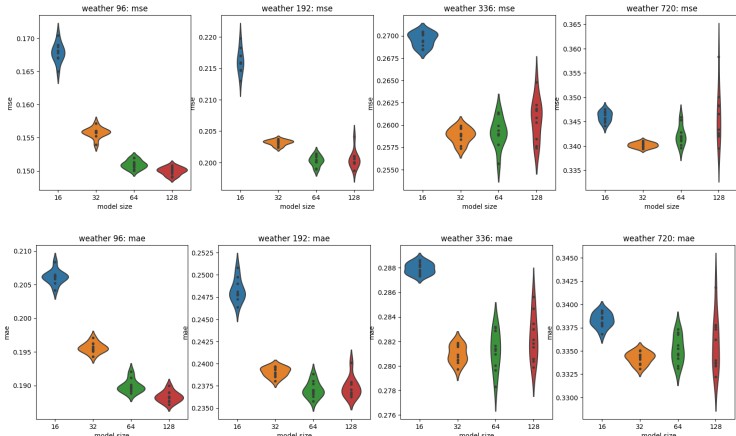

Figure 20: Weather experiments with different model sizes.

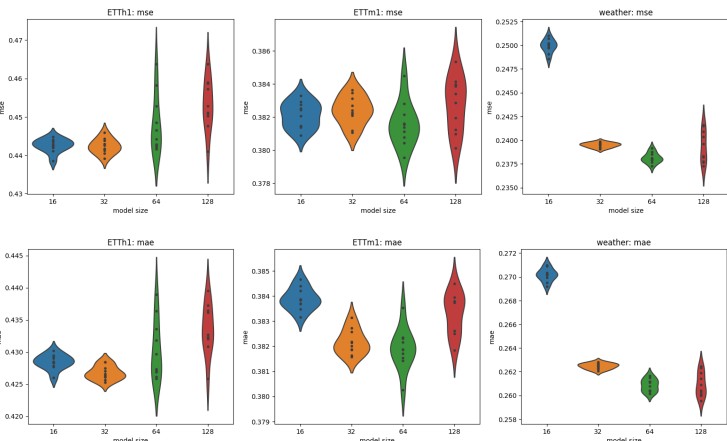

Figure 21: The average results of ETTm1, ETTh1 and Weather experiments with different model sizes.

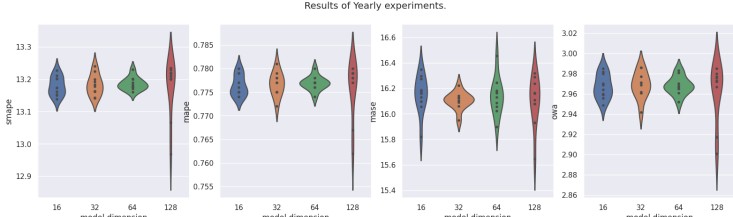

Figure 22: M4 Yearly experiments with different model sizes.

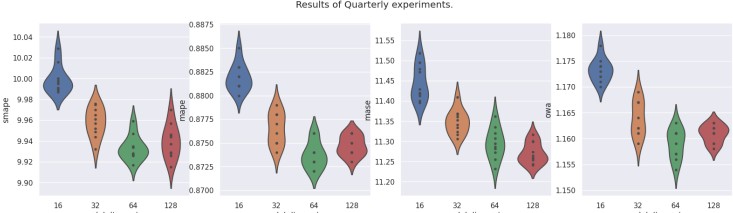

Figure 23: M4 Quarterly experiments with different model sizes.

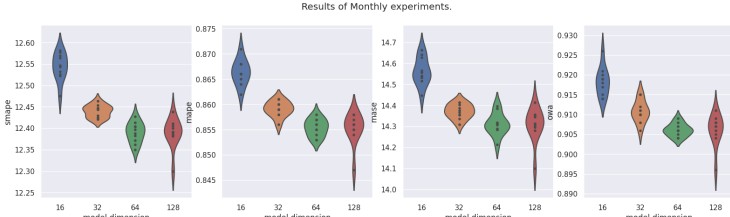

Figure 24: M4 Monthly experiments with different model size.

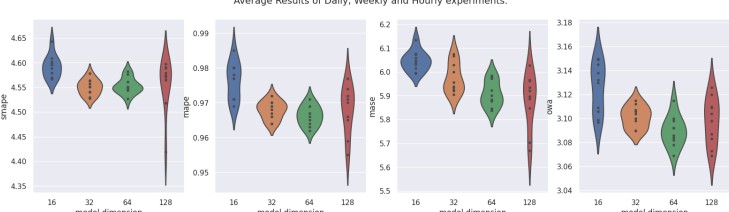

Figure 25: M4 average results of Daily, Weekly and Hourly experiments with different model sizes.

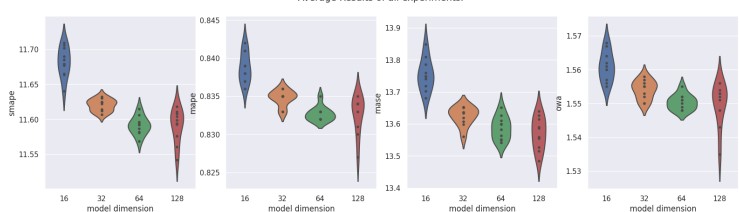

Figure 26: Average results of all M4 experiments with different model sizes.

## L.2 INFLUENCE OF DIFFERENT LEARNING RATES AND SCHEDULERS

In this section, we report the robustness test when varying learning rates and schedulers. We conduct experiments on ETTh1, ETTm1, Weather, and M4 datasets and repeat each setting with 10 random

seeds. We consider the fixed learning rate and cosine learning rate scheduler with the initial learning rate from 1e-3 to 1e-5. The results are summarized in Figure 27-Figure 35. We observe slight improvements when changing the fixed learning rate to the cosine learning rate decaying. For the relatively large learning rate, the variance of testing MSE/MAE increases and for the small enough learning rate, the model tends to underfit for the given training epochs. In practice, results suggest the learning should be set on the order of 1e-4.

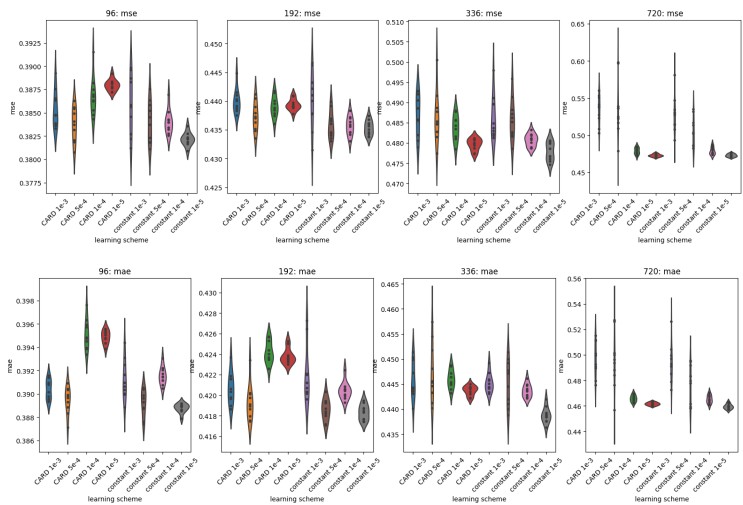

Figure 27: ETTh1 experiments with different learning rates and schedulers.

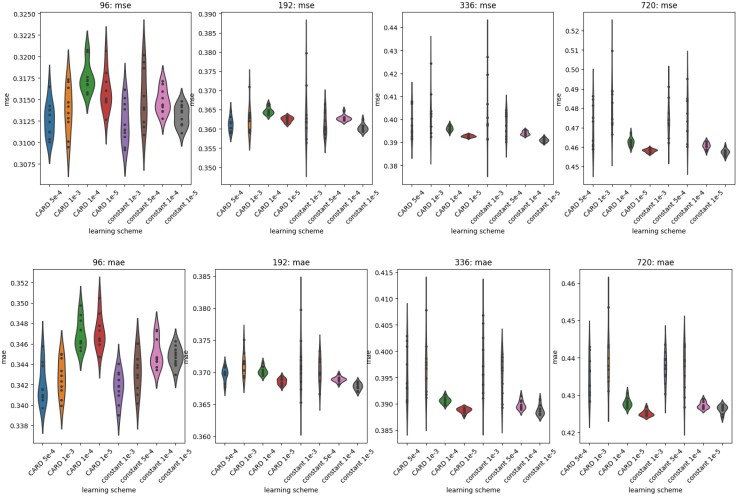

Figure 28: ETTm1 experiments with different learning rates and schedulers.

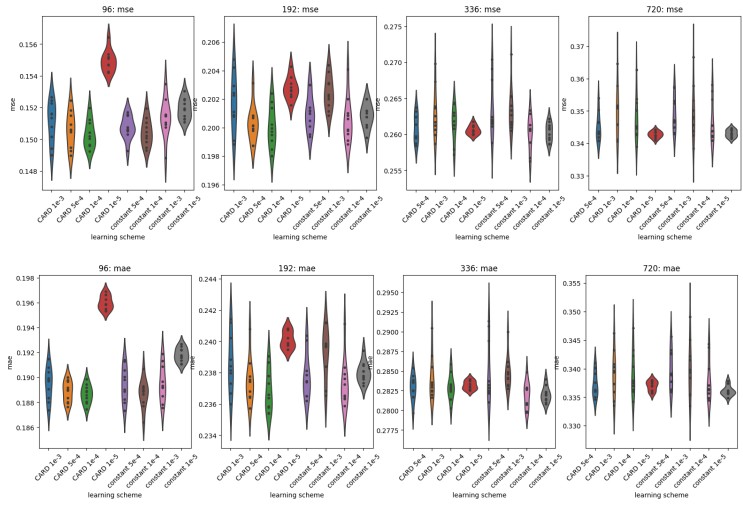

Figure 29: Weather experiments with different learning rates and schedulers.

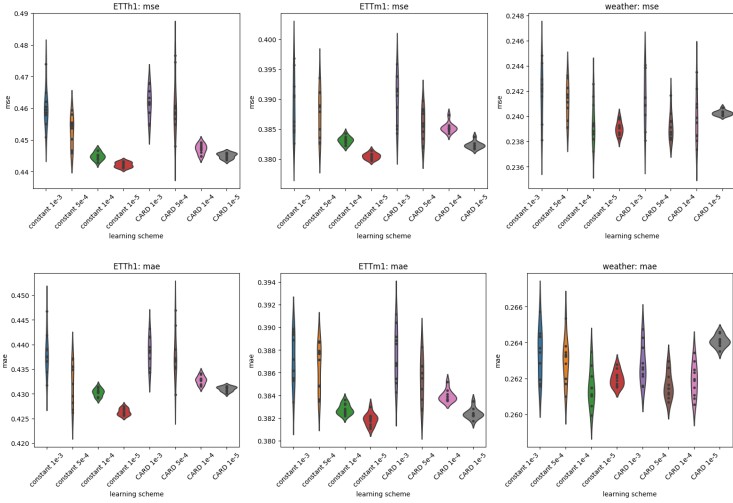

Figure 30: The average results of ETTm1, ETTh1 and Weather experiments with different learning rates and schedulers.

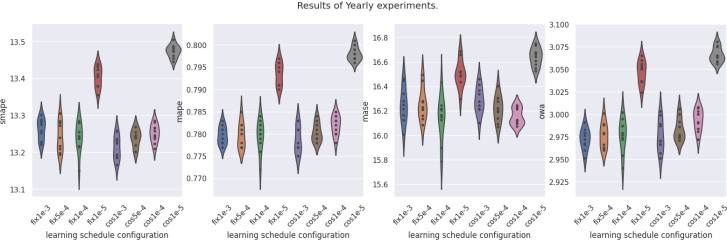

Figure 31: M4 Yearly experiments with different learning schemes.

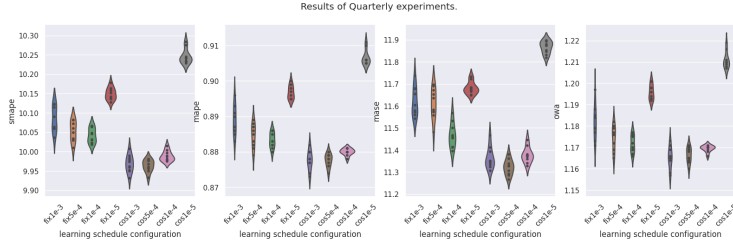

Figure 32: M4 Quarterly experiments with different learning schemes.

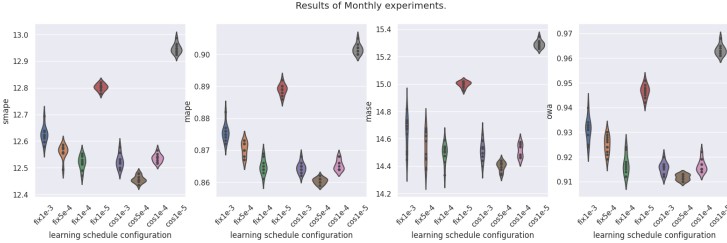

Figure 33: M4 Monthly experiments with different learning schemes.

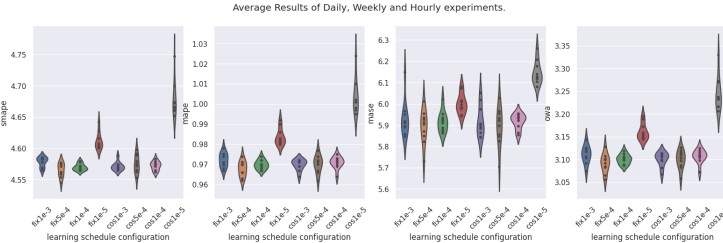

Figure 34: M4 average results of Daily, Weekly and Hourly experiments with different learning schemes.

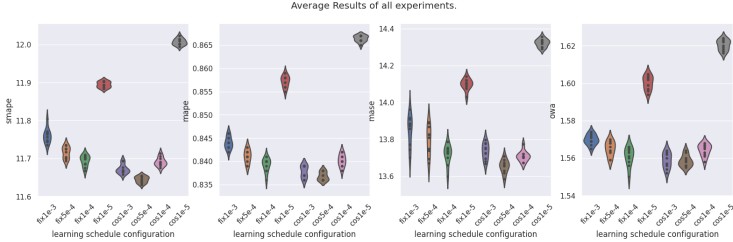

Figure 35: Average results of all M4 experiments with different learning schemes.

# M   TRAINING SPEED FOR DIFFERENT INPUT SEQUENCE LENGTH AND MODEL SIZE

In this section, we report the training speed differences when varying input sequence lengths and model sizes. For all experiments, we fix the batch size being 2 and use the average step time cost of 50 training epochs as the speed measure. We use the time cost of the experimental setting with 96 input length and 16 model dimensions as the base and report the ratio of the time increasing when using the longer input sequence and/or model size. The results are summarized in Table 23. Due to the patchified tokenization, when changing the input sequence length from 96 to 720 (7.5 times longer), the time increases less than $600\%$ and thus we don't observe the quadratic time differences. It implies our model can efficiently handle long input sequences.

Table 23: Model variants. All models are evaluated on 4 different predication lengths $\{96, 192, 336, 720\}$. The best results are in boldface.

| | model dimension | 16 | 32 | 64 | 128 |
|---|---|---|---|---|---|
| ETTh1 | 96 | 100.00% | 101.66% | 109.63% | 113.28% |
| | 192 | 108.28% | 104.49% | 1012.58% | 168.11% |
| | 336 | 113.39% | 109.12% | 159.18% | 275.57% |
| | 720 | 125.71% | 182.27% | 299.45% | 580.04% |
| Weather | 96 | 100.00% | 102.85% | 111.58% | 200.53% |
| | 192 | 104.90% | 113.98% | 169.15% | 312.11% |
| | 336 | 119.56% | 158.80% | 274.05% | 537.06% |
| | 720 | 115.42% | 167.38% | 298.41% | 604.03% |
| Electricity | 96 | 100.00% | 105.67% | 118.21% | 125.63% |
| | 192 | 102.13% | 108.96% | 128.34% | 140.41% |
| | 336 | 104.05% | 110.66% | 133.88% | 291.26% |
| | 720 | 111.04% | 112.03% | 193.18% | 481.24% |
| Traffic | 96 | 100.00% | 101.61% | 105.43% | 386.92% |
| | 192 | 103.02% | 113.02% | 117.51% | 454.12% |
| | 336 | 112.68% | 134.68% | 168.98% | 485.92% |
| | 720 | 128.17% | 183.90% | 307.04% | 561.57% |

## N  MODEL COMPLEXITIES AND RUNNING TIME

As we use the patching trick, the order of the total complexity would be the same as PatchTST and Crossformer, which is $\mathcal{O}(L^2/S^2)$. Some Transformer type models (e.g., FEDformer and Autoformer) may even break the quadratic dependent in $L$ and reach linear or nearly linear complexity in $L$. Other CNN and RNN type models (e.g., TimesNet and FilM) by nature maintain the $\mathcal{O}(L)$ complexity. When $S$ is set as a not-very-small number (e.g. $S \approx \mathcal{O}(\sqrt{L})$), our model's complexity can also be nearly linear. The condition $S \approx \mathcal{O}(\sqrt{L})$ is not very restrictive. Take $L = 900$ as an example, the length of $S = 30$ would be enough. For the case $L = 96$, the corresponding $S$ would be around 10.

The results of the experiments on running time are reported in Table 24. The input/forecasting lengths are set as 96/96 and we keep the batch size the same for all benchmarks and run the experiments on a single A100/80G GPU. Our proposed model yields comparable running time to transformer baselines as well as linear complexity baselines except Dlinear, which implies in practice model could also behave like a linear time model and won't introduce overhead computational cost.

Table 24: The average per step running time in seconds. The input/forecasting lengths are set as 96/96 and we keep the batch size the same for all benchmarks and run the experiments on a single A100/80G GPU. oom is short for out of memory.

| | | CARD | Autoformer | PatchTST | Crossformer | FEDformer | TimesNet | MICN | Dlinear | FilM | ETSFormer |
|---|---|---|---|---|---|---|---|---|---|---|---|
| ETTh1 | Train | 0.0197 | 0.1091 | 0.0164 | 0.2512 | 0.2107 | 0.0672 | 0.0423 | 0.0074 | 0.0747 | 0.0714 |
| Hidden=16, Batch=128 | Inference | 0.0046 | 0.0102 | 0.0021 | 0.0127 | 0.0178 | 0.0132 | 0.0030 | 0.0009 | 0.0123 | 0.0061 |
| Weather | Train | 0.0779 | 0.1525 | 0.0785 | 0.1186 | 0.2189 | 0.2457 | 0.0613 | 0.0330 | oom | 0.1354 |
| Hidden=128, Batch=128 | Inference | 0.0048 | 0.0139 | 0.0036 | 0.0123 | 0.0378 | 0.0224 | 0.0038 | 0.0014 | oom | 0.0092 |
| Electricity | Train | 0.2156 | 0.0835 | 0.3280 | oom | 0.1903 | oom | 0.0405 | 0.0163 | oom | 0.1174 |
| Hidden=128, Batch=32 | Inference | 0.0064 | 0.0160 | 0.0052 | oom | 0.0349 | oom | 0.0045 | 0.0021 | oom | 0.0103 |
| Traffic | Train | 0.2271 | 0.0960 | 0.1329 | oom | 0.1649 | 0.6048 | 0.0322 | 0.0139 | oom | 0.0607 |
| Hidden=128, Batch=12 | Inference | 0.0101 | 0.0153 | 0.0052 | oom | 0.0369 | 0.0418 | 0.0074 | 0.0058 | oom | 0.0223 |

## O  ATTENTION PATTERN MAPS

In this section, we report the attention maps of each head in the last attention layers. We use ETTh1 and ETTh2 tasks with forecasting length 96. The input length is set as 96 and we use patch length 8 with stride 8 to convert 96 time steps into 12 tokens, and we use the model with 2 attention heads. In order to highlight the correlation between the attention maps w.r.t. the forecasting sequences. We also report the dynamic time warping (DTW) scores between patches and the forecasting sequences, and the sum of attention scores for each patch. The DTW score can be treated as a rough ground truth to evaluate which input patches are most useful for forecasting. The results are summarized in Figure 36-Figure 37. We observe that attention maps have smooth landscapes and we believe it is due to the usage of EMA module to query and keyword tensors. Moreover, we find that the sum of attention scores for each patch is positively correlated with the post-hoc computed DTW scores between the patch and the forecasting sequence. It implies the proposed model can effectively

capture the relationship between the input sequences and forecasting sequences and lead to good final performance.

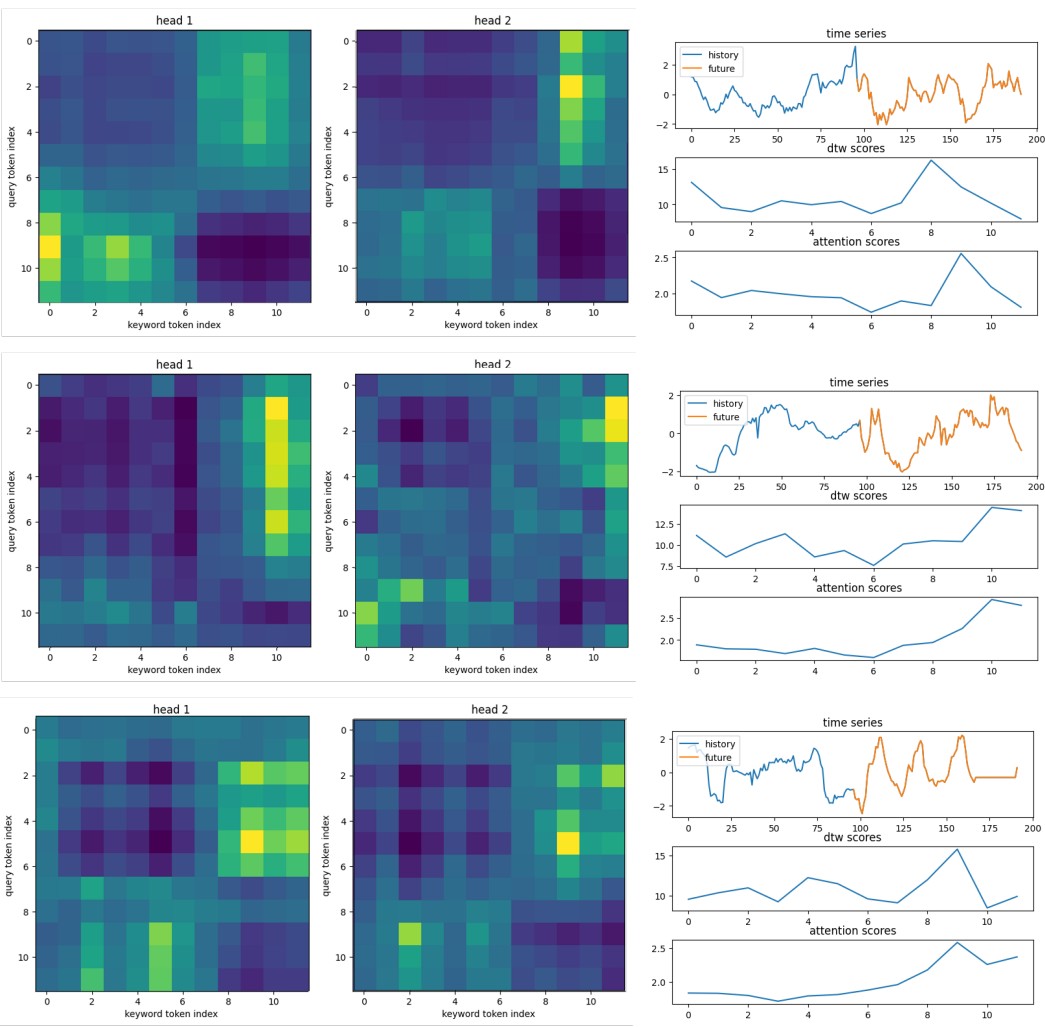

Figure 36: Attention Map Samples of ETTh1 task.

## P    RELATED WORKS

**Patched Transformers in other Domains** Transformer (Vaswani et al., 2017) has demonstrated significant potential in different data modalities. Among all applications, patching is an essential part when local semantic information is important. In NLP, BERT (Devlin et al., 2018), GPT (Radford et al., 2019) and their follow-up models consider subword-based tokenization and outperform character-based tokenization. In CV, Vision Transformers (e.g., Dosovitskiy et al. 2020; Liu et al. 2021; Bao et al. 2022; Ding et al. 2022; He et al. 2022) split an image into patches and then feed into the Transformer models. Similarly, in speech fields, researchers use convolutions to extract information in sub-sequence levels from a raw audio input (e.g., Hsu et al. 2021; Radford et al. 2022; Chen et al. 2022; Wang et al. 2023a).

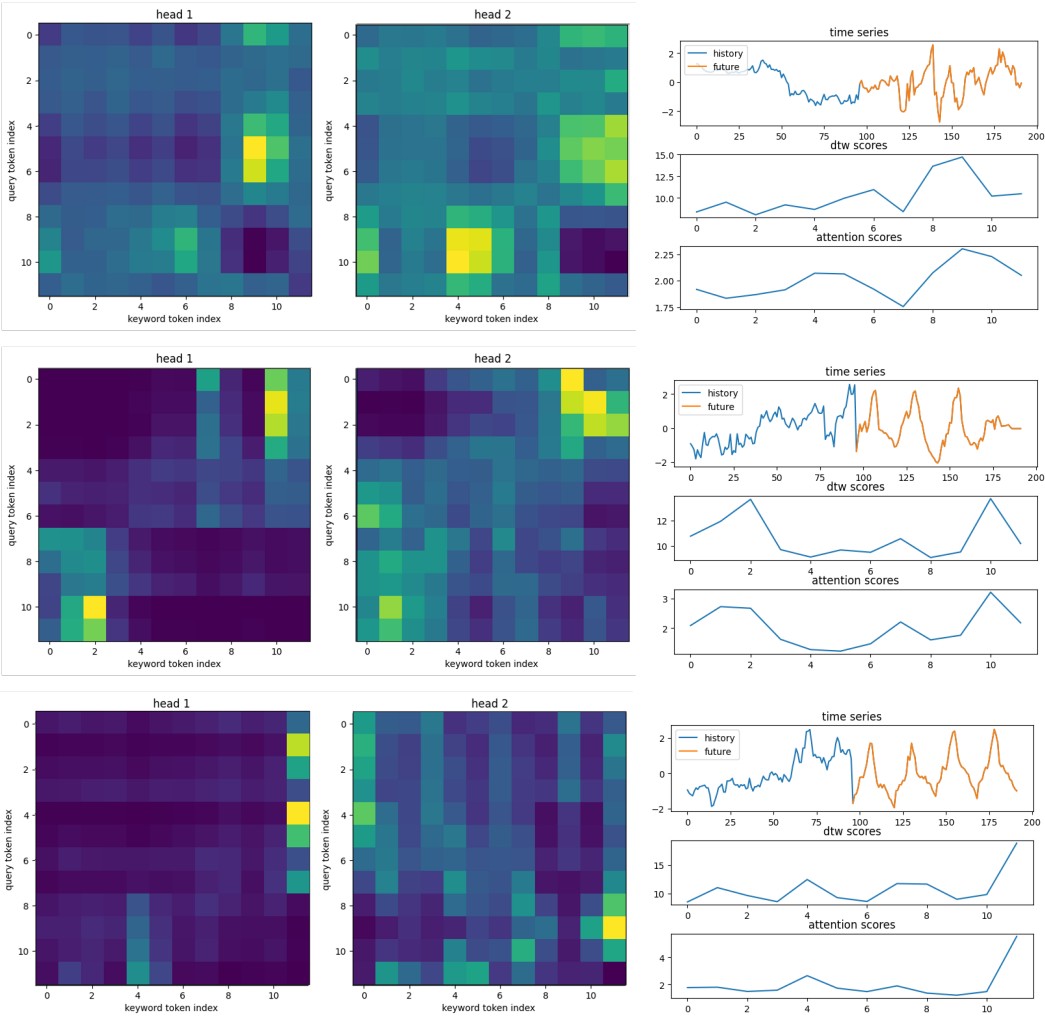

Figure 37: Attention Map Samples of ETTh2 task.

# Q   OTHERS

## Q.1   ARCHITECTURE VARIANTS

The present study encompasses the design of five distinct sequential and parallel feature flow architectures, with the aim of integrating both temporal signal and channel-aligned information. Here we consider 5 different settings as shown in Figure 38 and the results are summarized in Table 25. Following an exhaustive analysis, it is concluded that the architecture featuring the channel branch, complemented by channel/time blend, is the most resilient variant. Consequently, this specific architecture is adopted as the default approach in this work.

## Q.2   COMPONENT ABLATION EXPERIMENTS

**Ablation on attention over hidden dimensions and over channels.**   We conducted a series of ablation experiments by removing the attention over hidden dimensions and channels sequentially. Consistent with our design, as shown in table 26, the channel branch contributes to the reduction of mean squared error (MSE); its removal resulted in a 2%, 7% and 6%increase in MSE for ETTm1, Weather and Electricity respectively. The attention over hidden dimensions branch contributes approximately 1% to the reduction of MSE in ETTm1 and Weather tasks and in Electricity tasks dropping the over hidden dimensions attention branch results in 11% more MSE score on average.

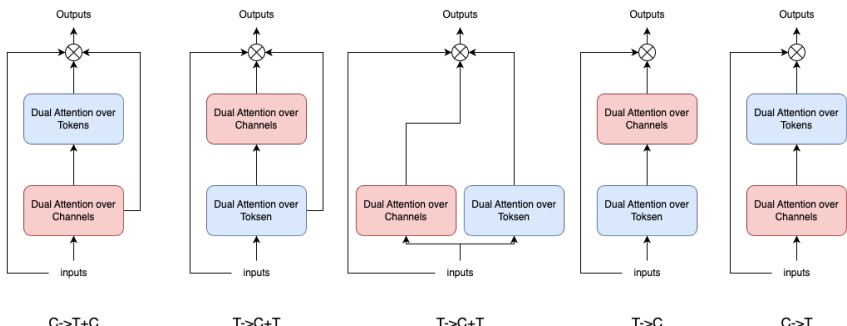

Figure 38: Architecture Variants

Table 25: Model variants. All models are evaluated on 4 different predication lengths $\{96, 192, 336, 720\}$. The best results are in boldface.

| Models | c->t+c (CARD) | | t->c+t | | t+c | | t->c | | c->t | |
|---|---|---|---|---|---|---|---|---|---|---|
| Metric | MSE | MAE | MSE | MAE | MSE | MAE | MSE | MAE | MSE | MAE |
| ETTm1 96 | **0.316** | **0.347** | 0.318 | 0.346 | 0.318 | 0.346 | 0.326 | 0.363 | 0.334 | 0.368 |
| ETTm1 192 | **0.363** | **0.370** | 0.367 | 0.370 | 0.366 | 0.369 | 0.366 | 0.385 | 0.372 | 0.387 |
| ETTm1 336 | **0.393** | **0.390** | 0.399 | 0.391 | 0.396 | 0.391 | 0.400 | 0.404 | 0.401 | 0.407 |
| ETTm1 720 | **0.458** | **0.426** | 0.466 | 0.429 | 0.463 | 0.428 | 0.459 | 0.440 | 0.458 | 0.438 |
| ETTm1 avg | **0.383** | **0.384** | 0.388 | 0.384 | 0.386 | 0.384 | 0.388 | 0.398 | 0.391 | 0.400 |
| Weather 96 | **0.150** | **0.188** | 0.153 | 0.193 | 0.152 | 0.189 | 0.152 | 0.191 | 0.152 | 0.192 |
| Weather 192 | 0.202 | 0.238 | 0.203 | 0.239 | **0.201** | **0.236** | **0.201** | 0.239 | 0.203 | 0.240 |
| Weather 336 | **0.260** | 0.282 | 0.269 | 0.288 | 0.261 | 0.281 | 0.263 | 0.284 | 0.262 | 0.284 |
| Weather 720 | **0.343** | **0.335** | 0.345 | 0.339 | 0.344 | 0.337 | 0.347 | 0.339 | 0.344 | 0.337 |
| Weather avg | **0.239** | 0.261 | 0.243 | 0.265 | 0.240 | **0.261** | 0.241 | 0.263 | 0.240 | 0.263 |

Table 26: Component Ablation Experiments by removing the attention over hidden dimensions (wo. hidden column) and removing the attention over channels (w.o channel) sequentially. All models are evaluated on 4 different predication lengths $\{96, 192, 336, 720\}$. The differences in thousandths w.r.t. predecessor models are reported in parentheses.

| Models | CARD | | wo. hidden | | | | wo. channel | | | |
|---|---|---|---|---|---|---|---|---|---|---|
| Metric | MSE | MAE | MSE | diff | MAE | diff | MSE | diff | MAE | diff |
| ETTm1 96 | 0.316 | 0.347 | 0.322 | (-6) | 0.345 | (2) | 0.326 | (-4) | 0.348 | (-3) |
| ETTm1 192 | 0.363 | 0.370 | 0.364 | (-1) | 0.370 | (0) | 0.372 | (-8) | 0.371 | (-1) |
| ETTm1 336 | 0.393 | 0.390 | 0.395 | (-2) | 0.391 | (-1) | 0.404 | (-9) | 0.393 | (-2) |
| ETTm1 720 | 0.458 | 0.426 | 0.462 | (-4) | 0.427 | (-1) | 0.470 | (-8) | 0.429 | (-2) |
| ETTm1 avg | 0.383 | 0.384 | 0.386 | (-3.3) | 0.383 | (0) | 0.393 | (-7.3) | 0.408 | (-2) |
| Weather 96 | 0.150 | 0.188 | 0.151 | (-1) | 0.191 | (-3) | 0.173 | (-22) | 0.205 | (-14) |
| Weather 192 | 0.202 | 0.238 | 0.201 | (1) | 0.236 | (2) | 0.220 | (-19) | 0.247 | (-11) |
| Weather 336 | 0.260 | 0.282 | 0.263 | (-3) | 0.282 | (0) | 0.275 | (-12) | 0.287 | (-5) |
| Weather 720 | 0.343 | 0.335 | 0.341 | (-8) | 0.336 | (-1) | 0.354 | (-14) | 0.339 | (-3) |
| Weather avg | 0.239 | 0.261 | 0.239 | (-2.5) | 0.261 | (-0.5) | 0.256 | (-16.8) | 0.270 | (-8.3) |
| Electricity 96 | 0.141 | 0.233 | 0.154 | (-14) | 0.242 | (-9) | 0.175 | (-21) | 0.250 | (-8) |
| Electricity 192 | 0.160 | 0.250 | 0.172 | (-12) | 0.257 | (-7) | 0.182 | (-10) | 0.259 | (-2) |
| Electricity 336 | 0.173 | 0.263 | 0.190 | (-17) | 0.274 | (-11) | 0.197 | (-7) | 0.275 | (-1) |
| Electricity 720 | 0.197 | 0.284 | 0.229 | (-32) | 0.312 | (-24) | 0.237 | (-8) | 0.318 | (-6) |
| Electricity avg | 0.168 | 0.258 | 0.186 | (-18.8) | 0.271 | (-12.8) | 0.198 | (-11.5) | 0.276 | (-4.2) |

**Ablation on projected dimension in dynamic projection.** We conduct experiments varying projected dimensions in 1,8,16. The results are summarized in Figure 39. We observe the performance of dynamic projection is not very sensitive to the projected dimensions, and only slight performance improvement is achieved when increasing the projected dimensions. As the goal of dynamic projection is to control the computation cost, the robustness in performance implies it doesn't harm the forecasting accuracy by a large margin. In practice, we may start with a relatively small projected dimension and adaptively tune the hyperparameter if necessary.

**Ablation on EMA** The goal of EMA is to improve the CARD's robustness. We conduct an experiment with different EMA parameters and the results are summarized in Figure 40. In this experiment, we consider three cases, heavy smoothing (e.g., 0.1), medium smoothing (e.g., 0.5), and light smoothing (e.g., 0.9). We observe that, in the majority of cases, the standard deviations of MAE and MSE decrease when the strength of smoothing is increased. It confirms our hypothesis that the EMA can improve model robustness. Moreover, in our experiment, we also find that changing

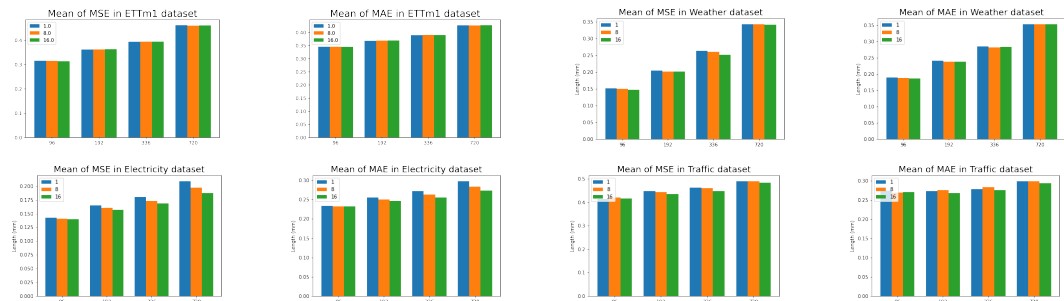

Figure 39: Experiments on dynamic projection dimensions. The projection dimension is varying in 1, 8, and 16.

the parameter of EMA from being fixed to learnable makes very few performance differences but significantly increases the training time. In practice, we would suggest using a fixed EMA parameter.

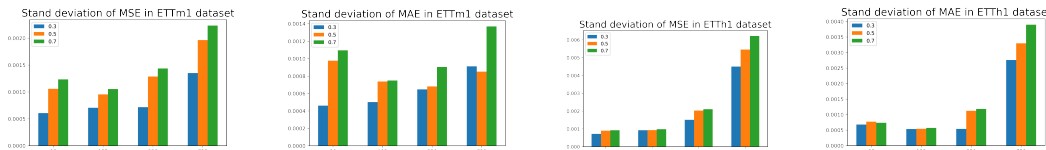

Figure 40: Experiments on stability of EMA module. Each setting is averaged over 10 random seeds.

# R MORE ANALYSIS ON TRANSFORMERS FOR TIME SERIES FORECASTING

## R.1 EXTENDED RESULTS ON INFLUENCE OF INPUT SEQUENCE LENGTH

The extended results on varying input sequence lengths are shown in Table 27.

Table 27: Influence of prolonging input sequence. The lookback length is set as 96,192,336,720: CARD(96) means using lookback length 96.

| Models | CARD(96) | | CARD(192) | | CARD(336) | | CARD(720) | |
|---|---|---|---|---|---|---|---|---|
| Metric | MSE | MAE | MSE | MAE | MSE | MAE | MSE | MAE |
| ETTh1 96 | 0.383 | 0.391 | 0.378 | **0.390** | 0.372 | **0.390** | **0.368** | 0.392 |
| ETTh1 192 | 0.435 | 0.420 | 0.427 | 0.418 | 0.413 | **0.416** | **0.407** | **0.416** |
| ETTh1 336 | 0.479 | 0.442 | 0.458 | 0.434 | 0.437 | 0.431 | **0.428** | **0.430** |
| ETTh1 720 | 0.471 | 0.461 | 0.452 | 0.456 | 0.436 | 0.453 | **0.418** | **0.449** |
| ETTh1 avg | 0.442 | 0.429 | 0.429 | 0.425 | 0.415 | 0.422 | **0.405** | **0.421** |
| ETTm1 96 | 0.316 | 0.347 | 0.296 | 0.333 | **0.284** | **0.328** | 0.288 | 0.332 |
| ETTm1 192 | 0.363 | 0.370 | 0.342 | 0.359 | **0.326** | **0.354** | 0.332 | 0.357 |
| ETTm1 336 | 0.393 | 0.390 | 0.375 | 0.379 | 0.368 | 0.377 | **0.364** | **0.376** |
| ETTm1 720 | 0.458 | 0.426 | 0.439 | 0.418 | 0.428 | 0.410 | **0.414** | **0.407** |
| ETTm1 avg | 0.383 | 0.384 | 0.363 | 0.372 | 0.352 | 0.367 | **0.349** | **0.368** |

## R.2 IS TRAINING DATA SIZE A LIMITING FACTOR FOR EXISTING LONG-TERM FORECASTING TRANSFORMERS?

We have observed a distribution shift phenomenon in fifty percent of the benchmark datasets: Traffic, ETTh2, and ETTm2. The model's performance demonstrates a significant enhancement with the use of only 70% training data samples compared to the standard training setting for long-term forecasting, as illustrated in table 28. While it has been argued that the transformer model exhibits a weakness where more training data fails to improve performance (Zeng et al., 2023), we contend that this issue is an inherent feature of each time series benchmark dataset, wherein changes in data distribution between historical and current data are not related to the transformer model. Nevertheless, further exploration of this phenomenon may lead to improved performance, and we thus leave it as a topic for future study.

Table 28: Less training data experiment.

| Tasks | | ETTm1 MSE MAE | ETTm2 MSE MAE | ETTh1 MSE MAE | ETTh2 MSE MAE | Weather MSE MAE | Electricity MSE MAE | Traffic MSE MAE |
|---|---|---|---|---|---|---|---|---|
| All samples | 96 | **0.316 0.347** | 0.169 0.248 | **0.383 0.391** | 0.281 0.330 | **0.150 0.188** | **0.141 0.233** | 0.419 0.269 |
| | 192 | **0.363 0.370** | 0.234 0.292 | **0.435 0.420** | 0.363 0.381 | **0.202 0.238** | **0.160 0.250** | 0.443 0.276 |
| | 336 | **0.393 0.390** | 0.294 0.339 | **0.479 0.442** | 0.411 0.418 | **0.260 0.282** | **0.173 0.263** | 0.460 0.283 |
| | 720 | **0.458 0.426** | 0.390 0.388 | **0.471 0.461** | 0.416 0.431 | **0.343 0.335** | **0.197 0.284** | 0.453 0.282 |
| | avg | **0.383 0.384** | 0.272 0.317 | **0.442 0.429** | 0.368 0.390 | **0.329 0.261** | **0.168 0.258** | 0.453 0.282 |
| 70% Samples | 96 | 0.350 0.431 | **0.163 0.242** | 0.425 0.431 | **0.272 0.325** | 0.245 0.263 | 0.157 0.239 | **0.404 0.263** |
| | 192 | 0.401 0.403 | **0.225 0.285** | 0.482 0.462 | **0.350 0.374** | 0.312 0.310 | 0.180 0.257 | **0.428 0.273** |
| | 336 | 0.440 0.428 | **0.284 0.324** | 0.528 0.485 | **0.394 0.411** | 0.382 0.352 | 0.197 0.270 | **0.444 0.471** |
| | 720 | 0.514 0.471 | **0.371 0.378** | 0.529 0.506 | **0.403 0.427** | 0.473 0.405 | 0.229 0.296 | **0.471 0.296** |
| | avg | 0.426 0.419 | **0.261 0.307** | 0.491 0.471 | **0.355 0.384** | 0.353 0.333 | 0.191 0.266 | **0.437 0.278** |

## R.3 EXPERIMENT ON REPLACING SELF-ATTENTION WITH LINEAR LAYER

(Zeng et al., 2023) suggests that a linear layer can be used as a substitute for the self-attention layer to achieve higher accuracy in transformer-based models. To highlight the effectiveness of self-attention in our model, we conduct experiments of replacing self-attention modules (e.g., attention over tokens and channels) with linear layer. The results are summarized in Table 29. Upon replacing channel-branch attention and token attention with a linear layer in CARD, we observe a consistent decline in accuracy across all datasets. The deterioration effect is particularly pronounced in the weather dataset, which contains more informative covariates, with a significant drop of over **13%**. These findings suggest that the self-attention scheme may be more effective in feature extraction than a simple linear layer for time series forecasting.

Table 29: The effectiveness of the self-attention scheme. The lookback length is set as 96. CARD(tMLP) uses an MLP layer to substitute the token attention layer in CARD, CARD(cMLP) uses an MLP layer to substitute the channel attention layer in CARD, CARD(dMLP) uses two MLP layers to substitute both token and channel attention, and CARD(oMLP) contains only the embedding layer and an MLP layer.

| Models | | CARD MSE MAE | CARD(tMLP) MSE MAE | CARD(cMLP) MSE MAE | CARD(dMLP) MSE MAE | CARD(oMLP) MSE MAE |
|---|---|---|---|---|---|---|
| ETTm1 | 96 | **0.316 0.347** | 0.333 0.369 | 0.324 0.357 | 0.355 0.376 | 0.356 0.376 |
| | 192 | **0.363 0.370** | 0.375 0.390 | 0.371 0.381 | 0.393 0.394 | 0.393 0.394 |
| | 336 | **0.393 0.390** | 0.405 0.409 | 0.403 0.402 | 0.425 0.415 | 0.424 0.414 |
| | 720 | **0.458 0.426** | 0.467 0.444 | 0.463 0.436 | 0.489 0.451 | 0.467 0.444 |
| | avg | **0.383 0.384** | 0.395 0.403 | 0.390 0.394 | 0.415 0.409 | 0.415 0.408 |
| Weather | 96 | **0.150 0.188** | 0.160 0.207 | 0.172 0.213 | 0.195 0.234 | 0.195 0.234 |
| | 192 | **0.202 0.238** | 0.211 0.254 | 0.220 0.255 | 0.240 0.270 | 0.240 0.270 |
| | 336 | **0.260 0.282** | 0.270 0.296 | 0.276 0.296 | 0.292 0.306 | 0.292 0.306 |
| | 720 | **0.343 0.335** | 0.358 0.351 | 0.353 0.346 | 0.364 0.353 | 0.364 0.353 |
| | avg | **0.239 0.261** | 0.250 0.277 | 0.255 0.277 | 0.272 0.291 | 0.273 0.291 |

## S EVALUATION ON IMPUTATION

We test the proposed model's imputation ability. We adopt the experimental settings in (Wu et al., 2023b) and results are reported in Table 30. CARD obtains top 2 ranks in 22/24 MSE scores and all MAE scores. In particular, in the Electricity dataset, CARD significantly reduces the MSE and MAE by 40% and 28% over the previous best results respectively. Those results suggest CARD may also generate good representations and thus can also work in the problem beyond forecasting.

Table 30: Imputation task. The time sequence is randomly masked 12.5%, 25%, 37.5%, and 50% points. MAE and MAE are reported. he best model is in boldface and the second best is underlined.

| Models | | | CARD | | TimesNet | | Dlinear | | LightTS | | ETSformer | | Stationary | | FEDformer | | Autoformer | | Informer | |
|---|---|---|---|---|---|---|---|---|---|---|---|---|---|---|---|---|---|---|---|---|
| Metric | | | MSE | MAE | MSE | MAE | MSE | MAE | MSE | MAE | MSE | MAE | MSE | MAE | MSE | MAE | MSE | MAE | MSE | MAE |
| ETTm1 | 12.5% | | 0.020 | **0.091** | **0.019** | 0.092 | 0.058 | 0.162 | 0.075 | 0.180 | 0.375 | 0.398 | 0.026 | 0.107 | 0.764 | 0.416 | .034 | 0.124 | 0.047 | 0.155 |
| | 25% | | 0.028 | 0.106 | **0.023** | **0.101** | 0.058 | 0.162 | 0.093 | 0.206 | 0.096 | 0.229 | 0.032 | 0.131 | 0.426 | 0.441 | 0.046 | 0.144 | 0.063 | 0.180 |
| | 37.5% | | 0.032 | 0.115 | **0.029** | **0.111** | 0.103 | 0.219 | 0.113 | 0.231 | 0.133 | 0.271 | 0.039 | 0.131 | 0.445 | 0.459 | 0.057 | 0.161 | 0.079 | 0.200 |
| | 50% | | **0.033** | **0.114** | 0.036 | 0.124 | 0.132 | 0.248 | 0.134 | 0.255 | 0.186 | 0.323 | 0.047 | 0.145 | 0.089 | 0.218 | 0.067 | 0.174 | 0.093 | 0.218 |
| | avg | | 0.028 | **0.107** | 0.027 | **0.107** | 0.093 | 0.206 | 0.104 | 0.218 | 0.120 | 0.253 | 0.036 | 0.126 | 0.062 | 0.177 | 0.051 | 0.150 | 0.071 | 0.188 |
| ETTm2 | 12.5% | | 0.019 | **0.076** | 0.018 | 0.080 | 0.062 | 0.166 | 0.034 | 0.127 | 0.108 | 0.239 | 0.021 | 0.088 | 0.056 | 0.159 | 0.023 | 0.092 | 0.133 | 0.270 |
| | 25% | | 0.021 | 0.081 | 0.020 | 0.085 | 0.085 | 0.196 | 0.042 | 0.143 | 0.164 | 0.294 | 0.024 | 0.096 | 0.080 | 0.195 | 0.026 | 0.101 | 0.135 | 0.272 |
| | 37.5% | | **0.022** | **0.086** | 0.023 | 0.091 | 0.106 | 0.222 | 0.051 | 0.159 | 0.237 | 0.356 | 0.027 | 0.103 | 0.110 | 0.231 | 0.030 | 0.108 | 0.155 | 0.293 |
| | 50% | | **0.024** | **0.090** | 0.026 | 0.098 | 0.131 | 0.247 | 0.059 | 0.174 | 0.323 | 0.421 | 0.030 | 0.108 | 0.156 | 0.276 | 0.035 | 0.119 | 0.200 | 0.333 |
| | avg | | **0.022** | **0.083** | 0.027 | 0.088 | 0.096 | 0.208 | 0.046 | 0.151 | 0.208 | 0.327 | 0.026 | 0.099 | 0.101 | 0.215 | 0.029 | 0.105 | 0.156 | 0.292 |
| ETTh1 | 12.5% | | **0.044** | **0.138** | 0.057 | 0.159 | 0.151 | 0.267 | 0.240 | 0.345 | 0.126 | 0.263 | 0.060 | 0.165 | 0.070 | 0.190 | 0.074 | 0.182 | 0.114 | 0.234 |
| | 25% | | **0.054** | **0.154** | 0.069 | 0.178 | 0.180 | 0.292 | 0.265 | 0.364 | 0.169 | 0.304 | 0.080 | 0.189 | 0.106 | 0.236 | 0.090 | 0.203 | 0.140 | 0.262 |
| | 37.5% | | **0.069** | **0.174** | 0.084 | 0.196 | 0.215 | 0.318 | 0.296 | 0.382 | 0.220 | 0.347 | 0.102 | 0.212 | 0.124 | 0.258 | 0.109 | 0.222 | 0.174 | 0.293 |
| | 50% | | **0.085** | **0.194** | 0.102 | 0.215 | 0.257 | 0.347 | 0.334 | 0.404 | 0.293 | 0.402 | 0.133 | 0.240 | 0.165 | 0.299 | 0.137 | 0.248 | 0.215 | 0.325 |
| | avg | | **0.063** | **0.165** | 0.078 | 0.187 | 0.201 | 0.306 | 0.284 | 0.373 | 0.202 | 0.239 | 0.094 | 0.201 | 0.117 | 0.246 | 0.103 | 0.214 | 0.161 | 0.279 |
| ETTh2 | 12.5% | | **0.040** | **0.122** | 0.040 | 0.130 | 0.100 | 0.216 | 0.101 | 0.231 | 0.187 | 0.319 | 0.042 | 0.133 | 0.096 | 0.212 | 0.044 | 0.138 | 0.976 | 0.754 |
| | 25% | | **0.041** | **0.128** | 0.046 | 0.141 | 0.127 | 0.247 | 0.115 | 0.246 | 0.279 | 0.390 | 0.049 | 0.147 | 0.137 | 0.258 | 0.050 | 0.149 | 0.322 | 0.444 |
| | 37.5% | | **0.045** | **0.135** | 0.052 | 0.151 | 0.158 | 0.276 | 0.125 | 0.257 | 0.400 | 0.465 | 0.056 | 0.158 | 0.187 | 0.304 | 0.060 | 0.163 | 0.353 | 0.462 |
| | 50% | | **0.051** | **0.146** | 0.060 | 0.162 | 0.183 | 0.299 | 0.136 | 0.268 | 0.602 | 0.572 | 0.065 | 0.170 | 0.232 | 0.341 | 0.068 | 0.173 | 0.369 | 0.472 |
| | avg | | **0.044** | **0.133** | 0.049 | 0.146 | 0.142 | 0.259 | 0.119 | 0.250 | 0.367 | 0.436 | 0.053 | 0.152 | 0.163 | 0.279 | 0.055 | 0.156 | 0.337 | 0.452 |
| Weather | 12.5% | | 0.027 | **0.040** | 0.025 | 0.045 | 0.039 | 0.084 | 0.047 | 0.101 | 0.057 | 0.141 | 0.027 | 0.051 | 0.041 | 0.107 | 0.026 | 0.047 | 0.037 | 0.093 |
| | 25% | | **0.029** | **0.042** | **0.029** | 0.052 | 0.048 | 0.103 | 0.052 | 0.111 | 0.065 | 0.155 | 0.029 | 0.056 | 0.064 | 0.163 | 0.030 | 0.054 | 0.042 | 0.100 |
| | 37.5% | | 0.033 | **0.045** | 0.031 | 0.057 | 0.057 | 0.117 | 0.058 | 0.121 | 0.081 | 0.180 | 0.033 | 0.062 | 0.107 | 0.229 | 0.032 | 0.060 | 0.049 | 0.111 |
| | 50% | | 0.036 | **0.048** | 0.034 | 0.062 | 0.066 | 0.134 | 0.065 | 0.133 | 0.102 | 0.207 | 0.037 | 0.068 | 0.138 | 0.312 | 0.037 | 0.067 | 0.053 | 0.114 |
| | avg | | 0.031 | **0.044** | 0.030 | 0.054 | 0.052 | 0.110 | 0.065 | 0.133 | 0.102 | 0.207 | 0.037 | 0.068 | 0.183 | 0.312 | 0.031 | 0.057 | 0.045 | 0.104 |
| Electricity | 12.5% | | **0.043** | **0.131** | 0.085 | 0.202 | 0.092 | 0.214 | 0.102 | 0.229 | 0.196 | 0.321 | 0.093 | 0.210 | 0.107 | 0.237 | 0.089 | 0.210 | 0.218 | 0.326 |
| | 25% | | **0.049** | **0.142** | 0.089 | 0.206 | 0.118 | 0.247 | 0.121 | 0.252 | 0.207 | 0.332 | 0.097 | 0.214 | 0.120 | 0.251 | 0.096 | 0.220 | 0.219 | 0.326 |
| | 37.5% | | **0.059** | **0.159** | 0.094 | 0.213 | 0.144 | 0.276 | 0.141 | 0.273 | 0.219 | 0.344 | 0.102 | 0.220 | 0.136 | 0.266 | 0.104 | 0.229 | 0.222 | 0.328 |
| | 50% | | **0.069** | **0.172** | 0.100 | 0.221 | 0.175 | 0.305 | 0.160 | 0.293 | 0.235 | 0.357 | 0.108 | 0.228 | 0.158 | 0.284 | 0.113 | 0.239 | 0.228 | 0.331 |
| | avg | | **0.055** | **0.151** | 0.092 | 0.210 | 0.132 | 0.260 | 0.131 | 0.262 | 0.214 | 0.339 | 0.100 | 0.218 | 0.130 | 0.259 | 0.101 | 0.225 | 0.222 | 0.328 |

