# OpenReview forum: "CARD: Channel Aligned Robust Blend Transformer for Time Series Forecasting"
_ICLR.cc/2024/Conference — ICLR 2024 poster_

### Official Review · Reviewer_fhNE · 2023-10-29

**Soundness:** 3 good
**Presentation:** 3 good
**Contribution:** 3 good
**Rating:** 6
**Confidence:** 3

**Summary:**

This paper proposes a novel time series forecasting method, called CARD. Compared with previous works, CARD builds an effective transformer to employ cross-channel information. CARD utilizes attention over tokens and attention over channels to robustly align the information among different channels. Furthermore, the token blend module introduces multi-scaling structural information. To bolster the model’s ability to concentrate on forecasting for the near future, this paper develops a robust signal decay-based loss function. The comprehensive experiments demonstrated the superiority of CARD for long-term forecasting, short-term forecasting, and other prediction tasks.

**Strengths:**

(1) The method proposed in this paper is highly effective, taking into consideration the interactions among different channels, thereby mitigating to some extent the overfitting issues commonly associated with previous channel-dependent models.

(2) The experiments are highly comprehensive, assessing the model's performance across long-term forecasting, short-term forecasting, and other prediction tasks. In addition, this paper conducts extensive ablation studies, including hyperparameter sensitivity analysis experiments, loss function analysis experiments, and so forth.

(3) The paper is generally well-written and easy to understand.

**Weaknesses:**

(1) The paper analyzed the complexity of the proposed model, which is an advantage. However, this paper did not specifically outline the complexities of the baseline models and there were no direct comparisons with the baselines on the running time or efficiency.

(2) Some of the experimental results are somewhat confusing. For example, in Table 18, the average performance of CARD and wo. hidden are identical, but the diffs are not 0.

(3) In the ablation study of the signal decay-based loss function, i.e. Table 12 and Table 13, I observe that the method proposed in this paper does not exhibit a significant advantage compared to the SOTA baseline without the robust loss. Furthermore, it would be better if the authors could provide test results on a broader range of datasets.

(4) The paper claims that the proposed method can effectively mitigate overfitting issues. Although some empirical analysis has been provided, supplementing with appropriate theoretical analysis would be more convincing.

**Questions:**

The concerns include the comparison with baselines on the complexity, the confusing experimental results, in-depth insight into the ablation study, and the theoretical analysis of how the model mitigates overfitting issues. Please see the weaknesses for details.

---

> ### Author Response · Authors · 2023-11-17
>
> We would like to express our sincere gratitude to Reviewer fhNE for their positive assessment of our work, specifically acknowledging the comprehensiveness of our experiments and the proposed structure. Your recognition of the significance we attributed to the appendix experiments is truly gratifying. We want to assure you that we are fully committed to addressing and resolving any concerns you have raised.
>
> ## Q1:Model complexities/Running time
> As we use the patching trick, the order of the total complexity would be the same as PatchTST and Crossformer, which is $\mathcal{O}(L^2/S^2)$. Some Transformer type models (e.g., FEDformer and Autoformer) may even break the quadratic dependent in $L$ and reach linear or nearly linear complexity in $L$. Other CNN and RNN type models (e.g., TimesNet and FilM) by nature maintain the $O(L)$ complexity.  When $S$ is set as a not-very-small number (e.g. $S\approx O(\sqrt{L})$), our model's complexity can also be nearly linear. The condition $S\approx O(\sqrt{L})$ is not very restrictive. Take $L = 900$ as an example, the length of $S=30$ would be enough. For the case $L = 96$, the corresponding $S$ would be around $10$.
>
> The results of the detailed experiment on running time are reported in the following table. We use the TimesNet's codebase. The input/forecasting lengths are set as 96/96 and we keep the batch size the same for all benchmarks and run the experiments on a single A100/80G GPU. Our proposed model yields comparable running time to transformer baselines as well as linear complexity baselines except Dlinear, which implies in practice model could also behave like a linear time model and won't introduce overhead computational cost.
>
> |                      |           | Card   | autoformer | patchTST | crossformer | fedformer | timesnet | micn   | Dlinear | FilM   | ETSformer |
> | -------------------- | --------- | ------ | ---------- | -------- | ----------- | --------- | -------- | ------ | ------- | ------ | --------- |
> | hidden=16,batch=128  | train     | 0.0197 | 0.1091     | 0.0164   | 0.2512      | 0.2107    | 0.0672   | 0.0423 | 0.0074  | 0.0747 | 0.0714    |
> | etth1                | inference | 0.0046 | 0.0102     | 0.0021   | 0.0127      | 0.0178    | 0.0132   | 0.0030 | 0.0009  | 0.0123 | 0.0061    |
> | hidden=128,batch=128 | train     | 0.0779 | 0.1525     | 0.0785   | 0.1186      | 0.2189    | 0.2457   | 0.0613 | 0.0330  | oom    | 0.1354    |
> | Weather              | inference | 0.0048 | 0.0139     | 0.0036   | 0.0123      | 0.0378    | 0.0224   | 0.0038 | 0.0014  | oom    | 0.0092    |
> | hidden=128,batch=32  | train     | 0.2156 | 0.0835     | 0.3280   | oom         | 0.1903    | oom      | 0.0405 | 0.0163  | oom    | 0.1174    |
> | ECL                  | inference | 0.0064 | 0.0160     | 0.0052   | oom         | 0.0349    | oom      | 0.0045 | 0.0021  | oom    | 0.0103    |
> | hidden=128,batch=12  | train     | 0.2271 | 0.0960     | 0.1329   | oom         | 0.1649    | 0.6048   | 0.0322 | 0.0139  | oom    | 0.0607    |
> | Traffic              | inference | 0.0101 | 0.0153     | 0.0052   | oom         | 0.0369    | 0.0418   | 0.0074 | 0.0058  | oom    | 0.0223    |
>
>
> ## Q2:Inconsistent value
> We sincerely apologize for the typographical error. The correct value for wo.hidden weather(720) should be 0.351, instead of 0.341, and the average should be 0.242. This mistake occurred due to inadvertently typing a 4 instead of a 5. We will diligently verify all the numbers to ensure accuracy and regret any confusion caused by this error.

---

> ### Author Response · Authors · 2023-11-17
>
> ## Q3:Model performance without Robust loss
> As depicted in the subsequent table, the model itself is capable of achieving state-of-the-art (SOTA) performance using the original MSE loss. The introduction of the robust loss proposed in this paper further enhances the performance. Moreover, it is noteworthy that for the traffic dataset, the original MSE loss yields significantly better results. Therefore, it is important to highlight that the primary driver behind the observed improvements is the proposed model itself, rather than relying solely on the choice of the loss function.
>
>
> |          | ETTm2 |       | ETTh2 |       | ECL   |       | Traffic |       |
> | ------ | ----- | ----- | ----- | ----- | ----- | ----- | ------- | ----- |
> |       | MSE       | MAE   | MSE   | MAE   | MSE   | MAE   |MSE   | MAE   |
> | 96           | 0.171 | 0.256 | 0.284 | 0.338 | 0.146 | 0.244 | 0.400   | 0.275 |
> | 192          | 0.241 | 0.302 | 0.373 | 0.391 | 0.164 | 0.260 | 0.426   | 0.288 |
> | 336          | 0.302 | 0.340 | 0.421 | 0.430 | 0.181 | 0.278 | 0.439   | 0.290 |
> | 720          | 0.398 | 0.397 | 0.423 | 0.442 | 0.215 | 0.305 | 0.475   | 0.312 |
> | avg          | 0.278 | 0.324 | 0.375 | 0.400 | 0.176 | 0.272 | 0.435   | 0.291 |
> | avg_with_robustloss | 0.271 | 0.316 | 0.367 | 0.390 | 0.169 | 0.258 | 0.450    | 0.278 |
> | sec_best_avg | 0.283 | 0.327 | 0.483 | 0.406 | 0.187 | 0.295 | 0.488   | 0.316 |
> ## Q4:Theoretical analysis for mitigating overfitting issues
>
> Thanks for your suggestion. In this paper, we mainly introduce the exponential smoothing (EMA) module and weighted robust loss to mitigate the overfitting.
>
> We believe the EMA part would use a weighted mean to average out some noise. We may use the following simple example to illustrate it. Let $\{x_1,....,x_{t}\}$ be sequences of i.i.d $\sigma^2$ sub-Gaussian variables and Let $f(t) = (1-\beta)\sum_{i=1}^{t}\beta^{t-i}x_i$ with some $\beta\in (0,1)$. One may verify $f(t) = (1-\beta)x_{t} + \beta f(t-1)$, which implies $\{f(t)\}$ is the sequence of $\{x_t\}$ after EMA. Via the Hoeffiding's inequality, there exist some constant $C>0$ such that with $1-\delta$, we have:
>
> \begin{align}
>     |f(t)-f(t-1)| \le \sigma\sqrt{(1-\beta)^2\left(1+\beta^2\right)C^{-1}\log\left(\frac{1}{\delta}\right)}.
> \end{align}
>
> If we set $\beta \to 0$, $f(t)\to x_t$ and above inequality shows the divergence between $x_t$ and $x_{t-1}$ would be $O(\sigma)$. When change $\beta$ toward $1$, the divergence decreased on the order of $O(\sigma^2\sqrt{1-\beta)^2(1+\beta^2)})$, which is faster than $O(\sigma)$. It implies that if the sequence $\{x_t\}$ contains high-frequency noise, the smoothed sequence $f(t)$ will have a higher chance of wiping it out and it eventually stabilizes the attention module. We will add those discussions into the final version.
>
> Another design to tackle the overfitting is the weighted loss.  In Appendix H of the paper, we use toy case with input length 2 to show the proposed loss may give better estimation accuracy.
>
>
> However,we have to admit that performing end-to-end theoretical analysis for the proposed deep model can be very complicated and beyond the scope of this paper.

---

> > ### Author Response · Authors · 2023-11-21
> > **Response to Reviewer fhNE (before the end of discussion)**
> >
> > Dear Reviewer fhNE,
> >
> > Since the End of author/reviewer discussions is just in one day, may we know if our response addresses your main concerns? If so, we kindly ask for your reconsideration of the score.
> >
> > Should you have any further advice on the paper and/or our rebuttal, please let us know and we will be more than happy to engage in more discussion and paper improvements. We would really appreciate it if our next round of communication could leave time for us to resolve any of your remaining or new questions.
> >
> > Thank you so much for devoting time to improving our work!

---

### Official Review · Reviewer_eTmJ · 2023-10-31

**Soundness:** 3 good
**Presentation:** 3 good
**Contribution:** 3 good
**Rating:** 6
**Confidence:** 4

**Summary:**

This paper introduces the Channel Aligned Robust Blend Transformer (CARD), a specialized transformer model designed for time series forecasting. It acknowledges the success of transformer models in this field, particularly due to the channel-independent (CI) strategy that improves training robustness. However, the CI approach overlooks the correlation among different channels, limiting the model's forecasting capacity. To address this limitation, CARD incorporates a channel-aligned attention structure to capture temporal correlations among signals and dynamical dependencies among multiple variables over time. Additionally, a token blend module is introduced to efficiently utilize multi-scale knowledge by generating tokens with different resolutions. The authors also present a robust loss function that considers prediction uncertainties to mitigate overfitting. Experimental evaluations on various long-term and short-term forecasting datasets demonstrate that CARD outperforms state-of-the-art methods in time series forecasting. The article concludes by discussing related work, presenting the detailed model architecture, describing the loss function design, and providing the results of numerical experiments and further analysis.

**Strengths:**

1. This paper introduces a novel transformer model specifically designed for time series forecasting. It addresses the limitations of the channel-independent (CI) strategy by incorporating a channel-aligned attention structure and a token blend module to capture correlations among different channels and utilize multi-scale information effectively.

2. This paper evaluates the proposed CARD model on multiple long-term and short-term forecasting datasets, comparing it against state-of-the-art methods, and demonstrates its superiority over existing techniques in various prediction-based tasks.

3. This paper introduces a robust loss function specifically designed for time series forecasting. By considering prediction uncertainties and weighting the importance of forecasting over a finite horizon, the proposed loss function addresses the potential issue of overfitting noises. This contribution enhances the model's ability to make accurate predictions.

4. This paper follows a well-organized structure. The inclusion of figures, such as the model architecture figure and experimental result figures, aids in understanding the proposed methods.

**Weaknesses:**

1. Experimental Concerns:

   (1) Fixed-parameter EMA: The article directly applies a fixed-parameter Exponential Moving Average (EMA) without providing specific explanations regarding the potential drawbacks of using a learnable-parameter EMA. The advantages or disadvantages of employing a learnable-parameter EMA are not thoroughly explored.

   (2) Lack of comparison with different lookback values: The article fails to compare the results of different lookback values (the number of previous time steps considered) on the M4 dataset. Such a comparison would provide a more comprehensive analysis and insights into the impact of lookback on forecasting performance.

   (3) Inconsistency in short-term forecasting results: While the article highlights the third point of short-term forecasting, the visual results presented in the figures do not seem to align well with the data shown in the tables. Additionally, the evaluation of short-term forecasting appears to be limited to only the M4 dataset, which may not be sufficient to fully demonstrate the model's performance.

**Questions:**

1. Presentation Aspect:
   It would be beneficial for the article to include a schematic diagram illustrating the limitations of previous models at the beginning, to better introduce the motivation and background of the study. This would help readers understand the context more clearly and provide a more coherent structure to the article.

2. Experimental concerns are listed in the weakness part.

---

> ### Author Response · Authors · 2023-11-17
>
> We would like to express our sincere gratitude to Reviewer eTmJ for providing a positive evaluation of our work's intriguing and challenging contributions. We are truly encouraged by the thorough and well-summarized strengths section that you have provided. We greatly appreciate your detailed and insightful comments, as they have been instrumental in helping us improve our work. Rest assured, we are fully committed to addressing any concerns you have raised.
>
> ## Q1: Learnabe decay weight VS Fixed decay weight
> The results of the learnable decay weight experiment are presented in the following table. The numerical outcomes are found to be very similar to those obtained with the fixed decay weight. However, it is important to note that the training speed could be significantly slower when using the learnable decay weight requiring up to 50\% extra training time. Due to this time-consuming aspect, we have chosen to utilize a fixed decay weight in our approach.
>
> | ETTm1 | Learnable |       | Fixed |       | Diff  |        |
> | ----- | --------- | ----- | ----- | ----- | ----- | ------ |
> |       | MSE       | MAE   | MSE   | MAE   | MSE   | MAE    |
> | 96    | 0.321     | 0.345 | 0.316 | 0.347 | 0.005 | -0.002 |
> | 192   | 0.363     | 0.368 | 0.363 | 0.370 | 0.000 | -0.002 |
> | 336   | 0.394     | 0.390 | 0.392 | 0.390 | 0.002 | 0.000  |
> | 720   | 0.464     | 0.428 | 0.458 | 0.425 | 0.006 | 0.003  |
> | avg   | 0.386     | 0.383 | 0.382 | 0.383 | 0.003 | 0.000  |
>
> | ETTm2 | Learnable |       | Fixed |       | Diff   |        |
> | ----- | --------- | ----- | ----- | ----- | ------ | ------ |
> |       | MSE       | MAE   | MSE   | MAE   | MSE    | MAE    |
> | 96    | 0.169     | 0.247 | 0.169 | 0.248 | 0.000  | -0.001 |
> | 192   | 0.232     | 0.289 | 0.234 | 0.292 | -0.002 | -0.003 |
> | 336   | 0.292     | 0.328 | 0.294 | 0.339 | -0.002 | -0.011 |
> | 720   | 0.390     | 0.388 | 0.390 | 0.388 | 0.000  | 0.000  |
> | avg   | 0.271     | 0.313 | 0.272 | 0.317 | -0.001 | -0.004 |
>
> | ETTh1 | Learnable |       | Fixed |       | Diff  |       |
> | ----- | --------- | ----- | ----- | ----- | ----- | ----- |
> |       | MSE       | MAE   | MSE   | MAE   | MSE   | MAE   |
> | 96    | 0.384     | 0.391 | 0.383 | 0.391 | 0.001 | 0.000 |
> | 192   | 0.438     | 0.422 | 0.435 | 0.420 | 0.003 | 0.002 |
> | 336   | 0.482     | 0.444 | 0.479 | 0.442 | 0.003 | 0.002 |
> | 720   | 0.475     | 0.463 | 0.471 | 0.461 | 0.004 | 0.002 |
> | avg   | 0.445     | 0.430 | 0.442 | 0.429 | 0.003 | 0.002 |
>
> | ETTh2 | Learnable |       | Fixed |       | Diff   |        |
> | ----- | --------- | ----- | ----- | ----- | ------ | ------ |
> |       | MSE       | MAE   | MSE   | MAE   | MSE    | MAE    |
> | 96    | 0.285     | 0.331 | 0.281 | 0.330 | 0.004  | 0.001  |
> | 192   | 0.362     | 0.381 | 0.363 | 0.381 | -0.001 | 0.000  |
> | 336   | 0.411     | 0.417 | 0.411 | 0.418 | 0.000  | -0.001 |
> | 720   | 0.414     | 0.432 | 0.416 | 0.431 | -0.002 | 0.001  |
> | avg   | 0.368     | 0.390 | 0.368 | 0.390 | 0.000  | 0.000  |
>
> | Weather | Learnable |       | Fixed |       | Diff   |        |
> | ------- | --------- | ----- | ----- | ----- | ------ | ------ |
> |         | MSE       | MAE   | MSE   | MAE   | MSE    | MAE    |
> | 96      | 0.153     | 0.206 | 0.150 | 0.188 | 0.003  | 0.018  |
> | 192     | 0.207     | 0.249 | 0.202 | 0.238 | 0.005  | 0.011  |
> | 336     | 0.258     | 0.287 | 0.260 | 0.282 | -0.002 | 0.005  |
> | 720     | 0.345     | 0.338 | 0.343 | 0.353 | 0.002  | -0.015 |
> | avg     | 0.241     | 0.270 | 0.239 | 0.265 | 0.002  | 0.005  |
>
>
> ## Q2: About the short-term forecasting M4 benchmark
> Following the discussion with reviewer ajLB on short-term forecasting using the M4 dataset, we have decided to relocate this section to the supplementary information as an initial exploration. In exchange, we will incorporate relevant ablation study findings from the supplementary information into the main draft. We concur with reviewer eTmJ's assessment that relying solely on the M4 dataset for short-term forecasting is inadequate, especially considering the issue of different input lengths for baseline models. We acknowledge that our incorporation of the settings proposed in the recent Timesnet paper lacked a deeper analysis. It is crucial to recognize that as the M4 dataset is a competition dataset, most participants leverage ensemble methods to enhance their performance. However, introducing ensemble methods would significantly complicate the problem and deviate from our main research focus. Additionally, we acknowledge that certain baseline models in the M4 dataset employ different input lengths due to the competition's lack of requirement for uniformity. To strengthen the soundness of our findings, we intend to explore alternative datasets for the short-term forecasting task, an aspect we plan to address in our future work.

---

> > ### Author Response · Authors · 2023-11-21
> > **Thank you for the review! Have we clearly addressed the concerns?**
> >
> > Dear Reviewer eTmJ,
> >
> > We greatly appreciate the time you took to review our paper.  Due to the short duration of the author-reviewer discussion phase, we would appreciate your feedback on whether your main concerns have been adequately addressed.
> >
> > We are ready and willing to provide further explanations and clarifications if necessary. Thank you very much!

---

### Official Review · Reviewer_gMyA · 2023-11-10

**Soundness:** 3 good
**Presentation:** 3 good
**Contribution:** 2 fair
**Rating:** 5
**Confidence:** 4

**Summary:**

This paper presents CARD, a particular Transformer model for time series forecasting.  CARD contains three core designs：

* A channel-aligned attention structure
* A token blend module
* A robust loss function for time series forecasting

The author provides extensive experiments to verify the effectiveness of the proposed method in multiple long-term, short-term forecasting and anomaly detection datasets.

**Strengths:**

1. The paper is well written and easy to follow, with good figures for explanation.

2. The proposed method combines various time series forecasting effective design ideas.

3. The proposed method has conducted experiments on multiple time series forecasting tasks.

**Weaknesses:**

**Method design**

The proposed approach needs more novelty. Some of the main design ideas are presented and verified by the previous work, such as channel independence, patchify, EMA, and dynamic projection technique.

**Experimental rationality**

* The main experimental setting needs to be consistent with the SOTA method to be convincing, and the experimental results need more justification.

* The rationality of some analytical experiments needs further verification.

**Questions:**

1. In long-term forecasting tasks, I have questions regarding the experimental settings of the current state-of-the-art model PatchTST. The input length for PatchTST/42 is 336, and for PatchTST/64 it is 512. I am curious whether CARD can outperform PatchTST if the authors use the same input length. If you aim to validate your experimental settings, you should use the official code of the SoTA model and adjust the parameters to ensure a fair comparison.

2. Channel independence introduces a significant computational overhead. To mitigate this issue, the authors employ dynamic projection (DP) to project head dimensions from d_head to a fixed $r$ with $r ≪ C$. They demonstrate through ablation experiments involving projected dimensions of 1, 8, and 16 on ETT datasets that DP does not compromise forecasting accuracy. However, it's worth noting that ETT datasets only consist of 7 channels, which may not be sufficient to draw a definitive conclusion. Therefore, it remains unclear how this technique would perform in datasets with a larger number of variables, such as Traffic (862) or ECL (321).

3. In ablation experiments, our primary focus is on evaluating the effectiveness of the new design between CARD and PatchTST. Instead of solely changing parameters, we aim to demonstrate the method's rationality and validity with greater certainty through more detailed ablation studies.

4. In experiments with less training data, using only 70% of the training data samples compared to the standard training setting for long-term forecasting may not be sufficient.

---

> ### Author Response · Authors · 2023-11-17
>
> We extend our sincere appreciation to Reviewer gMyA for providing a positive evaluation of our methodology's design and the clarity of our visualized explanation. We are particularly grateful for the opportunity you have given us to elaborate on the two experimental settings and address the related question, as we have invested significant time in investigating this particular issue. Please rest assured that we are fully committed to addressing and resolving any concerns you have raised.
>
> ## Q1: Some of the main design ideas are presented and verified by the previous work, such as channel independence and etc.
>
>
> In our paper, we did not utilize channel independence. Instead, our primary objective is to develop a transformer model that effectively harnesses channel-dependent (CD) information. This focus is stated in the abstract, where we introduce the CARD model's channel-aligned attention structure that aims to capture dependencies among multiple variables over time, and other parts of the paper also echo this statement (e.g., summary of contribution, Figure 1, and Section 3.3).
>
> The core innovation of our paper lies in addressing an important research question: how to construct an efficient transformer for CD settings. In existing literature, CD transformers, such as Autoformer and FEDformer, generally underperform compared to channel-independent (CI) transformers like PatchTST. Various experiments by Han et al., 2023 suggest that enhancing the robustness of CD transformers is crucial. We therefore introduced the EMA module to improve the model's stability and considered a robust loss design. Additionally, to maintain reasonable computational cost and align the information within patch, dynamic projection and dual attention are introduced respectively.
> While we agree with the reviewer that similar structures may have been explored in deep learning literature, we believe such a combination is new in CD transformer design in the time series forecasting domain. Our focus is not on claiming the invention of these structures but on their effective combination to improve CD transformer performance, which we consider a significant novelty of our research.
>
> Moreover, we want to highlight our robust loss. In recent literature (e.g., Autoformer, FEDformer, and PatchTST), the major point is trying to design new model structures but not focusing on training procedures. In this paper, we analyze the variance-varying issue in multi-step forecasting and propose a weighted loss. This relatively simple modification not only enhances the performance of our model but also improves other models, as demonstrated in Table 4. To our best knowledge, this loss design is also new in recent transformer time series forecasting literature.
>
> ## Q2: Comparision setting: fixed short input or better performance
> Perhaps we didn't clarify it enough, which may have led to confusion for reviewer gMyA. We want to highlight that there are two distinct experimental settings for long-term forecasting benchmarks in our study. The first approach, used by Autoformer, FEDformer, Timesnet, etc., involves a fixed input length of 96 and forecasting for all horizons. These models compare accuracy across different algorithms, and we refer to this approach as "fixed short input". The complete table of results for this particular setting can be found in Appendix E, specifically in Table 8.
>
> On the other hand, models like PatchTST, Dlinear, Crossformer, and some other baselines focus more on the final result by using a much longer input length. Some of them even tune the input length to achieve better performance. As the reviewer correctly noted, PatchTST reports results for input lengths of both 336 and 512 in their main table. However, it is worth mentioning that PatchTST also presents results for an input length of 720 in its paper appendix table 9, although it is not as good as the 336 and 512 cases.
>
> We refer to them as the "better performance" approach. The complete table of results for this particular setting can be found in Appendix F, specifically in Table 10.
>
> We believe that both approaches are valid and contribute to testing different aspects of the algorithms. One approach evaluates the forecasting ability with a limited input length, while the other tests the ability to achieve better performance using longer inputs. Therefore, we have included results for both settings. The reason for using an input length of 720 is that we achieved better performance in this hard setting length, whereas PatchTST's performance slightly deteriorated. However, we didn't want to lower the reported performance of the baseline models, so we included their best performance setting which was for an input length of 512, to align with the "better performance" approach.
>
> We hope that this clarification makes our approach clearer.  If reviewer gMyA still prefers the results on 336/512 input length, we are also more than happy to include them in the final version.

---

> ### Author Response · Authors · 2023-11-17
>
> ## Q3: Dynamic projection with Large channel datasets like Traffic and ECL
> Thanks for your suggestion. We've added the results of Weather, ECL, and Traffic datasets under the same setting in Appendix N.2. The projecteddimension varies in 1,8,16. The results are summarized in the following table. We have observed similar performance behavior, suggesting that altering the dynamic projection dimension does not significantly impact the forecasting accuracy.
>
> | Weather | MSE   | MAE   | MSE   | MAE   | MSE   | MAE   | MSE   | MAE   |
> | ------- | ----- | ----- | ----- | ----- | ----- | ----- | ----- | ----- |
> |Proj Dim| 1     | 1     | 8     | 8     | 16    | 16    | 32    | 32    |
> | 96      | 0.152 | 0.189 | 0.150 | 0.188 | 0.147 | 0.186 | 0.151 | 0.189 |
> | 192     | 0.204 | 0.240 | 0.202 | 0.238 | 0.201 | 0.237 | 0.201 | 0.238 |
> | 336     | 0.263 | 0.284 | 0.260 | 0.282 | 0.262 | 0.283 | 0.259 | 0.282 |
> | 720     | 0.343 | 0.353 | 0.343 | 0.353 | 0.341 | 0.352 | 0.339 | 0.350 |
> | avg     | 0.241 | 0.266 | 0.239 | 0.265 | 0.238 | 0.264 | 0.238 | 0.265 |
>
> | ECL      | MSE   | MAE   | MSE   | MAE   | MSE   | MAE   | MSE   | MAE   |
> | -------- | ----- | ----- | ----- | ----- | ----- | ----- | ----- | ----- |
> | Proj Dim | 1     | 1     | 8     | 8     | 16    | 16    | 32    | 32    |
> | 96       | 0.142 | 0.234 | 0.141 | 0.233 | 0.140 | 0.232 | 0.141 | 0.233 |
> | 192      | 0.165 | 0.256 | 0.160 | 0.250 | 0.157 | 0.247 | 0.158 | 0.248 |
> | 336      | 0.180 | 0.272 | 0.173 | 0.263 | 0.168 | 0.256 | 0.171 | 0.261 |
> | 720      | 0.209 | 0.298 | 0.197 | 0.284 | 0.187 | 0.273 | 0.196 | 0.281 |
> | avg      | 0.174 | 0.265 | 0.168 | 0.258 | 0.163 | 0.252 | 0.166 | 0.255 |
>
> | Traffic  | MSE   | MAE   | MSE   | MAE   | MSE   | MAE   |
> | -------- | ----- | ----- | ----- | ----- | ----- | ----- |
> | Proj Dim | 1     | 1     | 8     | 8     | 16    | 16    |
> | 96       | 0.413 | 0.274 | 0.419 | 0.269 | 0.416 | 0.270 |
> | 192      | 0.447 | 0.273 | 0.443 | 0.276 | 0.435 | 0.268 |
> | 336      | 0.462 | 0.278 | 0.460 | 0.283 | 0.447 | 0.275 |
> | 720      | 0.490 | 0.299 | 0.490 | 0.299 | 0.482 | 0.294 |
> | avg      | 0.453 | 0.281 | 0.453 | 0.282 | 0.445 | 0.277 |
>
> ## Q4: Ablation for structures to verify the difference between CARD and PatchTST.
> From the model structure perspective, our model has two major differences: attention over hidden and channel-aligned modules, which are trying to address the PatchTST potential two drawbacks: ignorance of the cross-channel information and lack of usage within patch information. In Table 18, we examine the performance changes when removing those two structures and verify the effectiveness of those two parts. Moreover, we have also included our ablation analysis for architecture variants in the appendix, specifically in Figure N.1, which visually demonstrates the differences. If they are more engaging for the readers, we can certainly move this analysis to the main draft. The only reason we did not include it here initially was due to limited space constraints.
>
>
> ## Q5: Data size for transformer
> The purpose of this experiment is to challenge the claim made in Dlinear that the transformer model cannot benefit from more data. We believe this claim to be an overstatement, as we demonstrate that even a slight increase in dataset size, from 70\% to 100\%, can result in performance improvement. Although we could show even larger improvements with an even smaller dataset size, we deliberately avoid getting into a debate about whether a 10\% data size is too small and leads to underfitting the model. We acknowledge that these findings are preliminary. To truly test whether the transformer model follows a power law accuracy curve, we would need a much larger dataset. Our goal is simply to highlight that the claim made in Dlinear is underexplored at the very least, and we hope to encourage further investigation within the research community.

---

> ### Comment · Reviewer_gMyA · 2023-11-21
>
> Thanks for your response, I think the method is more complicated than the current SOTA method, and the comparison between the main experiment and SOTA method uses unofficial code and does not align the experimental settings. At this stage, I will modify my score to 5.

---

> ### Author Response · Authors · 2023-11-22
>
> Dear Reviewer gMyA,
>
> Thanks for your valuable feedback. While we believe that our analysis using two different settings offers a more comprehensive understanding, we are more than happy to provide more results that align with the settings used in PatchTST. We hope these results will address your concerns regarding the comparison to PatchTST.
>
> As mentioned in the Appendix, our 720 experiments use the codebase in Nie et al., 2023, i.e., the official PatchTST code. We have modified the input length from 720 to 336/512 to further align with the settings presented in Table 3 of the PatchTST paper. Due to resource limitations, we currently only completed the experiments on 5 out of the 7 datasets, and the results are summarized in the following tables:
>
>
>
> | 336 length | Etth1               | Etth2                | Ettm1                | Ettm2                | Weather              |
> |------------|---------------------|----------------------|----------------------|----------------------|----------------------|
> | PatchTST   | MSE  $\   $     MAE | MSE  $\   $      MAE | MSE  $\   $      MAE | MSE  $\   $      MAE | MSE  $\   $      MAE |
> | 96         | 0.375     0.399     | 0.274	0.336          | 0.290	0.342          | 0.165	0.255          | 0.152	0.199          |
> | 192        | 0.414     0.421     | 0.339	0.379          | 0.332	0.369          | 0.220	0.292          | 0.197	0.243          |
> | 336        | 0.431     0.436     | 0.331	0.380          | 0.366	0.392          | 0.278	0.329          | 0.249	0.283          |
> | 720        | 0.449     0.466    | 0.379	0.422          | 0.420	0.424          | 0.367	0.385          | 0.320	0.335          |
> | Average    | 0.417     0.431     | 0.331	0.379          | 0.352	0.382          | 0.258	0.315          | 0.230	0.265          |
> | CARD       | MSE  $\   $     MAE | MSE  $\   $      MAE | MSE  $\   $      MAE | MSE  $\   $      MAE | MSE  $\   $      MAE |
> | 96         | 0.366	0.389         | 0.285	0.336          | 0.284 0.329          | 0.162	0.247          | 0.149	0.190          |
> | 192        | 0.413	0.415         | 0.356	0.381          | 0.333	0.358          | 0.214	0.283          | 0.193 0.232          |
> | 336        | 0.428	0.425         | 0.347	0.384          | 0.364	0.377          | 0.263	0.315          | 0.247	0.274          |
> | 720        | 0.439	0.456         | 0.377 0.413          | 0.426	0.416          | 0.345	0.369          | 0.315	0.324          |
> | Average    | 0.411	0.421         | 0.341	0.379          | 0.352 0.370          | 0.246	0.303          | 0.226	0.255          |
>
>
>
>
> | 512 length | Etth1               | Etth2                | Ettm1                | Ettm2                | Weather              |
> |------------|---------------------|----------------------|----------------------|----------------------|----------------------|
> | PatchTST   | MSE  $\   $     MAE | MSE  $\   $      MAE | MSE  $\   $      MAE | MSE  $\   $      MAE | MSE  $\   $      MAE |
> | 96         | 0.370	0.400         | 0.274	0.337          | 0.293	0.346          | 0.166	0.265          | 0.149	0.198          |
> | 192        | 0.413	0.429         | 0.341	0.382          | 0.333	0.370          | 0.223	0.296          | 0.194	0.241          |
> | 336        | 0.422	0.440         | 0.329	0.384          | 0.369	0.392          | 0.274	0.329          | 0.245	0.292          |
> | 720        | 0.447	0.468         | 0.379	0.422          | 0.416	0.420          | 0.362	0.385          | 0.314	0.334          |
> | Average    | 0.413	0.434         | 0.331	0.381          | 0.353	0.382          | 0.256	0.319          | 0.226	0.266          |
> | CARD       | MSE  $\   $     MAE | MSE  $\   $      MAE | MSE  $\   $      MAE | MSE  $\   $      MAE | MSE  $\   $      MAE |
> | 96         | 0.369 0.395         | 0.273	0.332          | 0.286	0.330          | 0.160	0.246          | 0.146	0.187          |
> | 192        | 0.407 0.417         | 0.334	0.375          | 0.330	0.357          | 0.210	0.282          | 0.190	0.229          |
> | 336        | 0.418 0.427         | 0.328	0.380          | 0.366	0.376          | 0.261	0.317          | 0.240	0.270          |
> | 720        | 0.416 0.443         | 0.370	0.413          | 0.421  0.419         | 0.353	0.376          | 0.312	0.324          |
> | Average    | 0.402 0.420         | 0.326	0.375          | 0.352	0.371          | 0.246	0.305          | 0.222	0.252          |
>
>
>
> In both 336 and 512 settings, CARD gives very promising performance. Additionally, the average results across four different forecasting lengths show that CARD outperforms PatchTST in all experiments with 512 input length and only underperforms in one metric with 336 input length.

---

### Official Review · Reviewer_ajLB · 2023-11-10

**Soundness:** 3 good
**Presentation:** 3 good
**Contribution:** 3 good
**Rating:** 8
**Confidence:** 5

**Summary:**

The paper proposes a transformer architecture whose main feature seems to be the attention applied both across time dimension and across feature dimension.

**Strengths:**

- The architecture design seems novel and interesting
- Extensive ablation studies and additional results in Appendices
- I am happy to see, finally, that the transformer based paper discusses and outperforms results on basic baselines, most importantly
    - Zeng et al. Are transformers effective for time series forecasting? AAAI'23
- In general the paper seems to be well written and clear. I like the level of detail and clarity of Figures 1,2

**Weaknesses:**

- The paper makes same methodological mistake as the previous line of this research. We really need to compare against basic baselines! Please include results of Zeng et al. in Table 1. Today, this is perhaps the most important baseline to include in the Table, so I'd rather not see some of the other entries in the table than to see this one missing
- The results in Table 2 are not as impressive as in Table 1. The model is not performing well compared to the first entry in the M4 competition and is barely performing better than the second entry, which is not reflected in Table 2. Again, same methodological mistake of not comparing comprehensively against basic baselines. Please include them in the table if you want to demonstrate methodological rigour and promote consistency across studies. The N-BEATS results are probably presented in the table without ensembling, but even the basic ensemble of 18 models is way better than the proposed model as follows from Figure 3 in https://arxiv.org/pdf/1905.10437.pdf. I suggest that the authors follow well accepted protocol for M4 benchmark evaluation and present the ensemble-based results, or remove Table 2 from the paper and replace it with the results of their ablation studies.

**Questions:**

- Please add results on ILI and Exchange to Table 1
- I flagged a few concerns and will revise my score if they are addressed appropriately

---

> ### Author Response · Authors · 2023-11-17
>
> We express our sincere gratitude to Reviewer ajLB for providing a positive assessment of the architecture and empirical contributions of our work. Your belief in the significance of our findings, especially in comparison to the claims made by previous work, serves as great encouragement for us. We deeply appreciate the detailed and insightful feedback you have provided, and we are fully committed to addressing your concerns.
>
> ## Q1: Including some early baselines
>
> We highly concur with the reviewer's suggestion that certain early baselines have not been thoroughly examined. Notably, algorithms like nbeats and even persistent models exhibit excellent performance across certain datasets. The reason for excluding them in this paper lies solely in the limitation of available space, as we were apprehensive that some reviewers may raise concerns regarding the omission of recent works. We have run the experiment on Nlinear, Linear, and Repeat models in our experimental settings and summarized the results in the following table. Our model consistently outperforms those baselines.  We assure you that we will incorporate these baselines in a comprehensive table, presented in smaller font size, in the refined version of our manuscript.
>
>  | Dataset  | ETTm1   | ETTm2   | ETTh1   | ETTh2   | Wea.   | ECL   | Traffic |
>  | -------- | ----- | ----- | ----- | ----- | ----- | ----- | ---|
>  | -------- | MSE | MSE | MSE | MSE | MSE | MSE | MSE|
>  | Nlinear      | 0.429    | 0.291     | 0.602     | 0.413  |0.281   | 0.234    | 0.641    |
>  | Linear       | 0.432    |0.374     | 0.625     | 0.649   |0.271  | 0.224    | 0.633 |
>  | Repeat     | 1.269    | 0.385    |1.321      | 0.536      | 0.353 | 1.612 |  2.770|
>  | CARD      | 0.383 | 0.271 | 0.443 | 0.367 | 0.242 | 0.169 |0.450|
>
> ## Q2: About the short-term forecasting M4 benchmark
>
> After careful consideration, we align with the reviewer's evaluation. In order to test the M4 dataset, we have incorporated the setting proposed in the recent Timesnet paper. However, it is worth noting that as M4 is a competition dataset, most participants employ ensemble methods to enhance performance. Notably, the Nbeats algorithm has exhibited exceptional performance on this dataset, because of its 180 models ensembling trick. The original paper introduces structural and parameter variations to diversify the base model, optimizing it for ensembling. The success of ensembling depends on two crucial factors: the ensembling method itself and the choice of baseline algorithms. It is important to note that different ensembling methods can be better suited for different datasets and may facilitate varying baseline algorithms. Although incorporating such a comparison would undoubtedly introduce complexity, we acknowledge the need to rerun and construct an ensemble framework for all baseline models, including the card method proposed in our study. However, it is important to note that even with this comparison, ensuring absolute fairness may not be guaranteed, as a particular ensemble method might inherently favor a specific baseline model. We recognize that this may deviate from our original narrative, as a model that excels in ensembling may not necessarily outperform others on its own. Ensembling relies heavily on the diversity among baseline models. Additionally, as raised by another reviewer, eTmJ, it is important to acknowledge that certain baseline models in the M4 dataset utilize different input lengths due to the competition's lack of requirement for uniform input lengths. Consequently, we believe your suggestion is highly fitting and thoughtful. We propose relocating this portion to the supplementary information, treating it as a preliminary exploration, and highlighting the significantly improved performance of algorithms like Nbeats when utilized within an ensemble framework.

---

> ### Author Response · Authors · 2023-11-17
>
> ## Q3: Adding ILI and Exchange datasets
> We have included their results in the following table. Although we did run these two datasets and found that the card's performance was good, we were hesitant to consider them as suitable benchmarks. The ILI dataset is both small and highly diverse, with normalized predicted MSE values close to 2. We are uncertain whether the relatively good performance on such a dataset can be extrapolated to larger and more practical datasets in real-world scenarios. Similarly, when it comes to the Exchange dataset, its nature as a financial dataset governed by free-market principles makes it extremely challenging to make accurate predictions. The signal-to-noise ratio in this dataset is significantly low, posing challenges for short-term forecasting tasks, and rendering long-term forecasting even more arduous, if not seemingly impossible. For example, all baselines in TimesNet paper fail to beat even the simple repeat forecasting reported in Zeng et al.  2023. In the final version, we will include them in the main table.
>
>
> |     | exchange |       |     | illness |       |
> | --- | -------- | ----- | --- | ------- | ----- |
> | Output length | mse | mae | Output length | mse | mae |
> | 96  | 0.084    | 0.202 | 24  | 1.665   | 0.803 |
> | 192 | 0.174    | 0.295 | 36  | 2.200   | 0.890 |
> | 336 | 0.323    | 0.295 | 48  | 1.875   | 0.821 |
> | 720 | 0.841    | 0.689 | 60  | 1.923   | 0.853 |
> | avg | 0.356    | 0.371 | avg | 1.916   | 0.842 |

---

> > ### Author Response · Authors · 2023-11-21
> > **Response to Reviewer ajLB before the end of discussion**
> >
> > Dear Reviewer ajLB,
> >
> > Since the End of author/reviewer discussions is just in one day, may we know if our response addresses your main concerns? If so, we kindly ask for your reconsideration of the score. Should you have any further advice on the paper and/or our rebuttal, please let us know and we will be more than happy to engage in more discussion and paper improvements.
> >
> > Thank you so much for devoting time to improving our paper!

---

### Comment · Area_Chair_5YBn · 2023-11-23
**From AC at the end of rebuttal: Reviewer response required**

Dear Reviewers,

Thanks for your time and commitment to the ICLR 2024 review process.

As we approach the conclusion of the author-reviewer discussion period (Wednesday, Nov 22nd, AOE), I kindly urge those who haven't engaged with the authors' dedicated rebuttal to please take a moment to review their response and share your feedback, regardless of whether it alters your opinion of the paper.

Your feedback is essential to a thorough assessment of the submission.

Best regards,

AC

---

### Meta-Review · Area_Chair_5YBn · 2023-12-11

**Metareview:**

This paper proposed a transformer-based method for time series forecasting, with a new channel-aligned attention structure to capture temporal and variable correlations. The authors performed an effective rebuttal, which addressed most of the concerns with post-rebuttal scores raised. The AC's takes on the paper are as follows: 1. The originality of this work is not overwhelming since the modules are relatively common in time series domain; 2. The final combined model is way too complex, which may not be used as a workhorse in the future; 3. The evaluation is not wholly rigorous since the settings and protocols are not consistent. While AC opted to follow the majority votes and recommended acceptance, the authors are advised to improve the paper according to the meta-review, especially, consolidate the experiments with fair and consistent settings and protocols, otherwise this field will be flooded with results that cannot be compared.

**Justification For Why Not Higher Score:**

Majority votes for acceptance, but with limitations in novelty, elegance, and consistency of evaluation.

**Justification For Why Not Lower Score:**

Majority votes of acceptance, reasonable since the paper has highlighted the importance of correlation modeling across both timepoints and variables.

---

### Decision · Program_Chairs · 2024-01-16

Accept (poster)